



# Pan-European rural atmospheric monitoring network shows dominance of NH₃ gas and NH₄NO₃ aerosol in inorganic pollution load

Y. Sim Tang[1], Chris R. Flechard[2], Ulrich Dämmgen[3], Sonja Vidic[4], Vesna Djuricic[4], Marta Mitosinkova[5], Hilde T. Uggerud[6], Maria J. Sanz[7,8,9], Ivan Simmons[1], Ulrike Dragosits[1], Eiko Nemitz[1], Marsailidh Twigg[1], Netty van Dijk[1], Yannick Fauvel[2], Francisco Sanz-Sanchez[7], Martin Ferm[10], Cinzia Perrino[11], Maria Catrambone[11], David Leaver[1], Christine F. Braban[1], J. Neil Cape[1], Mathew R. Heal[12] & Mark A. Sutton[1]

[1]UK Centre for Ecology & Hydrology (UKCEH), Bush Estate, Penicuik, Midlothian EH26 0QB, UK

[2]French National Research Institute for Agriculture, Food and Environment (INRAE), UMR 1069 SAS, 65 rue de St-Brieuc, 35042 Rennes Cedex, France

[3]von Thunen Institut (vTI), Bundesallee 50, 38116 Braunschweig, Germany

[4]Meteorological and Hydrological Service of Croatia (MHSC), Research and Development Division, Gric 3, 10000 Zagreb, Croatia

[5]Slovak Hydrometeorological Institute (SHMU), Department of Air Quality, Jeseniova 17, 833 15 Bratislava, Slovak Republic

[6]Norwegian Institute for Air Research (NILU), P.O.Box 100, N-2027 Kjeller, Norway

[7]The Mediterranean Center for Environmental Studies (Fundación CEAM), Parque Tecnológico, C/Charles H. Darwin 14, 46980 Paterna (Valencia), Spain

[8]Basque Centre for Climate Change, Sede Building 1, Scientific Campus of the University of the Basque Country, 48940, Leioa, Bizkaia, Spain

[9]Ikerbasque, Basque Science Foundation, María Díaz Haroko Kalea, 3, 48013 Bilbo, Bizkaia, Spain

[10]IVL Swedish Environmental Research Institute, P.O. Box 5302, S-400 14, Gothenburg, Sweden

[11]C.N.R. Institute of Atmospheric Pollution Research, via Salaria Km. 29, 300 – 00015, Monterotondo st, Rome, Italy

[12]School of Chemistry, University of Edinburgh, David Brewster Road, Edinburgh EH9 3FJ, UK

*Correspondence to*: Y Sim Tang (yst@ceh.ac.uk)





**Abstract**

A comprehensive European dataset on monthly atmospheric $NH_3$, acid gases ($HNO_3$, $SO_2$, $HCl$) and aerosols ($NH_4^+$, $NO_3^-$, $SO_4^{2-}$, $Cl^-$, $Na^+$, $Ca^{2+}$, $Mg^{2+}$) is presented and analyzed. Speciated measurements were made with a low-volume denuder and filter pack method (DELTA®) as part of the EU NitroEurope (NEU) integrated project. Altogether, there were 64 sites in 20 countries (2006-2010), coordinated between 7 European laboratories. Bulk wet deposition measurements were carried out at 16 co-located sites (2008-2010). Inter-comparisons of chemical analysis and DELTA® measurements allowed an assessment of comparability between laboratories.

The form and concentrations of the different gas and aerosol components measured varied between individual sites and grouped sites according to country, European regions and 4 main ecosystem types (crops, grassland, forests and semi-natural). Smallest concentrations (with the exception of $SO_4^{2-}$ and $Na^+$) were in Northern Europe (Scandinavia), with broad elevations of all components across other regions. $SO_2$ concentrations were highest in Central and Eastern Europe with larger $SO_2$ emissions, but particulate $SO_4^{2-}$ concentrations were more homogeneous between regions. Gas-phase $NH_3$ was the most abundant single measured component at the majority of sites, with the largest variability in concentrations across the network. The largest concentrations of $NH_3$, $NH_4^+$ and $NO_3^-$ were at cropland sites in intensively managed agricultural areas (e.g. Borgo Cioffi in Italy), and smallest at remote semi-natural and forest sites (e.g. Lompolojänkkä, Finland), highlighting the potential for $NH_3$ to drive the formation of both $NH_4^+$ and $NO_3^-$ aerosol. In the aerosol phase, $NH_4^+$ was highly correlated with both $NO_3^-$ and $SO_4^{2-}$, with a near 1:1 relationship between the equivalent concentrations of $NH_4^+$ and sum ($NO_3^-$ + $SO_4^{2-}$), of which around 60% was as $NH_4NO_3$.

Distinct seasonality were also observed in the data, influenced by changes in emissions, chemical interactions and the influence of meteorology on partitioning between the main inorganic gases and aerosol species. Springtime maxima in $NH_3$ were attributed to the main period of manure spreading, while the peak in summer and trough in winter were linked to the influence of temperature and rainfall on emissions, deposition and gas-aerosol phase equilibrium. Seasonality in $SO_2$ were mainly driven by emissions (combustion), with concentrations peaking in winter, except in Southern Europe where the peak occurred in summer. Particulate $SO_4^{2-}$ showed large peaks in concentrations in summer in Southern and Eastern Europe, contrasting with much smaller peaks occurring in early spring in other regions. The peaks in particulate $SO_4^{2-}$ coincided with peaks in $NH_3$ concentrations, attributed to the formation of the stable $(NH_4)_2SO_4$. $HNO_3$ concentrations were more complex, related to traffic and industrial emissions, photochemistry and $HNO_3$:$NH_4NO_3$ partitioning. While $HNO_3$ concentrations were seen to peak in the summer in Eastern and Southern Europe (increased photochemistry), the absence of a spring peak in $HNO_3$ in all regions may be explained by the depletion of $HNO_3$ through reaction with surplus $NH_3$ to form the semi-volatile aerosol $NH_4NO_3$. Cooler, wetter conditions in early spring favour the formation and persistence of $NH_4NO_3$ in the aerosol phase, consistent with the higher springtime concentrations of $NH_4^+$ and $NO_3^-$. The seasonal profile of $NO_3^-$ was mirrored by $NH_4^+$, illustrating the influence of gas:aerosol partitioning of $NH_4NO_3$ in the seasonality of these components.

Gas-phase $NH_3$ and aerosol $NH_4NO_3$ were the dominant species in the total inorganic gas and aerosol species measured in the NEU network. With the current and projected trends in $SO_2$, $NO_x$ and $NH_3$ emissions, concentrations of $NH_3$ and $NH_4NO_3$ can be expected to continue to dominate the inorganic pollution load over the next decades, especially $NH_3$ which is linked to substantial exceedances of ecological thresholds across Europe. The shift from $(NH_4)_2SO_4$ to an atmosphere more abundant in $NH_4NO_3$ is expected to maintain a larger fraction of reactive N in the gas phase by partitioning to $NH_3$ and $HNO_3$ in warm weather, while $NH_4NO_3$ continues to contribute to exceedances of air quality limits for $PM_{2.5}$.



## 1 Introduction

Air quality policies and research on atmospheric sulfur (S) and nitrogen (N) pollutant impacts on ecosystem and human health have focused on the emissions, concentrations and depositions of sulfur dioxide ($SO_2$), nitrogen oxides ($NO_x$), ammonia ($NH_3$) and their secondary inorganic aerosols (SIAs: ammonium sulfate, $(NH_4)_2SO_4$; ammonium nitrate, $NH_4NO_3$) (ROTAP, 2012;

EMEP, 2019). The aerosols, formed through neutralisation reactions between the alkaline $NH_3$ gas and acids generated in the atmosphere by the oxidation of $SO_2$ and $NO_x$ (Huntzicker et al., 1980; AQEG, 2012) are a major component of fine particulate matter ($PM_{2.5}$) (AQEG, 2012; Vieno et al., 2016a) and precipitation (ROTAP, 2012; EMEP, 2019).

The negative effects of these pollutants on sensitive ecosystems are mainly through acidification (excess acidity) and
eutrophication (excess nutrient N) processes that can lead to a loss of key species and decline in biodiversity (e.g. Hallsworth et al., 2010; Stevens et al., 2010). They are also implicated in radiative forcing, and influence climate change through inputs of nitrogen that can alter the carbon cycle (Reis et al., 2012; Sutton et al., 2013; Zaehle & Dalmonech, 2011).

A number of EU policy measures (e.g. 2008/50/EC Ambient Air Quality Directive, EU, 2008; 2016/2284/EU National
Emissions Ceilings Directive NECD, EU, 2016) and wider international agreements (e.g. Gothenburg protocol; UNECE, 2012) are targeted at abating the emissions and environmental impacts of $SO_2$, $NO_x$ and $NH_3$. The largest emissions reductions have been achieved for $SO_2$, which decreased by 82 % across the EEA-33 since 1990, to 4743 kt $SO_2$ in 2017 (EEA, 2019). Reductions in $NO_x$ emissions have been more modest, at 45 % over the same period, with emissions in 2017 of 8563 kt $NO_x$ exceeding those of $SO_2$. By contrast, the reductions in $NH_3$ emissions (of which over 90% come from agriculture) have been
more modest, decreasing by only 18 %. Here, the decrease was largely driven by reductions in fertiliser use and livestock numbers, in particular from eastern European countries, rather than through implementation of any abatement or mitigation measures. More worryingly, the decreasing trend has reversed in recent years, with emissions increasing by 5 % since 2010, to 4788 kt $NH_3$ in 2017 (EEA, 2019).

In recent assessments, critical loads of acidity were exceeded in about 5 % of the ecosystem area across Europe in 2017 (EMEP, 2018). While the substantial decline in $SO_2$ emissions has allowed the recovery of ecosystems from acid rain, $NH_3$ from agriculture and $NO_x$ from transport are increasingly contributing to a larger fraction of the acidity load. Although $NH_3$ is not an acid gas, nitrification of $NH_3$ and ammonium ($NH_4^+$) releases hydrogen ions ($H^+$) that acidify soils and freshwater. The deposition of reactive N ($N_r$, including oxidised N: $NO_x$, $HNO_3$, $NO_3^-$ and reduced N: $NH_3$, $NH_4^+$) and their contribution to
eutrophication effects have also been identified by the EEA as the most important impact of air pollutants on ecosystems and biodiversity (EEA, 2019). The deposition of $N_r$ throughout Europe remains substantially larger than the level needed to protect ecosystems, with critical loads thresholds for eutrophication from N exceeded in around 62 % of the EU-28 ecosystem area and in almost all countries in Europe in 2017 (EMEP, 2018).

Following emission, atmospheric transport and fate of the gases are controlled by the following processes: short range dispersion and deposition, chemical reaction and formation of $NH_4^+$ aerosols, and the long-range transport and deposition of the aerosols (Sutton et al., 1998; ROTAP, 2012). Atmospheric S and $N_r$ inputs from the atmosphere to the biosphere occur though i) dry deposition of gases and aerosols, ii) wet deposition in rain, and iii) occult deposition in fog and cloud (Smith et al., 2000; ROTAP, 2012). The deposition processes contribute very different fractions of the total S or $N_r$ input and different
chemical forms of the pollutants at different spatial scales. $NH_3$ is a highly reactive, water-soluble gas and deposits much faster than $NO_x$ (which is not very water soluble and has low deposition velocity). Dry N deposition by $NH_3$ therefore contributes a significant fraction of the total N deposition to receptors close to source areas and will often exert the larger ecological impacts, compared with other N pollutants (Cape et al., 2004; Sutton et al., 1998, 2007). Numerous studies have shown that $N_r$





deposition in the vicinity of NH₃ sources is dominated by dry NH₃-N deposition (e.g. Pitcairn et al., 1998; Sheppard et al., 2011), with removal of NH₃ close to a source controlled by physical, chemical and ecophysiological processes (Flechard et al., 2011; Sutton et al., 2007, 2013). Unlike NOₓ, HNO₃ (from oxidation of NOₓ) is very water-soluble, while NO₃⁻ particles can act as cloud condensation nuclei (CCN) so that they are both scavenged quickly and removed efficiently by precipitation.

Since NOₓ is inefficiently removed by precipitation, wet deposition of NOₓ near a source is small and only becomes important after NOₓ has been converted to HNO₃ and NO₃⁻.

Because of the large numbers of atmospheric N species and their complex atmospheric chemistry, quantifying the deposition of Nᵣ is hugely complex and is a key source of uncertainty for ecosystems effects assessment (Bobbink et al., 2010; Fowler et

al., 2007; Schrader et al., 2018; Sutton et al., 2007). Input by dry deposition can be estimated using a combination of measured and/or modelled concentration fields with high-resolution inferential models (e.g. Smith et al., 2000; Flechard et al., 2011), or by making direct flux measurements (e.g. Fowler et al., 2001; Nemitz et al., 2008). Although it is possible to measure Nᵣ deposition directly (e.g. Skiba et al., 2009), the flux measurement techniques are complex and resource intensive, unsuited to routine measurements at a large number of sites. The 'inferential' modelling approach provides a direct estimation of

deposition from Nᵣ measurements by applying a land-use dependent deposition velocity ($V_d$) to measured concentrations (Dore et al., 2015; Flechard et al., 2011; Simpson et al., 2006; Smith et al., 2000).

At present, there are limited atmospheric measurements that speciate the gas and aerosol phase components at multiple sites over several years. On a European scale, atmospheric measurements of sulfur (SO₂, particulate SO₄²⁻) and nitrogen (NH₃,

HNO₃, particulate NH₄⁺, NO₃⁻) have been made by a daily filter pack method across the European Monitoring and Evaluation Program (EMEP) networks since 1985, providing data for evaluating wet and dry deposition models (EMEP, 2016; Torseth et al., 2012). The method, however, does not distinguish between the gas and aerosol phase N species. Consequently, these data are reported as total inorganic ammonium (TIA = sum of NH₃ and NH₄⁺) and total inorganic nitrate (TIN = sum HNO₃ and NO₃⁻), limiting the usefulness of the data. Speciated measurements by an expensive and labour-intensive daily annular denuder

method are also made (Torseth et al., 2012), but are necessarily restricted to a small number of sites, due to the high costs associated with this type of measurement. There are also networks with a focus on specific N components, for example, the national NH₃ monitoring networks in the Netherlands (LML, van Zanten et al., 2017) and in the UK (National Ammonia Monitoring Network, NAMN; Tang et al., 2018a), or compliance monitoring across Europe in the case of SO₂ and NOₓ. The UK is unique in having an extensive set of speciated gas and aerosol monitoring data from the Acid Gas and Aerosol Network

(AGANet), with measurements from 1999 to the present (Tang et al., 2018b).

In this context, there is an ongoing need for cost-effective, easy-to-operate, time-integrated atmospheric measurement for the respective gas and aerosol phases at sufficient spatial scales. Such data would help to, 1) improve estimates of N deposition, 2) contribute to development and validation of long-range transport models, e.g. EMEP (Simpson et al., 2006) and EMEP4UK

(Vieno et al., 2014, 2016), 3) interpret interactions between the gas and aerosol phases, and 4) interpret ecological responses to nitrogen (e.g. ecosystem biodiversity or net carbon exchange). To contribute to this goal, a '3-level' measurement strategy in the EU Framework Programme 6 Integrated Project "NitroEurope" (NEU, http://www.nitroeurope.ceh.ac.uk/) between 2006 and 2010 delivered a comprehensive integrated assessment of the nitrogen cycle, budgets and fluxes for a range of European terrestrial ecosystems (Sutton et al., 2007; Skiba et al., 2009). At the most intensive level (Level 3), state-of-the-art

instrumentation for high resolution, continuous measurements at a small number of 13 'flux super sites' provided detailed understanding on atmospheric and chemical processes (Skiba et al., 2009). By contrast, manual methods with a low temporal frequency (monthly) at the basic level (Level 1) provided measurements of Nᵣ components at a large number of sites (> 50





sites) in a cost-efficient way in a pan-European network (Tang et al., 2009). Key species of interest included NH₃, HNO₃ and
ammonium aerosols ((NH₄)₂SO₄, NH₄NO₃).

In this paper, we present and discuss four years of monthly reactive gas (NH₃, HNO₃, HCl) and aerosol (NH₄⁺, NO₃⁻, SO₄²⁻,
Cl⁻, Na⁺, Ca²⁺, Mg²⁺) measurements from the Level 1 network set up under the NEU integrated project, complemented by two
years of bulk wet deposition data made at a subset of the network sites (Figure 1). A harmonised measurement approach with
a simple, cost-efficient time-integrated method, applied with high spatial coverage allowed a comprehensive assessment across
Europe. Measurements across the network were coordinated between multiple European laboratories. The measurement
approach and the operations of the networks, including the implementation of annual inter-comparisons to assess comparability
between the laboratories, are described. The data are discussed in terms of spatial and temporal variation in concentrations,
relative contribution of the inorganic nitrogen and sulfur components to the inorganic pollution load, and changes in
atmospheric concentrations of acid gases and their interactions with NH₃ gas and NH₄⁺ aerosol.

<INSERT FIGURE 1>

**2   Methods**

**2.1   NEU Level 1 DELTA® network**

The NitroEurope (NEU) Level 1 network was operated between November 2006 and December 2010 to deliver the core
measurements of reactive nitrogen gases (NH₃, HNO₃) and aerosols (NH₄⁺, NO₃⁻) for the project (Figure 1). A low-volume
denuder-filter pack method, the 'DEnuder for Long-Term Atmospheric sampling' system (DELTA®, Sutton et al., 2001a; Tang
et al., 2009, 2018b) with time-integrated monthly sampling was used, which made implementation at a large number of sites
possible. Other acid gases (SO₂, HCl) and aerosols (SO₄²⁻, Cl⁻, Na⁺, Ca²⁺, Mg²⁺) were also collected at the same time and
measured by the DELTA® method. DELTA® measurements were co-located with all NEU Level 3 sites with advanced flux
measurements (Skiba et al., 2009), and with the network of main CarboEurope-IP CO₂ flux monitoring sites
(www.carboeurope.eu) (Flechard et al., 2011, 2020). Two of the UK sites in the NEU DELTA® network are existing UK
NAMN (Tang et al., 2018a) and AGANet sites (Tang et al., 2018b). These are Auchencorth Moss (UK-Amo) and Bush (UK-
EBu) located in Southern Scotland. Monthly gas and aerosol data at the two sites, made as part of the UK national networks,
were included in the NEU network. NEU network Nᵣ data were used, together with a range of dry deposition models, to model
dry deposition fluxes (Flechard et al., 2011) and to assess the influence of Nᵣ on the C cycle, potential C sequestration and the
greenhouse gas balance of ecosystems using CO₂ exchange data from the co-located CarboEurope sites (Flechard et al., 2020).
Other measurements made at the Level 1 sites included estimation of wet deposition fluxes (Sect. 2.3) and also soil and plant
bioassays (Schaufler et al., 2010).

Altogether, the DELTA® network covered a wide distribution of sites across 20 countries and 4 major ecosystem types: crops,
grassland, semi-natural and forests. These sites can be described as 'rural', and were chosen to provide a regionally
representative estimate of air composition. The network site map is shown in Figure 2, with site details given in Supp. Table
S1. Further information on the network sites are also provided in Flechard et al. (2011). Network establishment started in
November 2006, with 57 sites operational from March 2007 onwards. Over the course of the network, some sites closed or
were relocated due to infrastructure changes and new sites were also added. A total of 64 sites provided measurements at the
end of the project, with 45 of the sites operational the entire time. In addition, replicated DELTA® measurements were made
at 4 sites:





1) Auchencorth Moss parallel (P) (UK-AMoP; $NH_3/NH_4^+$ measured only)

2) Easter Bush parallel (P) (UK-EBuP; same method as main site),

3) SK04 parallel (P) (SK04P; same method as main site).

4) Fougéres parallel (P) (FR-FgsP: different sample train with 2 x NaCl coated denuders instead of 2 x $K_2CO_3$/Glycerol
5   coated denuders to capture $HNO_3$; see Sect. 2.2.3) from February to December 2010 only.

**<INSERT FIGURE 2>**

### 2.1.1    Coordinating laboratories

10   A team of seven European laboratories shared responsibility for running the network. Measurement was on a monthly
time scale, with each laboratory preparing and analysing the monthly samples with documented analytical methods for between
and 16 DELTA sites (Figure 2). The use of a harmonised DELTA® methodology, coupled to defined quality protocols (Tang
et al., 2009) ensured comparability of data between the laboratories (see later in Sect. 3.1 and Sect. 3.2). A network of local
site operators representing the science teams of each site performed the monthly sample changes and posted the exposed
samples back to their designated laboratories for analysis. Air concentration data were submitted by the laboratories for their
respective sites in a standard reporting template to UKCEH. Following data checks against defined quality protocols (Tang et
al., 2009), the finalised dataset was uploaded to the NEU database (http://www.nitroeurope.ceh.ac.uk/). Establishment of the
network, including the first year of measurement results on $N_r$ components are reported in Tang et al. (2009). Information on
co-located measurements and agricultural activities at each of the sites were also collected and are accessible from the NEU
website (http://www.nitroeurope.ceh.ac.uk/).

### 2.2    DELTA® methodology

The DELTA® method used in the NEU Level 1 network is based on the system developed for the UK Acid Gas and Aerosol
monitoring network (AGANet, Tang et al., 2018b). Full details of the DELTA® method and air concentration calculations in
the NEU network are provided by Tang et al. (2009, 2018b). The method uses a small 6 V air pump to deliver low air sampling
rates of between 0.2 to 0.4 L min$^{-1}$, a high sensitivity gas meter to record the typically monthly volume of air collected and a
DELTA® denuder-filter pack sample train to collect separately the gas and aerosol phase components. The sample train is
made up of two pairs of base and acid impregnated denuders (15 cm and 10 cm long) to collect acid gases and $NH_3$,
respectively, under laminar conditions. A 2-stage filter pack with base and acid coated cellulose filters collects the aerosol
components downstream of the denuders. The base coating used was $K_2CO_3$/glycerol which is effective for the simultaneous
collection of $HNO_3$, $SO_2$ and HCl (Ferm, 1986), while the acid coating was either citric acid for temperate climates or
phosphorous acid for Mediterranean climates (Allegrini et al., 1987; Ferm, 1979; Perrino et al., 1999; Fitz, 2002). In this way,
artefacts between gas and aerosol phase concentrations are minimized (Ferm et al., 1979; Sutton et al., 2001a). The DELTA®
air inlet has a particle cut-off of ~ 4.5 µm which means fine mode aerosols in the $PM_{2.5}$ fraction and some of the coarse mode
aerosols < $PM_{4.5}$ will be collected (Tang et al., 2015).

A low voltage version of the AGANet DELTA® system was built centrally by UKCEH and sent to each of the European sites
where they were installed by local site contacts. These systems operated on either 6 V (off mains power with a transformer) or
12 V from batteries (wind and solar powered). Air sampling was direct from the atmosphere without any inlet lines or filters
40   to avoid potential loss of components, in particular $HNO_3$ that is very "sticky", to surfaces. Sampling height was 1.5 m above
ground/vegetation in open areas. In forested areas, the DELTA® equipment was set up either in large clearings, or on towers
at 2 – 3 m above the canopy (see Flechard et al., 2011).



### 2.2.1 Calculation of gas and aerosol concentrations

Atmospheric gas and aerosol concentrations in the DELTA® method are calculated from the amount of inorganic ions ($NH_4^+$, $NO_3^-$, $SO_4^{2-}$, $Cl^-$, and base cations) in the denuder/aerosol aqueous extracts and the volume of air sampled (from gas meter readings), which is typically 15 $m^3$ for a monthly sample. The volume of deionised water used to extract acid coated denuders and aerosols filters are 3 mL and 4 mL, respectively. For the base coated denuders and aerosol filters, the extract volume in both cases is 5 mL An example is shown here for calculating the atmospheric concentrations of $NH_3$ (gas) (Equation 1) and $NH_4^+$ (aerosol) (Equation 2) from the aqueous extracts, based on an air volume of 15 $m^3$ collected in a typical month.

$$\text{Gas } NH_3 (\mu g\, m^{-3}) = \frac{NH_4^+ (mg\, L^{-1})\, [sample-blank]\, x\, 3\, mL\, x\, (\frac{17}{18})}{15\ m^3} \qquad [1]$$

$$\text{Particle } NH_4^+ (\mu g\, m^{-3}) = \frac{NH_4^+ (mg\, L^{-1})\, [sample-blank]\, x\, 4\, mL}{15\ m^3} \qquad [2]$$

Pairs of base and acid coated denuders are used to collect the acid gases and alkaline $NH_3$ gas, respectively. This allows denuder collection efficiency of, for example, $NH_3$ (Equation 3) to be assessed as part of the data quality assessment process. An imperfect acid coating on the denuders for example can lead to lower capture efficiencies (Sutton et al., 2001a; Tang et al., 2003).

$$\text{Denuder collection efficiency, } NH_3\ (\%) = 100\ x\ \frac{NH_3\ (Denuder\ 1)}{NH_3\ (Denuder\ 1 + Denuder\ 2)} \qquad [3]$$

A correction, based on the collection efficiency, is applied to provide a corrected air concentration ($\chi_a$ (corrected), Equation 4) (Sutton et al., 2001a; Tang et al., 2018a, 2018b). With a collection efficiency of 95 %, the correction amounts to 0.3 % of the corrected air concentration. For an efficiency below 60 %, the correction amounts to more than 50 % and is not applied. The air concentration of ($\chi_a$) of $NH_3$ is then determined as the sum of $NH_3$ in denuders 1 and 2 (Tang et al., 2018a). By applying the infinite series correction, the assumption is that any $NH_3$ (and other gases) that is not captured by the denuders will be collected on the downstream aerosol filter. To avoid double counting, the estimated amount of 'NH$_3$ breakthrough' is subtracted from the $NH_4^+$ concentrations on the aerosol filter.

$$\chi_a\ (corrected) = \chi_a\ (Denuder\ 1)\ *\ \frac{1}{1 - \left[\frac{\chi_a (Denuder\ 2)}{\chi_a (Denuder\ 1)}\right]} \qquad [4]$$

### 2.2.2 Estimating sea salt and non-sea salt $SO_4^{2-}$ (ss-$SO_4^{2-}$ and nss-$SO_4^{2-}$)

Sea salt $SO_4^{2-}$ (ss-$SO_4^{2-}$) in aerosol was estimated according to Equation 5, based on the ratio of the mass concentrations of $SO_4^{2-}$ to the reference $Na^+$ species in seawater (Keene et al., 1986; O'Dowd and de Leeuw, 2007).

$$[ss\text{-}SO_4^{2-}]\ (\mu g\, ss\text{-}SO_4^{2-}\ m^{-3}) = 0.25\ x\ [Na^+]\ (\mu g\, Na^+\ m^{-3}) \qquad [5]$$

Non-sea salt $SO_4^{2-}$ (nss-$SO_4^{2-}$) was then derived as the difference between total measured $SO_4^{2-}$ and ss-$SO_4^{2-}$ (Equation 6).

$$[nss\text{-}SO_4^{2-}]\ (\mu g\, nss\text{-}SO_4^{2-}\ m^{-3}) = [SO_4^{2-}]\ (\mu g\, SO_4^{2-}\ m^{-3}) - [ss\_SO_4^{2-}]\ (\mu g\, ss\text{-}SO_4^{2-}\ m^{-3}) \qquad [6]$$





### 2.2.3    Artefact in HNO₃ determination

Results from the first DELTA® inter-comparison in the NEU network (Tang et al., 2009) (see also Sect. 2.5) and further work by Tang et al. (2015, 2018b) have shown that HNO₃ concentrations may be overestimated on the carbonate coated denuders used, due to co-collection of other oxidized nitrogen components, most likely from nitrous acid (HONO). In the UK AGANet,

HNO₃ data are corrected with an empirical factor of 0.45 derived by Tang et al. (2015). Since the correction factor for HNO₃ is uncertain (estimated to be ± 30 %) and derived for UK conditions, no attempt has been made to correct the HNO₃ data from the NEU network. The DELTA® method remained unchanged throughout the entire network operation and provided a consistent set of measurements by the same protocol. The caveat is that the HNO₃ data presented in this paper also includes an unknown fraction of oxidized N, most probably HONO, and therefore represents an upper limit in the determination of

HNO₃. Contribution from NO₂ is likely to be small, since this is collected with a low efficiency on carbonate coated denuders (Bai et al., 2003; Tang et al., 2015) and the network sites are rural, where NOₓ concentrations are expected to be in the low ppbs. At the French Fougéres parallel site (FR-FgsP), NaCl coated denuders were used to measure HNO₃, to compare with results from $K_2CO_3$/glycerol coated denuders at the main site (FR-Fgs) (see Sect. 2.1).

### 2.3    NEU Bulk wet deposition network

The NEU bulk wet deposition network (Figure 3, Supp. Table S2) was established to provide wet deposition data on NH₄⁺ and NO₃⁻. It was set up two years after the establishment of the NEU DELTA® network, with sites located at a subset of DELTA® sites that did not already have on-site wet deposition measurements. Sampling commenced at some sites in January 2008, with 14 sites operational from March 2008. Site changes also occurred during the operation of this network, again with some site

closures and new site additions over time. In total, 12 sites provided 2 years of monthly data, with a further 6 sites providing 1 year of monthly data between 2008 to October 2010 when measurements ended.

**\<INSERT FIGURE 3\>**

The type of bulk precipitation collector used was a Rotenkamp sampler (Dämmgen et al., 2005), mounted 1.5 m above ground, or in the case of forest sites, either in clearings or above the canopy. Each unit has two collectors providing replicated samples, comprising of a pyrex glass funnel (aperture area = 84.9 cm²) with vertical sides, connected directly to a 3 L collection bottle (material = low density polyethylene) which was changed monthly. Thymol (5-methyl-2-(1-methylethyl)phenol) (150 mg) was added as a biocide (Cape et al., 2012) to a clean, dry pre-weighed bottle at the start of each collection period. This provided

a minimum thymol concentration of 50 mg L⁻¹ for a full bottle to preserve the sample against biological degradation of labile nitrogen compounds during the month-long sampling.

Three European laboratories shared management and chemical analysis for the network (Figure 3). The laboratories were CEAM (all 3 Spanish sites), INRAE (French Renon site) and SHMU, designated the main laboratory responsible for all other

sites. A full suite of precipitation chemistry analyses were carried out that included: pH, conductivity, NH₄⁺, NO₃⁻, SO₄²⁻, PO₄³⁻, Cl⁻, Na⁺, K⁺, Ca²⁺ and Mg²⁺. Rain volumes and precipitation chemistry data were submitted in a standard template to UKCEH for checking and then uploaded to the NEU database (http://www.nitroeurope.ceh.ac.uk/). Samples with high P (> 1 µg L⁻¹ PO₄³⁻), high K⁺ and/or NH₄⁺ values that are indicative of bird contamination were rejected. Annual wet deposition (e.g. kg N ha⁻¹ yr⁻¹) were estimated from the product of the species concentrations and rain volume. Determinations of organic N were

also carried out on some of the rain samples in a separate investigation reported by Cape et al. (2012).



### 2.4 Laboratory inter-comparisons: chemical analysis

All laboratories in the DELTA® and bulk wet deposition networks participated in water chemistry proficiency testing (PT) schemes in their own countries, as well as the EMEP (once annual, http://www.emep.int) and/or WMO-GAW (twice annual, http://www.qasac-americas.org/lab_ic.html) laboratory inter-comparison schemes. PT samples for analysis are synthetic precipitation samples for determination of pH, conductivity and all the major inorganic ions at trace levels. In addition, UKCEH also organised an annual PT scheme for the duration of the project (NEU-PT) to compare laboratory performance in the analysis of inorganic ions at higher concentrations relevant for DELTA® measurements. This comprised the distribution of reference solutions containing known concentrations of ions that were analysed by the laboratories as part of their routine analytical procedures.

### 2.5 Laboratory inter-comparisons: DELTA measurements

Prior to the NEU DELTA® network establishment, a workshop was held to provide training to participating laboratories on sample preparation and analysis. This was followed by a 4-month inter-comparison exercise (July to October 2006) between six laboratories at four test sites (Montelibretti, Italy; Braunschweig, Germany; Paterna, Spain, and Auchencorth, UK). Results of the inter-comparison on $N_r$ components were reported by Tang et al. (2009), which demonstrated good agreement under contrasting climatic conditions and atmospheric concentrations of the $N_r$ gases and aerosols. The first DELTA® inter-comparison allowed the new laboratories to gain experience in making measurements, and was an extremely useful exercise to check how the whole system works, starting with coating of denuders and filters and DELTA® train preparation, sample exchange *via* post, sample handling and inter-comparing laboratory analytical performance. Further DELTA® inter-comparisons between laboratories were conducted each year for the duration of the project, details of which are summarised in Table 1. At each test site, DELTA® systems were randomly assigned to each of the participating laboratories. All laboratories provided DELTA® sampling trains for each of the inter-comparison sites and carried out chemical analysis on the returned exposed samples. Measurement results were returned in a standard template to UKCEH, the central coordinating laboratory for collation and analysis.

<INSERT TABLE 1>

### 2.6 European emissions data

National emissions data: With the exception of Russia and Ukraine, official reported national emissions data on $SO_2$, $NO_x$ and $NH_3$ are available for all other 18 countries in the NEU network from the European Environment Agency (EEA) website (EEA, 2020). Emissions data for the period 2007 to 2010 were extracted and the emission densities of each gas (tonnes (t) $km^{-2}$ $yr^{-1}$) in each country was derived by dividing the 4-year averaged total emissions by the land area ($km^2$).

Gridded emissions data: Gridded emissions data (at 0.1° x 0.1° resolution) for $SO_2$, $NO_x$ and $NH_3$ are available from the EMEP emissions database (EMEP, 2020). The 0.1° x 0.1° gridded data for the period 2007 to 2010 were downloaded and were used to:

1. Estimate national total emissions (sum of all grid squares in each country) and 4-year averaged emission densities (t $km^{-2}$ $yr^{-1}$) for Russia and Ukraine. As a check, total emissions for the other 18 countries were also calculated by this method and were the same as the national emission totals reported by the EEA (EEA, 2019).

2. Extract gas emissions for individual grids (0.1° x 0.1°) that contains a NEU DELTA® site.

3. Extract gas emissions for groups of 4 grids (each = 0.1° x 0.1°) that surrounds a NEU® site and derive grid-averaged emissions.





### 2.7 National air quality network data from the Netherlands and UK

#### 2.7.1 Dutch LML network data

Atmospheric $NH_3$ has been monitored at 8 sites in the Dutch national air quality monitoring network (LML, Landelijk Meetnet Luchtkwaliteitl) since 1993 (van Zanten et al., 2017). The low density, high time-resolution LML network is complemented

by a high density monthly diffusion tube network, the Measuring Ammonia in Nature (MAN) network (http://man.rivm.nl) (Lolkema et al., 2015). The MAN network has 136 monitoring locations sited within nature reserves that includes 60 Natura 2000 sites, with concentrations ranging between 1.0 and 14 µg m$^{-3}$ (Lolkema et al., 2015). The focus of the MAN network is to provide site-based $NH_3$ concentrations for the nature conservation sites, rather than a representative spatial concentration field for the country. Hourly $NH_3$ and $SO_2$ data which were also available from the 8 sites in the LML network were

downloaded from the RIVM website (http://www.lml.rivm.nl/gevalideerd/index.php). The 4-year averaged $NH_3$ and $SO_2$ concentrations for the period 2007 to 2010 were calculated and used to complement measurement data from the 4 Dutch sites in the NEU DELTA® network.

#### 2.7.2 UK NAMN and AGANet network data

Atmospheric $NH_3$, acid gases and aerosols are measured in the UK NAMN (since 1996) and AGANet (since 1999) (Tang et al., 2018a, 2018b). The UK approach is a high density network with low time-resolution (monthly) measurements, combining an implementation of the DELTA® method used in the present NEU DELTA® network and a passive ALPHA® method (Tang et al., 2001) to increase network coverage in $NH_3$ measurements (Sutton et al., 2001b; Tang et al., 2018a). Monthly and annual data for the overlapping period of the project were extracted from the UK-AIR website (https://uk-air.defra.gov.uk/) and nested

with the NEU network data for analysis in this paper.

### 3 Results and Discussion

#### 3.1 Laboratory inter-comparison results: chemical analysis

Figure 4 compares the percentage deviation of results from reference solution concentrations ('true value') reported by the laboratories for different chemical components in the EMEP, WMO-GAW and NEU proficiency testing (PT) schemes,

combined from 2006 to 2010. Each data point is colour-coded in the graphs according to the laboratory providing the measurements.

**\<INSERT FIGURE 4\>**

Altogether, results from the combined PT schemes produced >100 observations for each reported chemical component over the 4 year period. The performances of laboratories in Figure 4 can be summarised in terms of the percentage of reported results agreeing within 10 % of the true values (see summary table below Figure 4), where the true values represent the nominal concentrations in the aqueous test solutions. The best agreements was for $SO_4^{2-}$ and $NO_3^-$, with an average of 92 % and 87 % of all reported results agreeing within 10 % of the true value across the concentration range covered in the PT schemes. In the

case of $NH_4^+$, while an average of 90 % of reported results were within 10% of the reference at 1 mg L$^{-1}$ $NH_4^+$, laboratory performance was poorer (68 % agreeing within 10 %) at lower concentrations (0.1 − 0.9 mg L$^{-1}$). Poorer performance at the low concentrations was largely due to two laboratories (CEAM and SHMU) with > 50 % of their results reading high. For Na$^+$ and Cl$^-$, the percentages of results agreeing within 10 % of the reference were 81 % and 86 %, respectively, across the full range of PT concentrations. At concentrations above 1 mg L$^{-1}$, the agreement improved and increased to 89 % for Na$^+$ and

96% for Cl$^-$. A larger spread around the reference values were provided for the base cations Ca$^{2+}$ and Mg$^{2+}$ at low concentrations





(< 1 mg L$^{-1}$). The percentage of results passing at low concentrations below 1 mg L$^{-1}$ was 36 % (Ca$^{2+}$) and 59 % (Mg$^{2+}$), increasing to 80 % (Ca$^{2+}$) and 90 % (Mg$^{2+}$) above 1 mg L$^{-1}$. The larger scatter at low concentrations is likely due to uncertainty in the chemical analysis at or close to the method limit of detection, and reflects challenges of measuring base cations, in particular Ca$^{2+}$ as this is very 'sticky' and adsorbs/desorbs from surfaces leading to analytical artefacts.

To show what the PT reference solution concentrations would correspond to if they were a denuder and/or aerosol extract, equivalent gas (Equation 1) and/or aerosol concentrations (Equation 2) (Sect. 2.2.1) are calculated for each of the ions and provided in the summary table in Figure 4. A 0.5 mg L$^{-1}$ NH$_4^+$ solution, for example, is equivalent to an atmospheric concentration of 0.09 µg NH$_3$ m$^{-3}$ (gas), or 0.13 µg NH$_4^+$ m$^{-3}$ (aerosol) for a monthly sample. In Figure 5, scatter plots are

shown comparing all NEU laboratory reported results with PT reference, where all ion concentrations (mg L$^{-1}$) from Figure 4 have been converted to equivalent gas and aerosol concentrations (µg m$^{-3}$), based on a typical volume of 15 m$^3$ over a month. With the exception of a small number of outliers, most data points are close to the 1:1 line with laboratory results agreeing within ± 0.05 µg m$^{-3}$ in equivalent gas and/or aerosol concentrations. These are low ambient concentrations and show that the measurement uncertainty in the analysis of very low concentrations in the PT schemes will be small for the majority of sites

in the network, where concentrations were found to be much higher (see Figure 6).

**<INSERT FIGURE 5>**

### 3.2    Laboratory inter-comparison results: DELTA® measurements

Results from 4 years of annual DELTA® field inter-comparisons (2006 – 2009), for all field sites, are combined and summarised in Figure 6. The gas and aerosol concentrations measured and reported by each of the laboratories are compared with the median estimate of all laboratories in each of the scatter plots, with the colour of the symbols identifying the laboratory providing the measurements. Regression results (slope and R$^2$) in the table below the plots provide the main features of the inter-comparison. The slope is equivalent to the mean ratio of each laboratory against the median value, where values close to

unity indicate closer agreement to the median value. Overall, the scatter plots show good agreement between the laboratories, with some laboratories showing very close agreement to the median estimates, and more scatter observed from the others.

**<INSERT FIGURE 6>**

The occurrence of outliers in some of the individual monthly values indicates that caution needs to be exercised in the interpretation of these data points in the inter-comparison. To average out the influence of a few individual outliers, the mean concentrations from each of the seven laboratories for each of the four field sites were calculated and compared with averaged median estimates of all laboratories for each site. A summary of the mean concentrations and the percentage difference from median is presented in Table 2. Since the INRAE laboratory did not join the NEU network until 2008, averaged median values

from the 2008 and 2009 inter-comparisons are used to compare with the INRAE results, included in the table for clarity. The mean concentrations between laboratories are broadly comparable. Each of the laboratories were also able to resolve the main differences in mean concentrations at the four field sites, ranging from the smallest concentrations at Auchencorth (e.g. median = 1.4 µg NH$_3$ m$^{-3}$) to higher concentrations representing a more polluted site at Paterna (e.g. median = 5.2 µg NH$_3$ m$^{-3}$) for the test periods (Table 2). Larger differences for HCl, Ca$^{2+}$ and Mg$^{2+}$ are due to clear outliers from one or two laboratories at the

very low concentrations of these species encountered and may be related to measurement uncertainties at the low air concentrations. The comparability between laboratories for each of the components is next considered in turn.





<INSERT TABLE 2>

### 3.2.1 Inter-comparisons: NH₃, NH₄⁺, HNO₃, NO₃⁻

The best agreement between laboratories was for the $N_r$ gases ($NH_3$, $HNO_3$) and aerosol species ($NH_4^+$, $NO_3^-$), with slopes within ± 10 % of the median values and $R^2 > 0.9$ in the regression analysis from five of the laboratories (Figure 6, Table 2). This is important since $N_r$ species were the primary focus for the NEU DELTA® network. Slightly poorer agreement for $NH_3$ and $NH_4^+$ were provided by CEAM and MHSC laboratories, with data points both above and below the 1:1 line (Figure 6). The outliers above the 1:1 line from MHSC were from the 2006 inter-comparison exercise. Removal of these 2006 outliers improved the MHSC regression slope for $NH_3$ from 1.21 ($R^2 = 0.87$, $n = 41$) to 0.99 ($R^2 = 0.99$, $n = 10$) (Supp. Figure S1). While this seems to suggest that the performance of MHSC for $NH_3$ improved following the first inter-comparison exercise, the regression slope for aerosol $NH_4^+$ increased instead from a slope of 1.26 ($R^2 = 0.83$, $n = 41$) to 1.48 ($R^2 = 0.93$, $n = 10$), suggesting an over-estimation of $NH_4^+$ concentrations (Supp. Figure S1). A possible cause may be the quality and/or variability in the aerosol filter blank values for $NH_4^+$, as laboratory blanks are subtracted from exposed samples to estimate aerosol $NH_4^+$ concentrations. Laboratory blank results were however not reported to allow this assessment. Another possibility is a breakthrough of $NH_3$ from the acid coated denuders onto the aerosol filters. The denuder collection efficiency of $NH_3$ gas (Equation 3, Sect. 2.2.1) reported by MHSC was on average 88 % for all years and 91 % where 2006 data have been excluded (Supp. Table S3). This is comparable with the mean collection efficiencies of all laboratories (91 and 90 %) (Supp. Table S3), which makes $NH_3$ breakthrough an unlikely explanation for the higher readings. The assessment of $NH_4^+$ is however more uncertain from the reduced number of data points ($n = 10$).

For the CEAM laboratory, reported $NH_3$ concentrations were on average 16 % lower ($n = 41$) than the median, with a slope of 0.89 ($R^2 = 0.87$) and particulate $NH_4^+$ were on average 13 % lower ($n = 41$) than the median, with a slope of 0.42 ($R^2 = 0.22$) (Figure 6). A need to improve the $NH_4^+$ analysis (Indophenol colorimetric assay) in the acid coated denuders and aerosol filters by the CEAM laboratory was identified from the 2006 inter-comparison (Tang et al., 2009). The Indophenol method for aqueous $NH_4^+$ determination is pH sensitive. Calibration solutions and quality control checks for the colorimetric assays are made up in deionised water (pH 7), whereas the aqueous extracts from the DELTA® acid coated denuders and cellulose filters are acidic (pH ~3). Determination of $NH_4^+$ in the denuder extracts may therefore be under-estimated if the pH of the indophenol reaction has not been adjusted for the increased acidity in the sample extracts. When the 2006 data are excluded from the regression analysis, the slopes for $NH_3$ and $NH_4^+$ increased to 1.02 ($R^2 = 0.94$, $n = 12$) and 0.98 ($R^2 = 0.51$, $n = 12$), respectively (Supp. Figure S1). The improved agreement with other laboratories after the 2006 inter-comparison suggests that the method under-read was largely resolved, reflected in an improvement in the slope. Despite some uncertainties in the $NH_3/NH_4^+$ measurements, the laboratories were able to clearly resolve the main differences in mean concentrations at the four different field sites in all years (Table 2). The results presented here for CEAM and MHSC highlight the importance of the initial inter-comparison exercise in identifying and resolving sampling and analytical issues at the start of the project.

### 3.2.2 Inter-comparisons: SO₂, SO₄²⁻

Six laboratories provided slopes within 12 % of the median values in the regression analysis for $SO_2$ (Figure 6). The smaller $R^2$ values were from two laboratories (CEAM and SHMU, $R^2 < 0.7$), with data points both above and below the 1:1 line. For INRAE, the larger slope of 1.6 ($R^2 = 9$) was due to a single high $SO_2$ reading reported for Auchencorth of 2.0 µg $SO_2$ m⁻³, compared with the median of 1.4 µg $SO_2$ m⁻³. When the mean $SO_2$ concentrations measured by INRAE are compared with the median, the difference was on average 13 %, providing acceptable agreement, which suggests that the high reading may just be an outlier. There was more scatter in the inter-comparison for $SO_4^{2-}$, although the majority of points are still close to the 1:1





line (Figure 6). Six laboratories provided slopes within 12 % of the median values in the regression analysis also for $SO_4^{2-}$. The regression slope from CEAM for $SO_4^{2-}$ was 1.2 ($R^2 = 0.9$) which is still within 20% of the median. The $SO_2$ and $SO_4^{2-}$ measurements were broadly comparable between the laboratories, with mean concentrations agreeing on average within 6 % of the median (Table 2).

**3.2.3    Inter-comparisons: HCl, Cl$^-$**

The HCl inter-comparison show clear outliers from the CEAM laboratory, with concentrations that were on average up to 2 times higher than other laboratories (slope = 1.8). For example, a mean concentration of 1.8 µg HCl m$^{-3}$ was reported by CEAM for Paterna, compared with a median of 0.7 µg HCl m$^{-3}$. Apart from CEAM, the mean concentrations of HCl reported by the other laboratories were generally comparable (Table 2). The larger % differences between the measured mean and
median at each site reflect the challenges of measuring the very low concentrations of HCl at these sites of < 0.5 µg HCl m$^{-3}$ (slightly higher at Paterna). HCl results were reported by NILU for the 2008 inter-comparison exercise only, limiting the number of measurements ($n = 4$) available for comparison.

The comparison for Cl$^-$ showed better agreement of the CEAM laboratory results with other laboratories, in both the inter-
comparison of individual monthly values (Figure 6) and the mean concentrations (Table 2). Like HCl, larger % differences between the measured concentrations and median at each site may be attributed to higher measurement uncertainties at the low concentrations of Cl$^-$. For NILU, there were only 2 data points for Cl$^-$ from the Auchencorth site in the 2008 inter-comparison. Overall, the inter-comparison for HCl and Cl$^-$ showed that the laboratories were able to resolve the main differences in mean concentrations at the different sites even at the low concentrations encountered.

**3.2.4    Inter-comparisons: Base cations (Na$^+$, Ca$^{2+}$, Mg$^{2+}$)**

Measurements of Ca$^{2+}$ and Mg$^{2+}$ were the most uncertain, with the largest scatter in the inter-comparisons (Figure 6). Despite the trace levels of these base cations at all field sites, 4 laboratories (INRAE, UKCEH, SHMU, VTI) provided data close to the 1:1 line, demonstrating close agreement between these laboratories. The clear outliers above the 1:1 line are from CEAM,
MHSC and NILU, with slopes > 2. While MHSC over-read Ca$^{2+}$ and Mg$^{2+}$, their results for Na$^+$ were in better agreement with other laboratories, with a slope of 0.9 ($R^2 = 0.5$) (Figure 6). There was a lot of scatter in the data however, with outlier points both above and below the 1:1 line, suggesting measurement uncertainties in their base cation measurements. For NILU, the only base cation results reported by the laboratory were for the 2008 DELTA® inter-comparisons at Auchencorth and Braunschweig. This accounts for the low number of data points ($n = 4$) from the NILU laboratory. The median concentrations
of Ca$^{2+}$ and Mg$^{2+}$ at both field sites were very low (< 0.1 µg m$^{-3}$), which makes comparison with the few data reported from NILU highly uncertain. Like NILU, CEAM also did not report base cations results for all of the DELTA® inter-comparison. Base cation results provided by CEAM were for 2007 – 2009 only.

**3.3    Variation in annual mean gas and aerosol concentrations and composition**

**3.3.1    Comparisons according to ecosystem types**

Annual averaged concentrations of gases and aerosols measured in the NEU DELTA® network are presented in Figure 7, with sites grouped according to each of four major ecosystem types: crops, grassland, forests and semi-natural. These are the classifications used in dry deposition models, where ecosystem-specific deposition velocities ($V_d$) are combined with
measurement data to produce estimates of N$_r$ dry deposition (Flechard et al., 2011). In some models such as the Concentration





Based Estimates of Deposition (CBED) model (Smith et al., 2000; Flechard et al., 2011), a canopy compensation point and the bi-directional exchange of $NH_3$ between vegetation-type and the atmosphere are also considered (e.g. Sutton et al., 1995; Massad et al., 2010; Flechard et al., 2011).

**<INSERT FIGURE 7>**

A total of 64 sites from 20 different countries, including replicated measurements at 4 of the sites, are compared in Figure 7. Not all of the sites were however operational all of the time or at the same time. Changes in the numbers and locations of sites occurred over the duration of the network, for example, due to site closures, relocations and/or new site additions. The annual
averaged concentrations plotted for each site are the mean of all available annual means. Where the annual averaged concentration is derived from less than 4 full years of data, the number of years providing the mean is shown, in brackets, next to the site data in the graph. To avoid bias in the calculation of annual means, due to seasonality in the data (see later in Sect. 3.5), years with incomplete data coverage (< 7 months of data in any year) were excluded. Applying these data exclusions, the number of sites that provided annual data was 55 sites for 2007, 57 sites for 2008, 54 sites for 2009 and 55 sites for 2010. The
number of sites that provided annual data for each year over the entire period was 45 sites.

Sites with parallel (P) DELTA® measurements were Auchencorth Moss (UK-AMoP), Easter Bush (UK-EBuP), Fougéres (FR-FgsP) and SK04P (EMEP site in Slovakia) (Figure 7). Overall, good reproducibility in DELTA® measurements was demonstrated by the parallel measurements. At the Auchencorth Moss parallel site (UK-AMoP), $NH_3$ and $NH_4^+$ only were
measured, and agreement for these 2 components were on average within 5 % at the low concentrations measured at this site (annual mean: $0.5 - 0.9$ µg $NH_3$ $m^{-3}$ and $0.3 - 0.5$ µg $NH_4^+$ $m^{-3}$). Parallel measurements at Easter Bush (UK-EBuP) stopped in March 2010. With the exception of $Ca^{2+}$ and $Mg^{2+}$, the comparison of annual mean data from the replicated measurements for 2007 to 2009 provided excellent agreement of 2 % ($Na^+$) to 13 % ($SO_4^{2-}$) at Easter Bush. At Fougéres, $HNO_3$ concentration measured on $K_2CO_3$/Glycerol coated denuders (FR-Fgs) was about 2-fold higher than on NaCl coated denuders in the parallel
DELTA® system (FR-FgsP), consistent with over-estimation of $HNO_3$ (on average 45 %) on carbonate coated denuders (see Sect. 2.2.3). The disadvantage of a NaCl coating, however, is that it can only collect $HNO_3$ and not the other acid gases. A third carbonate denuder is necessary in the sample train to collect and measure $SO_2$, since $SO_2$ is only partially captured and HCl cannot be measured on NaCl denuders (Tang et al., 2015, 2018b). This explains the smaller $SO_2$ concentrations reported by the FR-FgsP site, with break-through of $SO_2$ (inefficiently captured by NaCl denuders) onto the aerosol filters resulting in
larger particulate $SO_4^{2-}$ concentrations than the Fr-Fgs site. For the SK04 site, measurement reproducibility for the 4 years of parallel data for N and S component was good, with agreement ranging from 0.4 % ($NH_4^+$) to 15 % ($SO_4^{2-}$). HCl and $Na^+$ and determinations were however more uncertain with differences of 21 and 28%, respectively. It has to be noted, however, that the concentrations of the two components were very low, at < 0.2 µg HCl $m^{-3}$ and < 0.4 µg $Na^+$ $m^{-3}$. The differences in concentrations are therefore actually within ±0.1 µg $m^{-3}$ for HCl and within ±0.2 µg $m^{-3}$ for $Na^+$.

A key feature in Figure 7 is the dominance of N over S species at most sites, when expressed as µg $m^{-3}$ of the element. The mean percentage contribution of sum $N_r$ ($NH_3$-N, $HNO_3$-N, $NH_4^+$-N, $NO_3^-$-N) concentrations to the total mass of gas and aerosol species measured is 52 % (range = $24 - 80$%), twice as much as from sum S ($SO_2$-S and $SO_4^{2-}$-S; mean = 23 %, range = $7 - 53$ %) (Figure 8). This is consistent with more substantial reductions in $SO_2$ emissions (−72%) than achieved with $NO_x$
(−43%) or $NH_3$ (−18%) in Europe between $1991 - 2010$ (EEA, 2019). The differences in atmospheric composition of S and N species in the present assessment therefore reflected changes in emissions of the precursor gases, and are also in agreement with a recent assessment of air quality trends showing important changes in S and N composition in air and rain across the EMEP networks (EMEP, 2016).





<INSERT FIGURE 8>

Most of the $N_r$ concentrations at each site in turn are dominated by reduced N ($NH_3$-N, $NH_4^+$-N), rather than by oxidised N species ($HNO_3$-N, $NO_3^-$-N). Of the sum $N_r$ concentrations measured, $60-97$ % (mean = 76%, $n = 66$) were reduced N ($N_{red}$)

(Figure 8). Even more strikingly, $NH_3$ ($NH_3$-N) was by far the single most dominant component at the majority of sites, contributing on average 42% (range = $24-56$ %, $n = 10$) at cropland sites and 20 % ($6-46$%, $n = 35$) of the total gas/aerosol concentrations at forest sites (Figure 8). This illustrates very clearly the importance of $NH_3$ and by association agricultural emissions in contributing to $NH_3$-N concentrations and deposition in Europe, with 92 % of total $NH_3$ emissions in Europe estimated to come from agriculture (EEA, 2019). The reaction of $NH_3$ with the acid gases $HNO_3$ and $SO_2$ forms $NH_4^+$-

containing particulate matter (PM) that are primarily $NH_4NO_3$ and $(NH_4)_2SO_4$ (Figure 1) (see Sect. 3.4). Together, particulate $NH_4^+$-N, $NO_3^-$-N and $SO_4^{2-}$-S made up on average 28% ($17-40$ %, $n = 10$) of the total gas/aerosol concentrations measured at cropland sites (Figure 8). At semi-natural and forest sites however, that number was even bigger at 33% ($20-40$%, $n = 11$) and 37 % ($24-57$%, $n = 35$), respectively (Figure 8).

Secondary $NH_4^+$ particles are mainly in the 'fine' mode with diameters of less than 2.5 µm ($PM_{2.5}$) and estimated to contribute between 10 to 50 % of ambient $PM_{2.5}$ mass concentration in some parts of Europe (Putaud et al., 2010, Schwartz et al., 2016). An assessment by Hendriks et al. (2013) found that secondary $NH_4^+$ contributed $10-20$% of the $PM_{2.5}$ mass in densely populated areas in Europe and even higher contributions in areas with intensive livestock farming. Concentrations of $PM_{25}$ continue to exceed the EU limit values of 25 µg m$^{-3}$ annual mean in large parts of Europe in 2017 (EEA, 2019). Particulate

$NH_4^+$ data presented from the DELTA® network therefore highlights the potential contribution of $NH_3$ of agricultural origin to fine $NH_4^+$ aerosols in $PM_{2.5}$. The formation and transport of these secondary aerosols poses a serious risk to human health, since $PM_{2.5}$ are linked with increased mortality from respiratory and cardiopulmonary diseases (AQEG, 2012).

A considerable fraction of the aerosol components measured was made up of sea salt ($Na^+$ and $Cl^-$), with contributions from

sum($Na^+$ and $Cl^-$) ranging from 4 % of the total aerosol loading at the inland Höglwald site in Germany (DE-Hog) to 43 % at Dripsey (IE-Dri), a coastal site in Ireland (Figure 7). With the reduction in European emissions and concentrations of the gases $SO_2$, $NO_x$ and $NH_3$ for formation of $NH_4^+$-containing aerosols, sea salt is therefore assuming a proportionate increase of the aerosol composition, consistent with observations from a recent European assessment of composition and trends in long-term EMEP measurements (EMEP, 2016). The concentrations of $Ca^{2+}$ and $Mg^{2+}$ were very low across the network, with values

(mean of all sites = < 0.1 µg m$^{-3}$) that were at or below method limit of detection (LOD = ~ 0.1 µg m$^{-3}$). These data are also considered to be under-estimated due to the DELTA particle sampling cut-off (~ $PM_{4.5}$) and they were excluded from further assessment in this paper.

### 3.3.2 Comparisons with national gas emissions

In Figure 9, the annual averaged gas and aerosol concentrations of grouped sites from each country are plotted with the corresponding national emission densities derived for $NH_3$, $NO_x$ and $SO_2$. The emissions data in the graphs are the 4-year averages for the period 2007 to 2010, expressed as emissions per unit area of the country per year (t km$^{-2}$ yr$^{-1}$) (see Sect. 2.6) and ranked in order of increasing emission densities. The error bars, where shown, is the range (min and max) of annual averaged concentrations of sites in each country. Where error bars are not visible, this indicates either that the country has

measurement from just one site, or the range of concentrations measured are very close to the average. From the visual comparisons, national mean measured concentrations in each country appear to scale reasonably well with the ranked emission densities. This is supported by further regression analyses which showed significant correlation between annual averaged





concentrations of $NH_3$, $NO_x$ and $SO_2$ with emission densities of $NH_3$ ($R^2 = 0.49$, $p < 0.001$, Figure 10A1), $NO_x$ ($R^2 = 0.20$, $p < 0.05$, Figure 10A2) and $SO_2$ ($R^2 = 0.65$, $p < 0.001$, Figure 10A3), respectively (Table 3). The particulate components $NH_4^+$ and $NO_3^-$ were also correlated with both precursor gases $NH_3$ and $HNO_3$ (Table 3). By contrast, there was no relationship between $SO_4^{2-}$ with any of the three gases, possibly because of contributions to $SO_4^{2-}$ from long-range transport. All regression
plots of concentrations against emission densities, including summary statistics are provided in Supp. Figure S2.

**<INSERT FIGURE 9>**
**<INSERT FIGURE 10>**
**<INSERT TABLE 3>**

### 3.3.3    Comparisons with gridded emissions

The comparisons in Sect. 3.3.2 used national emission totals, where emissions have been summed and averaged across very large and heterogeneous areas in each country. Another approach is to compare the individual site mean data with gridded emissions from individual $0.1° \times 0.1°$ EMEP grids in which the NEU sites are located (see Sect. 2.6). This also provided
significant correlations for $NH_3$ ($p < 0.001$, $n = 66$, Figure 10B1) and $HNO_3$ vs $NO_x$ ($p < 0.05$, Figure 10B2), but not for $SO_2$ (Figure 10B3, Supp. Figure S3). Some interesting features also emerged in the $NH_3$ comparisons, with clustering of data according to ecosystem types (Figure 10B1). The cropland sites have highest $NH_3$ concentrations compared with gridded emissions (slope = 0.03, $R^2 = 0.34$, $p = 0.08$, $n = 10$), followed by grassland sites (slope = 0.01, $R^2 = 0.87$, $p < 0.001$, $n = 10$) (Fig. 10B1, Supp. Figure S3). Forest (slope = 0.007, $R^2 = 0.87$, $p < 0.001$, $n = 35$) and semi-natural sites (slope = 0.004, $R^2 = $
0.25, $p = 0.11$, $n = 11$) are similar, with smaller $NH_3$ concentrations compared with their gridded emissions. Since $NH_3$ is spatially heterogeneous even at a local sub-grid scale (e.g. Dragosits et al., 2002), the smaller concentrations at semi-natural and forest sites in grids with large emissions indicates these sites may be located further away from sources in the grid (Tang et al., 2018a; van Zanten et al., 2017). Dry deposition of $NH_3$ is also largest to forests and semi-natural areas (larger $V_d$ than to crops/grass ecosystem types, e.g. Smith et al., 2000; Flechard et al., 2011), which could also contribute to the smaller
concentrations at higher emissions. Relationship between emissions and concentrations in the atmosphere is however complex, influenced by other factors such as chemical interactions, variations in meteorological conditions and long-range transboundary import.

The lack of correlation between $SO_2$ concentrations and gridded emissions (Figure 10B3) suggests that a $0.1° \times 0.1°$ grid may
be too local a spatial scale for an emission-concentration comparison for $SO_2$, as $SO_2$ is likely to be highly localised with emissions occurring from a smaller number of large point sources at an elevated height. Indeed, emissions in neighbouring grids surrounding each site are highly variable. For example, the 4-year averaged $SO_2$ emissions in the 4 EMEP grids around the Italian San Rossore site (IT-SRo) varied between 0.47 to 610 kt $SO_2$ yr$^{-1}$. Further analysis was also carried out comparing site mean concentrations against the averaged emissions of an extended number of EMEP grids ($4 \times$ grids) (Supp. Figure S4).
Since the analysis provided similar results to the comparisons with individual gridded emissions, they are not included for further discussions in this paper. All regression plots and summary statistics for both comparisons (gridded emissions from single grids or from average of 4 grids) are provided in Supp. Figures S3 and S4.

### 3.3.4    Spatial variability across geographical regions

The form and concentrations of the different gas and aerosol components measured also varied according to geographic regions across Europe (Figure 11). Smallest concentrations (with the exception of $SO_4^{2-}$ and $Na^+$) were in Northern Europe

 

(Scandinavia), with broad elevations across other regions. Gas-phase NH$_3$ and particulate NH$_4^+$ were the dominant species in all regions (Figure 11). NH$_3$ showed the widest range of concentrations, with largest concentrations in Western Europe (mean = 2.4 NH$_3$ m$^{-3}$, range = 0.2 – 7.1 µg NH$_3$ m$^{-3}$, $n$ = 26 in 4 countries). By contrast, HNO$_3$ and SO$_2$ concentrations were largest in high NO$_x$ and SO$_2$ emitting countries in Central and Eastern Europe (Sect. 3.3.3). Particulate SO$_4^{2-}$ concentrations were

however more homogeneous between regions, which may be attributed to atmospheric dispersion and long-range transboundary transport of this stable aerosol between countries in Europe (Szigeti et al., 2015; Schwarz et al., 2016). In the aerosol components, the spatial correlations between NO$_3^-$, NH$_4^+$ and NH$_3$ illustrates the potential for NH$_3$ emissions to drive the formation and thus regional variations in NH$_4^+$ and NO$_3^-$ aerosol. Particulate SO$_4^{2-}$ concentrations in Northern Europe (Scandinavia) were similar to other countries, despite having the smallest SO$_2$ and NH$_3$ emissions and concentrations (Figure

9). By comparison, the smaller particulate NH$_4^+$ and NO$_3^-$ concentrations in Northern Europe are consistent with smallest emissions (NH$_3$ and NO$_x$) and concentrations of NH$_3$ and HNO$_3$ (Figure 9). As discussed later in Sect. 3.4, the larger SO$_4^{2-}$ concentrations reported in Northern Europe were flagged up as anomalous from ion balance checks (ratio of NH$_4^+$:sum anions).

< INSERT FIGURE 11>

### 3.3.5    Comparisons by grouped components

In the following sections, variations in concentrations of the different gas and aerosol components according to ecosystem types (crops, grassland, forests and semi-natural) and in relation to emissions (NH$_3$, NO$_x$ and SO$_2$) are further discussed. For ease of interpretation, components are grouped as follows: reduced N (NH$_3$, NH$_4^+$), oxidised N (HNO$_3$, NO$_3^-$), S (SO$_2$, SO$_4^{2-}$),

HCl, Na$^+$ and Cl$^-$.

### Reduced N (NH$_3$ and NH$_4^+$)

Broad differences in NH$_3$ concentrations are observed between the grouped sites, with the largest concentrations at cropland sites, as expected, as these are intensively managed agricultural areas dominated by NH$_3$ emissions (Figure 7A). Borgo Cioffi

(IT-BCi) in an intensive buffalo farming region of Southern Italy provided the highest 4-year average of 8.1 µg NH$_3$-N m$^{-3}$ (*cf.* group mean = 3.8 µg NH$_3$-N m$^{-3}$, $n$ = 10) (Table 4, Supp. Table S4). Next highest in this group are the German Gebesee (DE-Geb) and the Belgian Lonzee (BE-Lon) sites with 4-year average concentrations of 4.9 and 4.8 µg NH$_3$-N m$^{-3}$, respectively (Supp. Table S4). At Gebesee, a decrease in NH$_3$ concentrations was observed over the 4 year period, falling almost 2-fold from an annual mean of 8.8 µg NH$_3$-N in 2007 to 4.8 µg NH$_3$-N in 2010 (Supp. Table S4). Annual mean concentrations in

2008 (2.9 µg NH$_3$-N m$^{-3}$) and 2009 (3.2 µg NH$_3$-N m$^{-3}$) were similar, but smaller than in 2010. This illustrates the large inter-annual variability in concentrations that can occur even over a short time period. Variability between years may reflect changes in meteorological conditions on emissions from potential sources, with for example warmer, drier years increasing emissions and concentrations, contrasting with lower emissions and concentrations from the same source in a colder and wetter year. Episodic pollution events can also have a large influence on the annual mean concentration, rather than the direct effects of

changes in anthropogenic emissions over this short time scale. This suggests that for compliance assessment, an average over several years would provide a more robust basis than individual years. The assessment of trends also needs a longer time series of at least 10 years (Tang et al., 2018a, 2018b; Torseth et al., 2012; van Zanten et al., 2017).

< INSERT TABLE 4>

Grassland sites, with NH$_3$ emissions from grazing and fertilisers, provided the next highest concentrations, with annual averaged concentrations of 2.2 µg NH$_3$-N m$^{-3}$ from the 10 sites in this group (Table 4). Cabauw in the Netherlands (NL-Cab)



in this group was the second highest $NH_3$ concentration site in the DELTA® network, after Borgo Cioffi (IT-BCi), with a 4-year annual averaged concentration of 5.9 µg $NH_3$-N m$^{-3}$ (Supp. Table S4). Unlike the Gebesee site (DE-Geb), annual $NH_3$ concentrations were consistent between years at Cabauw, ranging from annual mean of 6.3 µg $NH_3$-N m$^{-3}$ in 2017 to 5.8 µg $NH_3$-N m$^{-3}$ in 2010 (Supp. Table S4).

At the clean end of the $NH_3$ gradient are semi-natural and forest sites. The smallest concentrations were found at remote background sites in Russia (Fyodorovskoe bog, RU-Fyo) and the Scandinavian countries, in Finland (Lompolojänkkä FI-Lom, Hyytiälä FI-Hyy, Sodankylä FI-Sod), Norway (Birkenes, NO-Bir) and Sweden (Norunda SE-Nor, Skyytopr SE-Sky), where $NH_3$ concentration at each site was < 0.3 $NH_3$-N m$^{-3}$ (Figure 7, Supp. Table S4). By contrast, the semi-natural Horstermeer (NL-Hor) and forest sites Speulder (NL-Spe) and Loobos (NL-Loo) in the Netherlands gave concentrations that were ten-fold higher (2.9 - 4.1 µg $NH_3$-N m$^{-3}$) (Figure 7, Supp. Table S4). This is consistent with much higher $NH_3$ emission density in the Netherlands (4-year average = 3.4 kt $NH_3$-N km$^{-2}$ yr$^{-1}$) (Figure 9).

With the exception of the Czech Republic, the annual averaged $NH_3$ concentrations scaled reasonably well with the 4-year averaged mean $NH_3$ emission density in each country (Figures 9, 10A1, 10B1) (see also Sect. 3.3.2 and Sect. 3.3.3). In the Czech Republic, measurement was made at a single site, BKFores (CZ-BK1), located at a remote forest location. The 4-year averaged emissions in the EMEP grid (1° x 1°) containing the site is very small, at 2 t $NH_3$-N yr$^{-1}$, compared with an average of 68 t $NH_3$-N yr$^{-1}$ (range = < 0.01 to 567 t $NH_3$-N yr$^{-1}$) across the Czech Republic. The low emissions, combined with the small concentrations measured at BKFores (0.5 µg $NH_3$-N m$^{-3}$), suggests it is highly likely to represent concentrations at the low end of the range of $NH_3$ concentrations that might be expected to be encountered in the Czech Republic. By comparison, Belgium has a similar emission density as the Czech Republic, but the mean concentrations from 3 sites (2.6 µg $NH_3$-N m$^{-3}$) encompassed sites located in cropland areas (Lonzee BE-Lon, 4.7 µg $NH_3$-N m$^{-3}$) and forest sites (Braschaat BE-Bra, 2.8 µg $NH_3$-N m$^{-3}$, and Vielsalm BE-Vie, 0.4 µg $NH_3$-N m$^{-3}$) (Supp. Table S4).

The markedly high concentrations of $NH_3$ across the NEU network indicates that contributions by emission and deposition of $NH_3$ would be a major contributor to the effects of $N_r$ on sensitive habitats. In comparing the annual averaged $NH_3$ concentration with the revised UNECE 'Critical Levels' of $NH_3$ concentrations (Cape et al., 2009), the lower limit of 1 µg $NH_3$ m$^{-3}$ annual mean for the protection of lichens-bryophytes were exceeded in 63 % of sites (40 sites in 15 countries) (Supp. Table S5). Even the higher 3 µg $NH_3$ m$^{-3}$ annual mean for the protection of vegetation was still exceeded at 27 % of sites (17 sites in 10 countries) (Supp. Table S5). Most notably, all 4 sites from the Netherlands were in exceedance of both the 1 and the 3 µg $NH_3$ m$^{-3}$ thresholds. The large concentrations in the Netherlands highlights the high levels of $NH_3$ that semi-natural and forest areas are exposed to within an intensive agricultural landscape, where 117 out of the 166 Natura2000 areas were reported to be sensitive to nitrogen input (Lolkema et al., 2015). A recent assessment estimated that critical loads for eutrophication were exceeded in virtually all European countries and over about 62 % of the European ecosystem area in 2016 (EMEP, 2018). In particular, the highest exceedances occurred in the Po Valley (Italy), the Dutch-German-Danish border areas and north-western Spain where the highest $NH_3$ concentrations have been measured in this network. Since $NH_3$ is preferentially deposited to semi-natural and forests (high $V_d$ to these ecosystem types, Sutton et al., 1995), then $NH_3$ will dominate dry $NH_3$-N dry deposition and exert the larger ecological impact. In Flechard et al. (2011), dry $NH_3$-N deposition from the first 2 years of $NH_3$ measurement in the NEU DELTA® network was estimated to contribute between 25 and 50% of total dry N deposition in forests, according to models. The fraction is larger in short semi-natural vegetation, since $V_d$ for $NH_4^+$ and $NO_3^-$ is smaller in short vegetation than to forests (Flechard et al., 2011).





**Comparison with NH₃ data from the Dutch LML network**

The 4-year averaged NH₃ concentrations from the Dutch LML air quality network (see Sect. 2.7.1) for the period 2007 to 2010 are plotted alongside the NH₃ measurements made at the 4 Dutch sites in the DELTA® network (Figure 9A). The 4-year averaged concentrations from the 8 LML sites were between 1.5 to 15 µg NH₃-N m⁻³, highlighting the high concentrations and

spatial variability in concentrations in the Netherlands. The mean NH₃ concentrations measured at the 4 Dutch sites in the DELTA® network of 2.9 µg NH₃-N m⁻³ (Horstermeer, NL-Hors; semi-natural) to 5.9 µg NH₃-N m⁻³ (Cabauw, NL-Cab; grassland) were within the range of concentrations measured in the Dutch LML network.

**Comparison with NH₃ data from the UK NAMN network**

The 4-year averaged NH₃ concentrations calculated from the 72 sites in the NAMN (see Sect. 2.7.2) for the period 2007 to 2010 were smaller than the Dutch LML network, ranging from 0.05 to 6.7 µg NH₃-N m⁻³ that are consistent with smaller NH₃ emission from the UK (Figure 9A). In a joint collaboration between the UK and Dutch networks, inter-comparison of NH₃ measurements by the DELTA® method (monthly) with the Dutch network AMOR wet chemistry system (hourly, van Zanten et al., 2017) were carried out at the Zegweld site (ID 633) in the Dutch LML network (van Zanten et al., 2017) between 2003

and 2015. Good agreement was provided lending support for comparability between the independent measurements, reported in Tang et al. (2018a).

**Particulate NH₄⁺**

Particulate NH₄⁺ concentrations across the 64 sites were more homogeneous than NH₃, varying over a narrower range between

0.13 µg NH₄⁺-N m⁻³ at Sodankylä (Finland, FI-Sod) and 2.1 µg NH₄⁺-N m⁻³ at Borgo Cioffi (Italy, IT-BCi) (Figure 7, Supp. Table S6). By comparison, the difference in NH₃ between the smallest (0.07 µg NH₃-N m⁻³ at Lompolojänkkä, Finland, FI-Lom) and largest (8.1 µg NH₃-N m⁻³ at Borgo Cioffi, Italy, IT-BCi) concentrations varied by a factor of 110 (Figure 7, Supp. Table S4). Secondary aerosols have longer atmospheric lifetimes and will therefore vary spatially much less than their precursor gas concentrations. While the concentrations of NH₃ vary at a local to regional level owing to large numbers of

sources at ground level, and high deposition in the landscape, NH₄⁺ is less influenced by proximity to NH₃ emission sources and varies in concentration at regional scales (Sutton et al., 1998; Tang et al., 2018a).

In Figure 9, annual averaged NH₄⁺ concentrations (µg NH₄⁺-N, Figure 9E; nmol m⁻³ in Figure 9G) are plotted with 4-year averaged emissions densities for NH₃, NOₓ and SO₂ from each country, with the combined total emission densities shown in

ranked order. Regression analyses showed NH₄⁺ concentrations to be correlated with NH₃ emissions ($R^2 = 0.36$, $p < 0.01$, $n = 20$) and NOₓ emissions ($R^2 = 0.27$, $p = 0.02$, $n = 20$), but not with SO₂ emissions (Table 3, Supp. Figure S2). The smallest NH₄⁺ concentrations were in Sweden, Norway and Finland (annual average < 0.3 µg NH₄⁺-N m⁻³) with the lowest emissions of NH₃, NOₓ and SO₂ and also the smallest concentrations of the precursors gases NH₃ (< 0.3 µg NH₃-N m⁻³), HNO₃ (< 0.1 µg HNO₃-N m⁻³) and SO₂ (< 0.3 µg SO₂-S m⁻³).

The UK and Irish sites have the next smallest NH₄⁺ concentrations of 0.4 and 0.5 µg NH₄⁺-N m⁻³ (*cf.* mean of all countries = 0.74 µg NH₄⁺-N m⁻³). Particulate NH₄⁺ data from the UK NAMN (Tang et al., 2018a) are also included for comparison. The 4-year average concentrations from the 30 sites (0.5 µg NH₄⁺-N m⁻³, range = 0.14 to 1.0 µg NH₄⁺-N m⁻³) are comparable with the mean of 0.40 µg NH₄⁺-N m⁻³ (range = 0.2 to 0.9 µg NH₄⁺-N m⁻³) from just 4 sites in the NEU network. A combination of

lower emissions of precursor gases (Figure 9) and being further away from the influence of long-range transport of NH₄⁺ aerosols from the higher emission countries on mainland Europe may be contributing factors to the small NH₄⁺ concentrations measured in the UK and Ireland.



The largest national mean concentration of particulate $NH_4^+$ (1.4 µg $NH_4^+$-N $m^{-3}$) was measured in the Netherlands, which also has highest $NH_3$ and $NO_x$ emissions (Figure 9E). Indeed, the $NH_4^+$ was matched by large $NO_3^-$ concentration (0.9 µg $HNO_3$-N $m^{-3}$) (Figure 9E), lending support to the contribution of $NH_4NO_3$ to the $NH_4^+$ and $NO_3^-$ load, together with contribution from $(NH_4)_2SO_4$ (0.6 µg $SO_4^{2-}$-S) (Figure 9F). The particulate $NH_4^+$ concentrations measured in Italy (mean = 1.0 µg $NH_4^+$-N $m^{-3}$)

(Figure 9E), which includes the site in the Po Valley (IT-PoV) with a mean concentration of 1.9 µg $NH_4^+$-N $m^{-3}$ (Supp. Table S6), is comparable with an assessment of $PM_{2.5}$ composition at 4 sites in the Po Valley (Ricciardelli et al., 2017).

**Oxidised N ($HNO_3$ and $NO_3^-$)**

The percentage mass contribution of oxidised N (sum of $HNO_3$ and $NO_3^-$, µg N $m^{-3}$) to the total gas and aerosol species measured was on average 13 % (range = 2 – 24 %) (Figure 8). This compares with 41 % (range = 17 – 70 %) from reduced N

(sum $NH_3$ and $NH_4^+$, µg N $m^{-3}$), and 23 % (range = 7 – 53 %) from sulfur (sum of $SO_2$ and $SO_4^{2-}$, µg S $m^{-3}$) (Figure 8). DELTA® measurements of $HNO_3$ also include contributions from co-collected oxidised N species such as HONO (see Sect. 2.2.3) and are therefore an upper estimate, that may in some cases be twice as large as the actual $HNO_3$ concentration, based on observations in the UK (Tang et al 2018b; correction factor of 0.45) and from the parallel DELTA® measurements made at Fougéres (FR-FgsP). At this site, $HNO_3$ measurement with NaCl coated denuders provided an annual mean concentration of

0.08 µg $HNO_3$-N $m^{-3}$, compared with 0.19 µg $HNO_3$-N $m^{-3}$ measured on carbonate coated denuders from the main site (FR-Fgs) (Supp. Table S7). With this caveat in mind, uncorrected annual mean $HNO_3$ concentrations were in the range of 0.03 µg $HNO_3$-N at Kaamenan (Finland, FI-Kaa) to 0.47 µg $HNO_3$-N at Braschaat (Belgium, BE-Bra) (Supp. Table S7). In Figure 9B, $HNO_3$ concentrations are compared with $NO_x$ emissions, the precursor gas for secondary formation of $HNO_3$. Russia has the

lowest $NO_x$ emission densities (0.04 t $NO_x$-N $yr^{-1}$), but $HNO_3$ from the single site (0.15 µg $HNO_3$-N $m^{-3}$) is larger than the smallest concentrations measured in Finland, Norway and Sweden (annual average < 0.1 µg $HNO_3$-N $m^{-3}$). $HNO_3$ concentrations in the UK and Ireland are marginally higher than the Scandinavian countries. Here, the annual averaged concentrations of $HNO_3$ are similar (0.10 *vs* 0.09 µg $m^{-3}$) (Supp. Table S7), despite $NO_x$ emissions density (t $km^{-2}$ $yr^{-1}$) in the UK being 3 times larger than in Ireland (Figure 9B). $HNO_3$ concentrations on the European continent were generally higher

(0.2 – 0.4 µg $HNO_3$-N $m^{-3}$). Overall, a weak, but significant correlation was observed between concentrations of $HNO_3$ and $NO_x$ emission densities across the 20 countries ($R^2$ = 0.2, $p < 0.05$) (Figure 10A2, Table 3, Supp. Figure S2).

In the UK, $HNO_3$ data are also available on a wider spatial scale from the AGANet (Tang et al., 2018b, Sect. 2.7.2). The 4-year average concentrations of $HNO_3$ from 30 sites in the AGANet are plotted alongside the NEU $HNO_3$ data from the 4 UK

sites in its network in Figure 9B. The UK $HNO_3$ data on the UK-AIR database (https://uk-air.defra.gov.uk/) have been corrected for HONO interference with a 0.45 correction factor (see Tang et al. 2018b). For consistency in Figure 9B, the UK raw uncorrected $HNO_3$ data are used for the present comparison. The 30-site mean (0.17 µg $HNO_3$-N $m^{-3}$) was higher than from just 4 UK sites in the NEU network (0.10 µg $HNO_3$-N $m^{-3}$). The range of concentrations were also wider, from 0.03 µg $HNO_3$-N $m^{-3}$ at a remote background site in Northern Ireland to 0.77 µg $HNO_3$-N $m^{-3}$ at a central London urban site, where

interference from HONO and $NO_x$ in $HNO_3$ determination is likely to be larger (Tang et al., 2015; 2018b).

Like particulate $NH_4^+$, $NO_3^-$ concentrations are also correlated with emission densities of $NH_3$ ($R^2$ = 0.57, $p < 0.001$, $n = 20$) and $NO_x$ (slope = 0.15, $R^2$ = 0.44, $p < 0.01$, $n = 20$), but not with $SO_2$ (Table 3, Supp. Figure S2). Smallest $NO_3^-$ concentrations were again in Sweden, Norway and Finland with low $NH_3$ and $NO_x$ emissions and also smallest concentrations of $HNO_3$, $SO_2$

and $NH_4^+$ in the network (Figure 9). Largest $NO_3^-$ concentrations was measured in the Netherlands with a mean of 0.92 µg $NO_3^-$-N $m^{-3}$, compared with a network average of 0.39 µg $NO_3^-$-N $m^{-3}$ (Figure 9E, Supp. Table S8). The higher $NO_3^-$ concentrations correlated well with the high $NH_3$, $HNO_3$ and $NH_4^+$ concentrations in the Netherlands (Figure 9). This suggests that concentrations of $NO_3^-$ are linked to local formation of $NH_4NO_3$, which is dependent on concentrations of $NH_3$ and $HNO_3$,





and also to the influence of meteorology on transport of $NH_4NO_3$ between countries on mainland Europe and export out of Europe. Countries in Scandinavia such as Sweden, Norway and Finland and in the British Isles are furthest from the influence of long-range transboundary transport from Europe, with concentrations of $NH_4NO_3$ that are smaller than on the continent.

## 5 Sulfur ($SO_2$ and $SO_4^{2-}$)

Annual averaged $SO_2$ concentrations measured across the network were between 0.9 and 2.3 µg $SO_2$-S m$^{-3}$ (Figure 9C, Supp. Table S9). This corroborates observations from monitoring made in the EMEP networks of large reductions in ambient concentrations and deposition of sulfur species during the last decades (EMEP, 2016), reflecting successes of air quality policies across Europe in achieving substantial reductions in $SO_2$ emissions, which decreased by 74 % between 1990 and 2010.

Annual mean $SO_2$ concentrations of 0.03 to 5.5 µg $SO_2$-S m$^{-3}$ were reported from the EMEP network from 58 rural background sites across Europe over the period of 2007 – 2010, with largest $SO_2$ concentrations from North Macedonia and Serbia (EMEP, 2016). Since the highest emitting countries in European countries were not included in the DELTA® network, the $SO_2$ concentrations provided by the DELTA® network are smaller, but are within the range reported by EMEP (EMEP, 2016).

$SO_2$ concentrations were also correlated with $SO_2$ emission density ($R^2 = 0.65$, $p < 0.001$, $n = 20$) in each country (Figure 10A3, Table 3). The smallest and largest $SO_2$ annual average concentrations corresponded with the lowest emissions in Norway and highest in the Czech Republic (Figure 9C). By contrast, $SO_2$ concentrations from the single measurement site Bugac in Hungary (HU-Bug) are much higher than expected on the basis of $SO_2$ emission density estimated for the country. Gridded emissions for the single grid (0.1° x 0.1°) containing the semi-natural Bugac site are all at the low end of the range of gridded 20 emissions across Hungary for $SO_2$, $NO_x$ and $NH_3$:

- $SO_2$-S: t yr$^{-1}$ = 2.1 (range = < 0.1 to 5144)
- $NO_x$-N: t yr$^{-1}$ = 11 (range = < 0.1 to 3230)
- $NH_3$-N: t yr$^{-1}$ = 63 (range = < 0.1 to 589)

Although the Bugac site is located in a grid with low emissions of all the gases, the higher $SO_2$ (1.2 µg S m$^{-3}$), together with elevated $NH_3$ (2.6 µg N m$^{-3}$) and $HNO_3$ (0.3 µg N m$^{-3}$) concentrations measured at this site suggests that it is likely to be affected by proximity to sources. This contrasts with the BKFores site in the Czech Republic (CZ-BK1) which had smaller $NH_3$ concentrations due to its location away from sources.

Following emission, $SO_2$ disperses and undergoes chemical oxidation in the atmosphere to form $SO_4^{2-}$ both in the gas phase and in cloud and rain droplets (Baek et al., 2004; Jones and Harrison, 2011). Particulate $SO_4^{2-}$ produced is generally associated with $NH_4^+$ and $NO_3^-$ (see Sect. 3.4). The regional pattern of $SO_4^{2-}$ was similar to, and correlated well with, particulate $NH_4^+$ and $NO_3^-$ (Figure 9G), suggesting well-mixed air on the continent, since $(NH_4)_2SO_4$ is stable and long-lived. Countries in the British Isles (UK and Ireland) and in Scandinavia (Sweden, Norway, Finland) have smaller concentrations of $SO_4^{2-}$ (Supp. 35 Table S10). They are located far enough away from sources and activities on continental Europe such that they are less influenced by the emissions from central Europe.

As discussed earlier, particulate $NH_4^+$ and $NO_3^-$ concentrations were smallest in the Scandinavian countries, which corresponded with low emission densities of the precursor gases $NH_3$ and $NO_x$. By analogy, since these countries also have 40 the lowest emission densities of $SO_2$ (Figure 9C), then particulate $SO_4^{2-}$ concentrations would be expected to be similarly low. Particulate $SO_4^{2-}$ in Finland and Norway (mean = 0.34 µg $SO_4^{2-}$-S m$^{-3}$) and Sweden (mean = 0.37 µg $SO_4^{2-}$-S m$^{-3}$) were however comparable with concentrations on mainland Europe (range = 0.33 to 1.0 µg $SO_4^{2-}$-S m$^{-3}$) and larger than the UK (0.18 µg $SO_4^{2-}$-S m$^{-3}$) and Ireland (0.24 µg $SO_4^{2-}$-S m$^{-3}$) (Figure 9F). An ion balance check on the ratio of equivalent concentrations of





NH$_4^+$ to the sum of NO$_3^-$ and SO$_4^{2-}$ (see next section 3.4) was less than 0.5. Since NH$_4^+$ is a counter-ion to NO$_3^-$ and SO$_4^{2-}$ formation, the imbalance suggests that SO$_4^{2-}$ concentrations may be over-estimated at the sites in Sweden, Norway and Finland.

**HCl, Cl$^-$ and Na$^+$**

The average concentrations of HCl across the network were of low magnitude, with limited variability, ranging from 0.07 in Russia to 0.36 µg HCl-Cl$^-$ m$^{-3}$ in Portugal (Figure 9D). At a site level, HCl concentrations varied between 0.06 at Renon (Italy, IT-Ren – inland location) to 0.48 µg HCl-Cl$^-$ m$^{-3}$ at Espirra (Portugal, PT-Esp – coastal location) (Supp. Table S11). In the UK AGANet network, the highest concentrations of HCl were found in the source areas in SE and SW of England, and also in central England, north of a large coal-fired power station (Tang et al., 2018b). HCl emissions and concentrations in the

atmosphere are mostly derived from combustion of fossil fuels (coal and oil), biomass burning and from the burning of municipal and domestic waste in municipal incinerators (Roth and Okada 1998; McCulloch et al., 2011; Ianniello et al., 2011). Several manufacturing processes, including cement production also emits HCl (McCulloch et al., 2011). At coastal sites, HCl released from the reaction of sea salt with HNO$_3$ and H$_2$SO$_4$ can be a significant source (Roth and Okada 1998; Keene et al., 1999; McCulloch et al., 2011; Ianniello et al., 2011). UK is the only country with available HCl emission estimates

(https://naei.beis.gov.uk/data/). Emissions of HCl in the UK (mainly from coal burning in power stations) have declined to very low levels, from 74 kt in 1999 to 5.7 kt in 2015. The 4-year averaged emission density for HCl for the period 2007 to 2010 was just 0.05 tonnes HCl-Cl$^-$ km$^{-2}$ yr$^{-1}$, although HCl emissions could still pose a threat to sensitive habitats close to sources (Evans et al., 2011). The low HCl concentrations measured in the network would suggest that the shift in Europe's energy system from coal to other sources has contributed to low HCl emissions (UK) and concentrations (observed across the

network).

Particulate Cl$^-$ on the other hand is predominantly marine in origin, with sea salt (NaCl) as the most significant source (Keene et al. 1999). Molar concentrations of Cl$^-$ and Na$^+$ are seen to be similar in most countries, demonstrating close coupling between the two components (Figure 9H). Largest concentrations of Na$^+$ and Cl$^-$ occurred at coastal countries such as the UK, Ireland,

Netherlands and Portugal, with the highest of country-averaged annual concentrations of 1.6 µg Cl$^-$ m$^{-3}$ and 0.9 µg Na$^+$ m$^{-3}$ from Ireland (Supp. Tables S12 and S13). Data from the 30 sites in the UK AGANet network showed a wider range of Cl$^-$ and Na$^+$ concentrations (Figure 9H), with the highest 4-year annual averaged concentrations of 3.8 µg Cl m$^{-3}$ and 2.0 µg Na$^+$ m$^{-3}$ from the coastal Lerwick monitoring site on the east coast of the Shetland Islands, exposed to the North Atlantic.

Further away from the coastal influence of marine aerosol, the smallest concentrations of Cl$^-$ and Na$^+$ were measured in land-locked countries such as Germany (mean of all sites = 0.27 µg Cl$^-$ m$^{-3}$ and 0.15 µg Na$^+$ m$^{-3}$). Concentrations in Hungary, Poland, the Czech Republic and Russia were also low, but inferences about these countries are necessarily limited by measurements at a single site in each of these countries. At coastal sites in Norway (NO-Bir) and Sweden (SE-Nor and SE-Sk2), the very low particulate Cl$^-$ concentrations (< 0.1 - 0.3 µg m$^{-3}$), and high Na:Cl molar ratios (3 – 5) are anomalous. It is

possible for sea salt to be depleted in Cl- (through the loss of HCl gas) by the reaction of NaCl particles with atmospheric acids (Finalyson-Pitts and Pitts, 1999; Keene et al., 1999), leading to high Na:Cl ratios for sea salts transported over long distances. The coastal locations of these sites (Figure 2) suggests that they are more likely to be influenced by freshly generated marine aerosols (*cf.* coastal sites in UK and Ireland), and larger concentrations of sea salt (Na$^+$ and Cl$^-$) and a 1:1 relationship between Na$^+$ and Cl$^-$ are expected. The Cl$^-$ concentrations are likely to be under-estimated at these sites (see Sect. 3.2.3) and further

discussed in the next section (Sect. 3.4).





### 3.4 Correlations between gas and aerosol components

Regression analyses was carried out between the mean molar equivalent concentrations of all inorganic gas and aerosol components measured at each site ($n = 66$; Fr-FgsP and UK-AmoP excluded) in the NEU network, with summary statistics provided in Table 5. With the exception of $SO_2$ $vs$ HCl ($R^2 = 0.05$, $p > 0.05$), the gases were positively correlated with each

other, possibly due to similarities in the regional distribution of their emissions and concentrations. Comparing the mean molar concentrations of $NH_3$ with $SO_2$ and $HNO_3$ showed that $NH_3$ was on average 6-fold and 7-fold higher, respectively, whereas molar concentrations of $SO_2$ and $HNO_3$ were similar (Table 6, Figure 11). The molar ratio of $NH_3$ to the sum of all acid gases ($SO_2$, $HNO_3$ and HCl) was on average 3 (Table 6, Figure 11), confirming that there is a surplus of the alkaline $NH_3$ gas to neutralise the atmospheric acids in the atmosphere, similar to that observed in the UK (Tang et al., 2018b). With the more

substantial decline in emissions of $SO_2$, compared with a more modest reduction in $NO_x$, the concentrations of $SO_2$ are at a level where it is no longer the dominant acid gas, such that $HNO_3$ and HCl are together contributing a larger fraction of the total acidity in the atmosphere in the present assessment.

<INSERT TABLE 5>

<INSERT TABLE 6>

<INSERT FIGURE 12>

In the aerosol phase, $NH_4^+$ correlated well with $NO_3^-$ ($R^2 = 0.75$, $p < 0.001$, Figure 12A) and $SO_4^{2-}$ ($R^2 = 0.75$, $p < 0.001$, Figure 12) (Tables 5 and 7), but not with Cl$^-$ (Table 5). Regression of the molar equivalent concentrations of the sum of $NO_3^-$ and

$SO_4^{2-}$ against $NH_4^+$ show points close to the 1:1 line (slope = 0.84) and significant correlation ($R^2 = 0.64$, $p < 0.001$), which demonstrates the close coupling between the base $NH_4^+$ and the acid $NO_3^- + SO_4^{2-}$ aerosols (Figure 12C, Table 7). The reaction of $NH_3$ with $H_2SO_4$ is irreversible (i.e. 'one-way') under atmospheric conditions (Baek et al., 2004; Finlayson-Pitts and Pitts, 1999; Jones and Harrison, 2011; Huntzicker et al., 1980), whereas any $NH_4NO_3$ or $NH_4Cl$ that are formed can dissociate to release $NH_3$ which can then be 'removed' by reaction with $H_2SO_4$. The lack of correlation between $NH_4^+$ and Cl$^-$ ($R^2 = 0.00$,

Table 5) in the analysis suggests that $NH_4^+$ is mainly associated with $NO_3^-$ and $SO_4^{2-}$.

<INSERT TABLE 7>

Particulate Cl$^-$ was correlated with $Na^+$ ($R^2 = 0.65$, $p < 0.001$) (Figure 12F, Tables 5, 7), consistent with observations that NaCl

in atmospheric aerosols are mainly sea salt in origin (O'Dowd and de Leeuw, 2007; Tang et al., 2018b). Like the precursor gases, the molar concentrations of particulate $NH_4^+$ are larger than either $NO_3^-$ or $SO_4^{2-}$ (Figure 12, Table 8). Particulate $NO_3^-$ concentrations were on average 2-fold higher than particulate $SO_4^{2-}$ (on a molar basis), so that there was twice as much $NH_4NO_3$ (Figure 12A) as $(NH_4)_2SO_4$ (Figure 12B). The shift in PM composition from $(NH_4)_2SO_4$ to $NH_4NO_3$ across Europe is well documented (Bleeker et al., 2009; Fowler et al., 2009; Tang et al. 2018b; Torseth et al., 2017).

<INSERT FIGURE 13>

<INSERT TABLE 8>

Non-sea salt $SO_4^{2-}$ (nss-$SO_4^{2-}$) was also estimated from the $SO_4^{2-}$ and $Na^+$ data (see Sect. 2.2.1). The nss-$SO_4^{2-}$ is estimated to

comprise on average 25 % (range = $3 - 83$ %, $n = 187$) of the measured total $SO_4^{2-}$ aerosol (Table 8). This demonstrates that sea salt $SO_4^{2-}$ (ss-$SO_4^{2-}$) aerosol makes up a large and variable fraction of the total $SO_4^{2-}$ measured, consistent with observations of the contribution by ss-$SO_4^{2-}$ to the total $SO_4^{2-}$ in precipitation (ROTAP, 2012). Regression of nss-$SO_4^{2-}$ $vs$ $NH_4^+$ (slope =





0.27, $R^2$ = 0.30) was not significantly different from the regression of $SO_4^{2-}$ *vs* $NH_4^+$ (slope = 0.27, $R^2$ = 0.28) (Table 5). This suggests that $NH_4^+$ is mainly associated with the nss-$SO_4^{2-}$.

Correlation between $NH_4^+$ and the sum of anions ($NO_3^- + SO_4^{2-}$) is an important point of discussion (Table 7), as the ion balance

serves as a quality check for the aerosol measurement. Due to some outliers in the comparison, the correlation between $NH_4^+$ and $SO_4^{2-}$ ($R^2$ = 0.28, Figure 12B) is weaker than between $NH_4^+$ and $NO_3^-$ ($R^2$ = 0.75, Figure 12C, Table 7). The outliers were measurements made by NILU and CEAM, although these vary according to monitoring locations. The NILU laboratory made DELTA® measurements for 16 sites in 6 different countries (Belgium, Denmark, Finland, Norway, Sweden and Switzerland). At 3 sites (Kaamanen FI-Kaa, Laegern CH-Lae, Oensingen CH-Oe1), the ion balance of equivalent concentrations of

$NH_4^+$:sum($NO_3^- + SO_4^{2-}$) was 1.0, whereas the ratios at the other 13 sites were between 0.4 and 0.7. The CEAM laboratory made measurements for all 3 sites in Spain. For CEAM, the ion balance ratio at Vall de Aliñá (ES-VDA) was 1, whereas the other 2 sites had ratios of 0.5 and 0.6.

Removal of the outlier NILU (7 out of 16) and CEAM (1 out of 3) data points with ion balance ratio < 0.5 improved both the

slope (new slope = 0.90) and correlation (new $R^2$ = 0.78) (Figure 12C). This indicates either an over-read of the anions ($NO_3^-$, $SO_4^{2-}$) or under-read of $NH_4^+$ concentrations by the two laboratories at some sites. Results reported by NILU in the DELTA® field inter-comparisons (Sect. 3.2) showed that, with the exception of a few high $NH_4^+$ and $NO_3^-$ readings, there was on average no overall bias in the $NH_4^+$, $NO_3^-$ or $SO_4^{2-}$ measurements by the NILU laboratory that could account for the high $SO_4^{2-}$ outliers in the regression (Figure 12). The ion balance checks suggest possible over-read and increased uncertainty in the $SO_4^{2-}$

measurements for 7 sites: Hyytiälä (FI-Hyy), Sodankylä (FI-Sod), Rimi (DK-Rim), Risbyholm (DK-Ris), Soroe (DK-Sor), Skyttorp (SE-Sk2) and Vielsalm (BE-Vie). For the CEAM lab, the uncertainty in $SO_4^{2-}$ measurements affected 2 sites, El Saler (ES-Els) and Las Majadas (ES-Lam) (see also Sect. 3.3.4).

The regression of $Na^+$ and $Cl^-$ also showed the majority of data points close to the 1:1 line, but with a small group of outliers

below the 1:1 line from the CEAM and NILU laboratories (Figure 12F). Both laboratories performed well in laboratory PT schemes (Sect. 3.1), with more than 80% of reported data agreeing within ± 10% of reference values in both $Na^+$ and $Cl^-$, with no bias in the analytical method. The outliers in the ion balance therefore suggests some problems with $Na^+$ and $Cl^-$ determination on the DELTA® aerosol filters. $Na^+$ and $Cl^-$ data for some of the field DELTA® inter-comparisons were omitted from submissions by CEAM and NILU, and submitted data were in poor agreement with other laboratories (Sect. 3.2). Further

regression analyses were carried out on individual monthly data, with sites grouped according to measurements made by each of the seven laboratories (Supp. Figure S5). Regressions for CEAM and NILU show the vast majority of data points below the 1:1 line, indicating a systematic under-estimation of particulate $Cl^-$ concentrations. The other 5 laboratories (INRAE, MHSC, SHMU, UKCEH and VTI) all have data points close to the 1:1 line, with larger scatter both above and below the 1:1 line at lower concentrations. In Figure 12F, a new regression line has therefore also been fitted where outlier data with Na:Cl ratios

> 2 from NILU (13 out of 16 sites) and CEAM (all 3 sites) have been removed. Exclusion of the outlier data points provided a regression line that is not significant different from unity (slope = 1.02), with a $R^2$ value of 0.95 ($p < 0.001$). The near 1:1 relationship between particulate $Na^+$ and $Cl^-$ is consistent with their origin from sea salt (NaCl).

The ion balance checks, together with the regular PT exercises and field inter-comparisons therefore provided the platform

against which to assess data quality and comparability of measurements between laboratories. This shows that overall, with the exception of a few identified outlier measurements, the laboratories are performing well and providing good agreement.



### 3.5 Seasonal variability in gases and aerosol

The time series of monthly averaged concentrations for the period 2006 to 2010 have been plotted to examine seasonality in the different gas and aerosol components according to ecosystem types (crops, grassland, semi-natural and forests) (Figure 14) and geographical regions (Figure 15). Distinct seasonality were observed in the data, influenced by seasonal changes in emissions, chemical interactions and the influence of meteorology on partitioning between the main inorganic gases and aerosol species.

**\<INSERT FIGURE 14>**
**\<INSERT FIGURE 15>**

#### 3.5.1 NH$_3$

Distinctive and contrasting features in the seasonal cycle are observed, with largest concentrations at cropland sites and smallest at semi-natural and forest sites (Figure 14A). Similar to that observed in the annual mean concentrations (Figure 9, 11), the monthly concentrations are also smallest in Northern Europe and largest in Western Europe (Figure 15A).

Semi-natural sites:
There are two distinct peaks in the seasonal cycle of grouped semi-natural sites, in April (mean = 2.2 µg NH$_3$ m$^{-3}$, $n$ = 12) and in July (mean = 1.9 µg NH$_3$ m$^{-3}$, $n$ = 12) (Figure 14A). Since these sites are located away from agricultural sources, the seasonality in NH$_3$ concentrations is mostly governed by changes in environmental conditions and regional changes in NH$_3$ emissions. The differences in concentrations between the summer and winter at these sites was by a factor of 3, with smallest concentrations in wintertime (Dec and Jan) when low temperatures and wetter conditions decrease NH$_3$ emissions from regional agricultural sources, while favouring a thermodynamic shift from gaseous NH$_3$ to the aerosol NH$_4^+$ phase. Conversely, warm, dry conditions in summer increases surface volatilization of NH$_3$ from low density grazing livestock and wild animals, and favour a thermodynamic shift to the gaseous (NH$_3$) phase, producing the summer peak. Vegetation is another potential source at these background sites under the right conditions (Flechard et al., 2013; Massad et al., 2010). A complex interaction between atmospheric NH$_3$ concentrations and vegetation can lead to both emission and deposition fluxes known as "bi-directional exchange", dependent on relative differences in concentrations. This process is controlled by the so-called "compensation point", defined as the concentration below which growing plants start to emit NH$_3$ into the atmosphere (Flechard et al., 1999; Massad et al., 2010; Sutton et al., 1995). At sites distant from intensive farming and emissions, the bi-directional exchange with vegetation will partly control NH$_3$ concentrations. Inclusion of bi-directional exchange in dispersion modelling of NH$_3$, by incorporating a 'canopy compensation point' is shown to improve model results for NH$_3$ concentrations in remote areas (e.g. Smith et al., 2000; Flechard et al., 1999, 2011; Massad et al., 2010). The larger peak in April at these sites on the other hand suggests the influence of emissions from agricultural sources, e.g. from land spreading of manures.

Forest sites:
The average seasonal cycle from the forest sites is similar to that of the semi-natural sites, but diverged over the summer months (Figure 14A). Here, the seasonal profile is characterised by the absence of any peaks in summer, with concentrations plateauing between May and August. Studies have shown that atmospherically deposited N is taken up by forest canopies, since growth in forest ecosystems is commonly limited by the availability of N (Sievering et al., 2007) and tree canopies are a potential sink for atmospheric NH$_3$ (Fowler et al., 1989; Theobald et al., 2001). The capture and uptake of NH$_3$ during the growing seasons over the summer period could therefore account for the absence of a summer peak in NH$_3$ concentrations at forest monitoring sites, although a similar effect would also be expected for semi-natural sites.



Cropland sites:

Fertilizers and arable crops are significant sources of $NH_3$ emissions and concentrations in an intensive agricultural landscape. Sites in this group showed considerably higher monthly mean monitored $NH_3$ concentrations than the other groups (Figure 14A). A more complex seasonal pattern can be seen, with three peaks in $NH_3$ concentrations. Concentrations here are also lowest in the winter, although the wintertime concentrations are 3 times larger than semi-natural and forest sites, reflecting the elevated regional background in $NH_3$ concentrations located within agricultural landscapes. This rises rapidly with improving weather conditions and peaks in the spring to coincide with the main period for manure spreading and fertiliser application before the sowing of arable crops (Hellsten et al., 2007). The distinct springtime maxima in $NH_3$ also reflects implementation of the Nitrates Directive (91/676/EEC), which prohibits manure spreading in winter. In summer, the second peak in $NH_3$ concentrations may be associated with increased land surface emissions promoted by warm, dry conditions, and possibly from the application of fertilisers. The smaller autumn peak is also expected to be related to seasonal farming activities/manure spreading. The key drivers for seasonal variability in $NH_3$ concentrations at crops sites are therefore a combination of seasonal changes in agricultural practices (e.g. timing of fertiliser/manure applications) and climate that will affect emissions, concentrations, transport and deposition of $NH_3$.

Grassland sites:

An additional major source of $NH_3$ in this group of sites is expected to come from grazing emissions and housed livestock (e.g. cattle). Concentrations in this group of sites were generally 2 - 3 times larger than semi-natural sites (Figure 13A), attributed to the increased emissions and concentrations from livestock (Hellsten et al., 2007). The spring peak is related to the practice of fertiliser and manure being spread on grazing fields to aid spring grass growth, which will be cut for hay and silage later in the year. $NH_3$ concentrations in June and July are smaller than in spring or late summer, possibly because grass will be actively growing with possible uptake and removal of $NH_3$ from the atmosphere. The concentrations are also larger in summer than winter, with warmer conditions promoting $NH_3$ volatilization and thermodynamic shift to the gas phase.

European regions:

The seasonal profiles of $NH_3$ for Central and Western European regions were similar, characterised by a large peak in spring that is likely to be agriculture–related (Figure 15A), as observed at cropland sites (Figure 14A). While the peak concentrations in both regions are of comparable magnitude (Central = 2.6 µg $NH_3$ m$^{-3}$, Western = 2.8 µg $NH_3$ m$^{-3}$), winter concentrations in the Centre Europe (0.6 µg $NH_3$ m$^{-3}$) were three times smaller than the West (1.5 µg $NH_3$ m$^{-3}$). This may be related to either lower regional background in $NH_3$ concentrations and/or suppressed emissions in colder temperature of Central Europe.in winter. By contrast, Eastern and Southern European regions have a broad peak in summer, although the Eastern region also has a second peak in October (likely agriculture related). Smallest concentrations were found in Northern Europe with the lowest $NH_3$ emissions (Figure 9). The three peaks in the profile shows elevated concentrations in summer driven by warming temperatures, with the spring and autumn peaks attributed to influence from $NH_3$ emissions from agricultural sources.

### 3.5.2    HNO$_3$

The seasonal distribution in $HNO_3$ is similar between the different ecosystem groups, varying only in magnitude of concentrations (Figure 14C) and reflects the secondary nature of this component that is formed from oxidation of $NO_x$ (Fahey et al., 1986; ROTAP, 2012). $HNO_3$ concentrations in the crops group are up to 2 times larger than the grassland group, while the smallest concentrations are in the semi-natural group. This is likely related to proximity of sites in the different groups to combustion sources. A weak seasonal cycle is seen in the secondary $HNO_3$ air pollutant in all cases, with slightly higher



concentrations in late winter, spring and summer and smallest in March and November. The reaction of $NO_2$ with the OH radical is an important source of $HNO_3$ during daytime, whereas $N_2O_5$ hydrolysis is considered an important source of $HNO_3$ at night time (Chang et al., 2011). Larger $HNO_3$ concentrations in summer are therefore from increased OH radicals for reaction with $NO_2$ to form $HNO_3$. Similarly, higher concentrations of ozone in spring in Europe (EMEP, 2016) can potentially increase

$HNO_3$ concentrations in springtime. Conversely, $HNO_3$ concentrations are lower in winter when oxidative capacity is less.

Seasonal variability in $HNO_3$ will also be influenced by gas-aerosol phase equilibrium. In the atmosphere, $HNO_3$ reacts reversibly with $NH_3$ forming the semi-volatile $NH_4NO_3$ aerosol if the necessary concentration product $[HNO_3].[NH_3]$ is exceeded (Baek et al., 2004; Jones and Harrison et al., 2011). Because of this process, the prime influences upon $HNO_3$

concentrations at sites where $NH_4NO_3$ is formed are expected to be ambient temperature, relative humidity and $NH_3$ concentrations that affect the partitioning between the gas and aerosol phase (Allen et al., 1989; Stelson and Seinfeld, 1982). The availability of surplus $NH_3$ in spring (Sect. 3.5.1) would tend to reduce $HNO_3$ and increase $NH_4NO_3$ formation, which is reflected in the reduced $HNO_3$ concentrations observed in March when $NH_3$ is at a maximum. In summer, warmer, drier conditions promotes volatilisation of the $NH_4NO_3$ aerosol, increasing the gas phase concentrations of $HNO_3$ and $NH_3$ relative

to the aerosol phase. Seasonality in $HNO_3$ is therefore complex, related to traffic and industrial emissions, photochemistry and $HNO_3$:$NH_4NO_3$ partitioning.

An analysis of the same data grouped according to geographical regions revealed distinctive cycles in $HNO_3$ in Eastern and Southern Europe (Figure 15C). These two regions showed highest concentrations in summer and smallest in winter, consistent

with enhanced photochemistry in warmer, sunnier climates and thermodynamic equilibrium favouring gas phase-$HNO_3$ (Figure 15C). Summertime peak concentrations in $NH_3$ were also observed in these 2 regions (Figure 15A). In comparison, the seasonal profiles of $HNO_3$ in other regions were similar to that described for different ecosystem types (Figure 14C).

### 3.5.3    $SO_2$

Seasonality in $SO_2$ show concentrations peaking in winter at most sites (Figure 14E), except in Southern Europe where the peak appeared in summer (Figure 15E). Increased $SO_2$ emissions from combustion processes (heating) in the winter months, coupled to stable atmospheric conditions can result in build-up of concentrations at ground level, thereby contributing to the peak wintertime concentrations. The largest winter concentrations in Central and Eastern regions exceeded summer values on average by a factor of 4, compared with smaller differences in other regions (Figure 15E). Enhanced oxidation processes in

summer also tend to further reduce concentrations of $SO_2$ through the oxidation of $SO_2$ to $H_2SO_4$ (Saxena and Seigneur, 1987; Sickles and Shadwick, 2007; Paulot et al., 2017). In Southern Europe, the seasonal cycle have winter minima and summer maxima instead, likely from increased combustion sources to meet energy demands for air-conditioning over the hot summer months. It was shown earlier in Section 3.4 that $SO_2$ was spatially correlated to $HNO_3$; differences in relative concentrations between the different ecosystem groups (Figure 14E) is thus also likely related to relative distance from emission sources.

### 3.5.4    $NH_4^+$, $NO_3^-$ and $SO_4^{2-}$

The seasonal profiles of particulate $NH_4^+$ (Figures 14B and 15B) were mirrored by particulate $NO_3^-$ (Figures 14D and 15D) in all groups, demonstrating temporal, as well as regional (see Sect.3.3.5) correlation between these two components. Since $NH_4NO_3$ is more abundant than $(NH_4)_2SO_4$, the seasonality of $NH_4^+$ is likely to be influenced more by the temperature and

humidity dependence of the semi-volatile $NH_4NO_3$, than by the stable $(NH_4)_2SO_4$. In summer, warmer and drier conditions promotes the dissociation of $NH_4NO_3$, decreasing particulate phase $NH_4NO_3$ relative to gas phase $NH_3$ and $HNO_3$. This process





accounts for the summertime minima in $NH_4^+$ (Figures 15B and 15B) and $NO_3^-$ (Figures 14D and 15D). Conversely, cooler temperatures and higher humidity conditions in winter, spring and autumn shift the equilibrium to the aerosol phase, with observed peaks in concentrations of $NH_4^+$ and $NO_3^-$. Since $NH_3$ concentrations are also generally higher in spring than in autumn (Figure 14A, 15A), the increased availability of $NH_3$ in this period contributes towards the higher concentrations of

$NH_4NO_3$ in spring than in autumn. In winter, the combination of $NH_4NO_3$ remaining in the aerosol phase, combined with the stable conditions that can often develop, maintains high concentrations of $NH_4^+$ and $NO_3^-$ in the atmosphere. The peak in $NO_3^-$ in Southern Europe was in February only, compared with broader peaks (Feb-April) in other regions (Figure 15D) which may reflect differences in climatic conditions. In Figures 14H and 15H, the ratio of the molar equivalent concentrations of $NO_3^-$ to $sum(NO_3^- + SO_4^{2-})$ are plotted. The ratios were highest in spring and autumn, and smallest in summer, lending support to the

importance of $NH_4NO_3$ in controlling the seasonality of $NH_4^+$.

In the seasonal profiles for particulate $SO_4^{2-}$, clear summer maxima and winter minima were provided by sites in Southern and Eastern Europe (Figure 15F). The peaks occurred at different times, in July (Southern Europe) and in August (Eastern Europe) (Figure 15F) and coincided with the timing of corresponding peaks in $NH_3$ concentrations (Figure 15A), illustrating the

importance of $NH_3$ in driving the formation of the stable $(NH_4)_2SO_4$. Since $(NH_4)_2SO_4$ is formed through the preferential and irreversible reaction between the precursor gases (Bower et al., 1997), particulate $SO_4^{2-}$ concentrations will be governed by the availability of $NH_3$ and $H_2SO_4$ (from oxidation of $SO_2$). As discussed earlier, $SO_2$ concentrations in Southern Europe have a different seasonal cycle from other regions, with higher concentrations in summer than in the winter months (Figure 15E). Although the seasonal cycle for Eastern Europe showed smallest $SO_2$ concentrations in the summer, the summer minima here

(mean = 1.3 µg $SO_2$ m$^{-3}$) are in fact larger than the summer peak in Southern Europe (mean = 1.1 µg $SO_2$ m$^{-3}$) and concentrations in other regions (0.4 - 1.0 µg $SO_2$ m$^{-3}$). Enhanced summertime concentrations in $HNO_3$ were observed in these two regions (Figure 15B) which also suggests potentially increased oxidative capacity for more of the $SO_2$ to be converted $H_2SO_4$ (Sect. 3.5.3). The ready availability of both $SO_2$ (and conversion to $H_2SO_4$) and $NH_3$ (Figure 15A) in Southern and Eastern regions in this period thus coincide to produce the summer peak in particulate $SO_4^{2-}$.

In other regions (Central, Northern, Western), formation of $(NH_4)_2SO_4$ will be limited by the availability of $SO_2$ which is lowest in summer (Figures 15E). Conversely, $SO_2$ concentrations is highest in winter (Figures 15E), but lower oxidative capacity at this time of year limits formation of $H_2SO_4$. Since $NH_3$ concentrations are also smallest in winter (Figures 15A), formation of $(NH_4)_2SO_4$ is also limited in winter. This accounts for the higher concentrations of particulate $SO_4^{2-}$ concentrations

in winter and in early spring in these regions (Figure 15F).

### 3.5.5   HCl, Cl$^-$ and Na$^+$

The concentrations of HCl measured at all sites, in all groups, were very small, with monthly mean concentrations varying between 0.1 and 0.3 µg HCl m$^{-3}$ (Figures 14G and 15G). There is no discernible seasonality in the data, which suggests either

sites in the network are not affected by any large sources of HCl, or that small differences between months are not detectable due to measurement uncertainties at the very low concentrations (method limit of detection ~ 0.1 µg HCl m$^{-3}$ for monthly sampling). By contrast, Cl$^-$ (Figures 14I and 15I) has a distinctive seasonal cycle with higher concentrations in the winter months than summer, similar to that of Na$^+$ (Figures 14J and 15J). The temporal correlation in the data therefore lends further support that Na$^+$ and Cl$^-$ in the measurements are mainly sea salt (see also spatial correlation in Sect. 3.4). The highest

concentrations of Na$^+$ and Cl$^-$ during winter months would be consistent with increased generation and transport of sea salt generated by more stormy weather from marine sources during those periods (O'Dowd and de Leeuw, 2007).



### 3.6 Bulk wet deposition measurements

Annual mean wet deposition of chemical species measured at the NEU bulk sampling sites was estimated by combining measured concentrations with annual precipitation. Site changes also occurred during the operation of the bulk wet deposition network, with some sites closed and new sites added. At Mitra (PT-Mi3), contamination of the rain samples from bird strikes

resulted in the rejection of a large proportion of the monthly data and this site was excluded from the data analysis. In total, 12 sites provided 2 years of monthly data, with a further 5 sites providing 1 year of monthly data over the period 2008 to 2010. Due to differences in start and end dates for bulk measurements between the sites, the annual mean data derived are for 12 month periods or 2 x 12 month periods, and not from calendar years.

**<INSERT FIGURE 16>**

Annual mean wet deposition data for the 17 sites from 6 countries (Belgium, France, Germany, Italy, Poland, Spain and Switzerland) are summarised in Figure 16. Using $Na^+$ as a tracer for sea-salt (Keene et al,. 1986), $nss-SO_4^{2-}$ concentrations were also estimated from the total $SO_4^{2-}$ (see Sect. 2.2.2) and are included for comparison. Since the measurements were made

at a limited number of sites across Europe, there is insufficient information to make inferences about spatial differences in concentrations. Detailed assessments of extensive precipitation chemistry across Europe are made elsewhere, for example from the EMEP wet deposition networks (EMEP, 2016; Torseth et al., 2012). What the NEU bulk network data clearly shows is that $N_r$ components in rain also exceed that of S (Figure 16), as was observed in the atmospheric data. The mean proportional contribution of total N ($NH_4^+$ and $NO_3^-$) to the sum total of all wet deposited species measured (by mass) was 19% (range = 3

– 39%), compared with a smaller 9 % (range = 1 – 19%) contribution from $nss-SO_4^{2-}$ (Supp. Table S14). Wet deposited N ($NH_4^+$ and $NO_3^-$) was on average 2 times higher than $nss-SO_4^{2-}$, similar to that seen in the relative proportion of total $N_r$ (sum of $NH_3$, $NH_4^+$, $HNO_3$, $NO_3^-$) to total S (sum of $SO_2$, $SO_4^{2-}$) in the atmospheric data (Sect. 3.3.5). Similar to the atmospheric data (Sect. 3.3.5), a considerable fraction of the wet deposited components was made up of sea salt ($Na^+$ and $Cl^-$), with the sum of $Na^+$ and $Cl^-$ contributing on average 50% of the total wet deposited components (range = 20 – 84 %, $n$ = 17). Contributions

by the other base cations $Ca^{2+}$ and $Mg^{2+}$ gave a further 20 % (range = 8 – 41 %, $n$ = 17) (Supp. Table S14).

The intention of the bulk network at the outset was to provide wet deposition data at DELTA® sites that do not already have such measurements on site. The wet deposition data on $NH_4^+$ and $NO_3^-$, combined with a wider precipitation chemistry dataset (e.g. from EMEP and other national precipitation networks) was used to estimate total $N_r$ deposition to a site (Flechard et al,

2011; 2020). Together, the dry (DELTA® network) and wet $N_r$ estimates (NEU bulk network, combined with data from other national precipitation chemistry networks) are used to compare with EMEP models and to examine the interactions between $N_r$ supply and greenhouse gas exchange at the NEU DELTA® sites, presented in a separate paper by Flechard et al. (2020).

### 4 Implications for a chemical climate dominated by $NH_3$ and $NH_4NO_3$ in Europe

International agreements such as the UNECE Convention on Long-Range Transboundary Air Pollution (CLRTAP 1999

Gothenburg Protocol, amended in 2012) (UNECE, 2018), NEC Directive 2016/2284 (revised also in 2012) (EU, 2016) and Ambient Air Quality Directives (EU Directive 2008/50/EC) (EU, 2008) have achieved reductions in emissions of $SO_2$ and $NO_x$, but with limited ambition in $NH_3$. The amended NEC Directive (2016/2284) sets further emission reduction commitments for $SO_2$, $NO_x$, $NH_3$, as well as primary fine particulate matter ($PM_{2.5}$), for the years 2020 to 2029 with 2005 as the base year and additional reductions beyond 2030. Provisions for ecosystem monitoring under Article 9 and Annex V of Directive

2016/2284 (EU, 2016) also require member states to monitor (Article 9) and report (Article 10.4) the negative impacts of $NH_3$,





$SO_2$ and $NO_x$ on ecosystems from national networks that are representative of the Member State's freshwater, natural and semi-natural habitats and forest ecosystem types.

In 2017, the European Commission published tighter new standards for large combustion plants, including many large coal-
fired power stations, giving them four years to meet the standards, detailed in the Decision (EU) 2017/1442 under Directive 2010/75/EU (EU, 2017). Tighter rules are set for emissions of $NO_x$, $SO_2$ and PM and concentrations of these are expected to continue to fall in future years. Measurements in the network have shown that the concentrations of $SO_2$ have declined to a level where it is no longer the dominant acid gas, such that $HNO_3$ and $HCl$ are together contributing an equal or larger fraction of the total acidity in the atmosphere in the present assessment (Figure 11). However, $SO_2$ (by mass) has a higher acidification
potential (1 kg $SO_2$ = 1.00 kg eq. $SO_2$ than $NO_x$ (1 kg $NO_2$ = 0.70 kg eq. $SO_2$ (see Hauschild and Wenzel, 1998), so $SO_2$ will remain important in contributing to exceedances of critical loads for acidification, estimated to be exceeded in 5 % of the European ecosystem area in 2015 (EEA, 2019).

Emissions of $NH_3$ in Europe have increased by about 3% from the agricultural sector between 2013 - 2016 (EEA, 2018) and
abatement measures are likely to be needed to meet emission targets set for $NH_3$. (Sutton and Howard, 2018). Thresholds for atmospheric concentrations and deposition of $N_r$ components to semi-natural habitats were exceeded in 63% of the EU-28 ecosystem area in 2016 (EMEP, 2018). In deposition models, oxidised nitrogen species currently included are $HNO_3$, $NO_2$ and aerosol nitrate ($NO_3^-$), with deposition velocities dependent on meteorology and vegetation characteristics (e.g. Flechard et al, 2011). $NH_3$ is the most important individual term in the calculation of total N dry deposition, along with $NH_4^+$ and $HNO_3$ dry
deposition and wet deposited $NH_4^+$ and $NO_3^-$. Although $NO_2$ (not measured in NEU DELTA® network) will also provide a relevant contribution to dry N deposition, it will (especially for rural semi-natural and forest ecosystems) be smaller than for $NH_3$, based on rather small deposition velocities for $NO_2$ (Smith et al., 2000).

The annually averaged data also show exceedance of the Critical Levels for annual mean $NH_3$ concentrations of 1 and 3 µg
$NH_3$ $m^{-3}$ for the protection of lichens-bryophytes (including ecosystems where they are important for integrity) and other vegetation, respectively, at many of the sites (62 % > 1 µg $NH_3$ $m^{-3}$ and 27 % > 3 µg $NH_3$ $m^{-3}$) (Supp. Table S5). The widespread exceedance of the Critical Levels for $NH_3$ concentrations across Europe represents an ongoing threat to the integrity of sites designated under the EU Habitats Directive (EU, 1992). In tandem, the growing relative importance of $NH_3$ and $NH_4^+$ to total acidic and total nitrogen deposition indicates that strategies to tackle acidification and eutrophication will also need to include
measures to abate emissions of $NH_3$.

The agricultural sector makes up 92% of the total estimated $NH_3$ emission in Europe (EEA, 2019), with 80 % of that generated by less than 10 % of the farms, so that the largest emission reduction potential could be attained by targeting the small number of industrial-scale farms (Maas and Greenfelt, 2016). A modelling study by Backes et al. (2016) suggested a halving of $NH_3$
emissions could deliver a 24% reduction in total $PM_{2.5}$ concentrations in northwest Europe, driven mainly by reduced formation of $NH_4NO_3$ and that targeting emission reductions during winter had a larger effect than at other times of the year. In recognising the need to tackle $NH_3$, the UNECE has published a guidance document and code of good agricultural practice (COGAP) for reducing $NH_3$ emissions (Bittman et al., 2014), which has also been adopted in the EC NECD and by the UK government in its Clean Air Strategy (Defra, 2019).



## 5    Conclusion

The NitroEurope DELTA® network has provided for the first time a comprehensive quality-assured multi-annual dataset on reactive gases ($NH_3$, $HNO_3$, $SO_2$, HCl) and aerosols ($NH_4^+$, $NO_3^-$, $SO_4^{2-}$, Cl⁻) across the major gradients of pollution, ecosystem type and climatic zones of Europe. The harmonised measurement approach of monthly time-integrated monitoring with a

simple low-cost DELTA® method represented an effective use of resources, making it possible to operate a network with a common measurement method across multiple laboratories at a large number of sites. At the same time, the concurrent measurement of the gas and aerosol components permitted an assessment of the atmospheric composition, spatial and seasonal characteristics in the gas and aerosol phase of these components. The dataset has also been used to develop estimates of site-based $N_r$ dry deposition fluxes across Europe, including supporting the development and validation of long-range transport

models. Combined with estimates of wet deposition (NEU bulk wet deposition network and data by other networks) to these sites, an assessment of the interactions between N supply and greenhouse gas exchange was addressed in a separate paper by Flechard et al. (2020), using $N_r$ and $CO_2$ flux data from the co-location of the NEU DELTA® with CarboEurope Integrated Project sites.

Two key features have emerged in the data. The first is the dominance of $NH_3$ as the largest single component at the majority of sites, with molar concentrations exceeding that of $HNO_3$ and $SO_2$, combined. Changes in the relative concentrations of these gases across Europe suggests that the deposition rates of $SO_2$ and $NH_3$ will increasingly be controlled by the molar ratio of $NH_3$ to combined acidity (sum of $SO_2$, $HNO_3$ and HCl) and deposition models should take these changes into account. As expected, the largest $NH_3$ concentrations were measured at cropland sites, in intensively managed agricultural areas dominated

by $NH_3$ emissions. The smallest concentrations were at remote semi-natural and forest sites, although concentrations in the Netherlands, Italy and Germany were up to 45 times larger than similarly classed sites in Finland, Norway and Sweden (< 0.6 µg $NH_3$-N m⁻³), illustrating the high $NH_3$ concentrations that sensitive habitats are exposed to in intensive agricultural landscapes in Europe.

Temporally, peak concentrations in $NH_3$ for crops and grassland sites occurred in spring, reflecting the implementation of the EU Nitrates Directive that prohibits winter manure spreading. The spring agriculture-related peak was seen even at semi-natural and forest sites, highlighting the influence of $NH_3$ emissions at sites that are more distant from sources. Summer peaks, promoted by increased volatilisation of $NH_3$, but also by gas-aerosol phase thermodynamics under warmer, drier conditions were seen in all ecosystem groups, except at Forest sites. The seasonality in the $NH_3$ concentrations captured for the different

groups is important, both for identifying periods when abatement might be targeted and for model development.

Seasonality in the other gas and aerosol components is also driven by changes in emission sources, chemical interactions and by changes in environmental conditions influencing partitioning between the precursor gases ($SO_2$, $HNO_3$, $NH_3$) and secondary aerosols ($SO_4^{2-}$, $NO_3^-$, $NH_4^+$). Seasonal cycles in $SO_2$ were mainly driven by emissions (combustion), with concentrations

peaking in winter, except in Southern Europe where the peak occurred in summer. $HNO_3$ concentrations were more complex, as affected by photochemistry, meteorology and by gas-aerosol phase equilibrium. Southern and eastern European regions provided the clearest seasonal cycle for $HNO_3$, with highest concentrations in summer and smallest in winter, attributed to increased photochemistry in the summer months in hotter climates. In comparison, a weaker seasonal cycle is seen in other regions, with marginally elevated concentrations in late winter, spring and summer and smallest in March and November.

Increased ozone in spring is likely to enhance oxidation of $NO_x$ to $HNO_3$ for forming the semi-volatile $NH_4NO_3$ by reaction with a surplus of $NH_3$. Cooler, wetter conditions in spring also favour the formation of $NH_4NO_3$ and more of the $NH_4NO_3$ remains in the aerosol or condensed phase. This accounts for the higher concentrations of $NH_4^+$ and $NO_3^-$ in spring and the



absence of a HNO$_3$ peak at this time of year. Conversely, increased partitioning to the gas phase in summer decreases NH$_4$NO$_3$ concentrations relative to gas phase NH$_3$ and HNO$_3$.

Particulate SO$_4^{2-}$ showed large peaks in concentrations in summer in Southern and also Eastern Europe, contrasting with much
smaller peaks occurring in early spring in other regions. The peaks in particulate SO$_4^{2-}$ coincided with peaks in NH$_3$ concentrations, illustrating the importance of NH$_3$ in driving the formation of (NH$_4$)$_2$SO$_4$. Since NH$_4$NO$_3$ is more abundant than (NH$_4$)$_2$SO$_4$, the seasonality of NH$_4^+$ is likely to be influenced more by the temperature and humidity dependence of the semi-volatile NH$_4$NO$_3$, than by the stable (NH$_4$)$_2$SO$_4$. This is supported by similarity in the the seasonal profiles of NH$_4^+$ and NO$_3^-$ at all sites, demonstrating temporal, as well as regional correlation between these two components.

The second key feature is the dominance of NH$_4$NO$_3$ over (NH$_4$)$_2$SO$_4$, with on average twice as much NO$_3^-$ as SO$_4^{2-}$ (on a molar basis). A change to an atmosphere that is more abundant in NH$_4$NO$_3$ will likely increase the atmospheric lifetimes and extend the footprint of the NH$_3$ and HNO$_3$ gases, due to the potential for the semi-volatile NH$_4$NO$_3$ to act as a reservoir and release NH$_3$ and HNO$_3$ in warm weather. The potential increase in atmospheric lifetime of NH$_3$ suggests that a larger fraction
of the reduced and oxidised N will remain in the gas phase as NH$_3$, resulting in a non-linearity in relationship between emissions and concentrations of NH$_3$. Ammonia is an important term in the calculation of total N dry deposition and a significant contributor to the exceedances of thresholds for atmospheric concentrations and deposition of N$_r$ components to sensitive habitats across much of Europe. In the DELTA® network, the Critical Levels of 1 and 3 µg NH$_3$ m$^{-3}$ for the protection of lichens-bryophytes and vegetation were exceeded at 62 % and 27 % of the sites, respectively. The importance of NH$_3$ is
therefore expected to further increase relative to oxidised N, as NO$_x$ emissions continue to decrease.

**Acknowledgements**

Research work under the NitroEurope (NEU) Integrated Project was carried out with funding from the European Union (Framework 6 Programme), together with supporting funds from NERC. Atmospheric measurements in the UK National Ammonia Monitoring Network (NAMN) and Acid Gas and Aerosol Monitoring Network (AGANet) are funded by the UK Department for Environment, Food and Rural Affairs (Defra) and devolved administrations. The Mediterranean Center for
Environmental Studies (CEAM) is partly supported by Generalitat Valenciana, Bancaja, and the Programm CONSOLIDER-INGENIO 2010 (GRACCIE). The authors gratefully acknowledge support and contributions by: 1) the large network of dedicated local site contacts, field teams and host organisations at NEU DELTA® and bulk wet deposition sites, 2) all chemical laboratory personnel involved in the sample preparations and chemical analyses from the chemical laboratories, 3) RIVM for hosting the DELTA-AMOR inter-comparsions at Vredepeel, and 4) Jan Vonk at RIVM for providing links to access NH$_3$ and
SO$_2$ data from the Dutch national network LML (Landelijk Meetnet Luchtkwaliteitl).

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



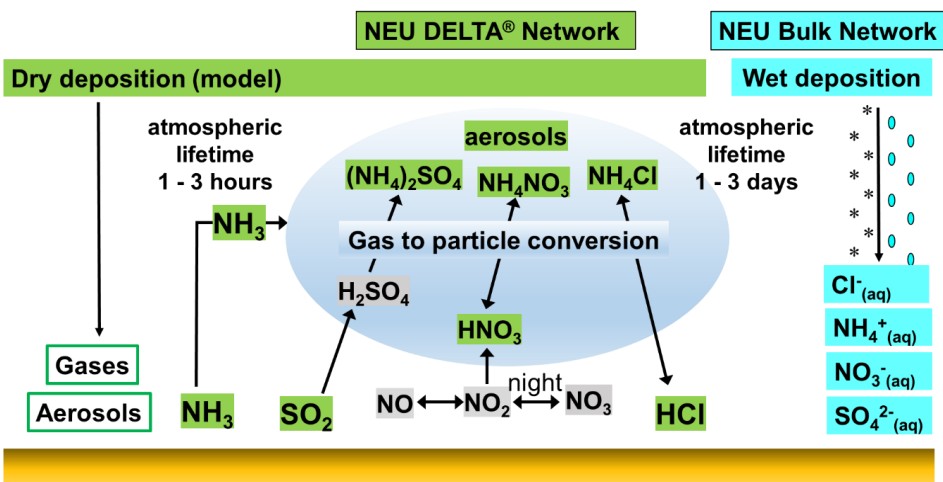

Figure 1: Reaction scheme for the formation of ammonium aerosols from interaction of NH₃ with acid gases HNO₃, SO₂ and HCl, showing the components (green) that were measured in NitroEurope (NEU) DELTA® network. Dry deposition of the gas and aerosol components was estimated by inferential modelling (Flechard et al., 2011), while wet deposition (blue) was measured in the NEU bulk wet deposition network at a subset of the DELTA® sites.







Figure 2: NitroEurope (NEU) DELTA® network sites operated between 2006 and 2010. The colour of the symbols indicates the responsible laboratories: CEAM (The Mediterranean Center for Environmental Studies), vTI (von Thunen Institut), INRAE (French National Research Institute for Agriculture, Food and Environment), MHSC (Meteorological and Hydrological Service of Croatia), UKCEH (UK Centre for Ecology & Hydrology), NILU (Norwegian Institute for Air Research), SHMU (Slovak Hydrometeorological Institute). Ecosystem types are C: Crops, G: Grassland, F: Forests and SN: short Semi-Natural (includes moorland, peatland, shrubland and unimproved/upland grassland). Replicated (P = parallel) DELTA measurements are made at 4 sites: SK04/SK04P; UK-AMo/UK-AMoP (NH$_3$/NH$_4^+$ only), UK-Bu/UK-BuP and FR-Fgs/FR-FgsP (NaCl coated denuders instead of K$_2$CO$_3$/glycerol in sample train).

| Nr | ID | Name / Ecosystem Type | | Nr | ID | Name / Ecosystem Type | | Nr | ID | Name / Ecosystem Type | | Nr | ID | Name / Ecosystem Type | |
|---|---|---|---|---|---|---|---|---|---|---|---|---|---|---|---|
| 1 | DE-Hai | Hainich | F | 18 | IT-MB | Monte Bondone | SN | 35 | DK-Ris | Risbyholm | C | 51P | SK-04P | Stara Lesna | G |
| 2 | DE-Wet | Wetzstein | F | 19 | IT-BCi | Borgo Cioffi | C | 36 | SE-Nor | Norunda | F | 52 | SK-06 | Starina | G |
| 3 | DE-Geb | Gebesee | C | 20 | FR-Hes | Hesse | F | 37 | SE-Sky | Skyttorp | F | 53 | SK-07 | Topolniky | G |
| 4 | DE-Tha | Tharandt | F | 21 | IE-Dri | Dripsey | G | 38 | BE-Bra | Braschaat | F | 54 | ES-ES1 | El Saler | F |
| 5 | DE-Gri | Grillenburg | G | 22 | FR-Gri | Grignon | C | 39 | BE-Vie | Vielsalm | F | 55 | ES-VDA | Vall de Aliñá | SN |
| 6 | DE-Kli | Klingenberg | C | 23 | FR-Fon | Fontainebleau | F | 40 | BE-Lon | Lonzee | C | 56 | ES-LMa | Las Majadas | F |
| 7 | DE-Hoe | Höglwald | F | 24 | FR-LBr | Le Bray | F | 41 | FI-Hyy | Hyytiälä | F | 57 | UK-AMo | Auchencorth | SN |
| 8 | PT-Mi1 | Mitra | F | 25 | FR-Lq2 | Laqueuille | G | 42 | IT-Ro2 | Roccarespampani | F | 57P | UK-AMoP | Auchencorth | SN |
| 9 | U-Pet | Petrodolinskoye | C | 26 | FR-Pue | Puechabon | F | 43 | NL-Ca1 | Cabauw | G | 58 | UK-Bu | Easter Bush | G |
| 10 | IT-Ren | Renon | F | 27 | UK-Gr | Griffin | F | 44 | NL-Hor | Horstermeer | SN | 58P | UK-BuP | Easter Bush | G |
| 11 | RU-Fyo | Fyodorovskoe bog | F | 28 | UK-ES | East Saltoun | C | 45 | NL-Spe | Speulder | F | 59 | DE-Meh | Mehrstedt | F |
| 12 | PT-Esp | Espirra | F | 29 | IE-Ca2 | Carlow | G | 46 | IT-Amp | Amplero | SN | 60 | FR-Fgs | Fougéres | F |
| 13 | CZ-BK1 | BKFORES | F | 30 | DK-Sor | Soroe | F | 47 | IT-Col | Collelongo | F | 60P | FR-FgsP | Fougéres | F |
| 14 | HU-Bug | Bugac | SN | 31 | FI-Sod | Sodankylä | F | 48 | IT-SRo | San Rossore | F | 61 | IE-Sol | Solohead | G |
| 15 | PL-Pol | Polwet | SN | 32 | FI-Kaa | Kaamanen | SN | 49 | IT-PoV | Po Valley Pavia | C | 62 | NO-Bir | Birkenes | F |
| 16 | CH-Oe1 | Oensingen | G | 33 | FI-Lom | Lompolojänkkä | SN | 50 | NL-Loo | Loobos | F | 63 | DK-Brj | Brandbjerg | SN |
| 17 | CH-Lae | Laegern | F | 34 | DK-Lva | Rimi | G | 51 | SK-04 | Stara Lesna | G | 64 | FR-Bil | Bilos | F |





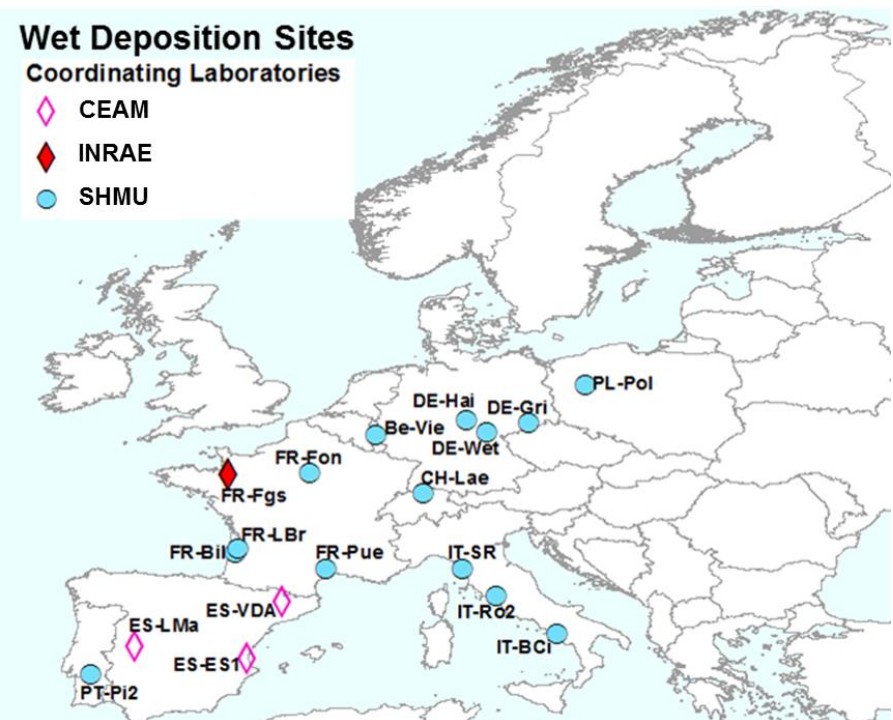

Figure 3: NitroEurope (NEU) Bulk wet deposition network sites operated between 2008 and 2010. The colour of the symbols indicates the responsible laboratories: CEAM (The Mediterranean Center for Environmental Studies), INRAE (French

5    National Research Institute for Agriculture, Food and Environment), and SHMU (Slovak Hydrometeorological Institute).





| Component | Reference solute concentration (mg L⁻¹) | ¹Equivalent gas concentration (µg m⁻³) | | ²Equivalent aerosol concentration (µg m⁻³) | | % of reported results within ± 10% of true value. Mean of all labs (range = min, max) | n |
|---|---|---|---|---|---|---|---|
| NH$_4^+$ | 0.1 - 0.9 | NH$_3$ | 0.02 - 0.17 | NH$_4^+$ | 0.03 - 0.24 | 68% (39 – 97 %) | 191 |
| | 1 | | 0.19 | | 0.27 | 90% (67 – 100 %) | 16 |
| NO$_3^-$ | 0.3 - 0.98 | HNO$_3$ | 0.06 - 0.2 | NO$_3^-$ | 0.08 - 0.26 | 85% (78 – 93%) | 197 |
| | 1 - 3 | | 0.2 - 0.6 | | 0.27 - 0.80 | 88% (81 – 96%) | 152 |
| SO$_4^{2-}$ | 0.5 - 0.8 | SO$_2$ | 0.07 - 0.11 | SO$_4^{2-}$ | 0.13 - 0.21 | 91% (83 – 100 %) | 199 |
| | 1 - 22 | | 0.13 - 2.9 | | 0.27 - 5.9 | 93% (85 – 100%) | 178 |
| Cl$^-$ | 0.07 - 0.8 | HCl | 0.01 - 0.16 | Cl$^-$ | 0.02 - 0.21 | 76% (48 – 93%) | 187 |
| | 1 - 10 | | 0.27 - 4.5 | | 0.27 - 5.9 | 96% (83 – 100%) | 45 |
| Ca$^{2+}$ | 0.07 - 0.6 | | | Ca$^{2+}$ | 0.02 - 0.16 | 36% (12 – 59%) | 176 |
| | 1 - 24 | | | | 0.27 - 6.4 | 80% (0 = 100 %) | 10 |
| Mg$^{2+}$ | 0.05 - 0.25 | | | Mg$^{2+}$ | 0.01 - 0.07 | 59% (22 – 75%) | 160 |
| | 1 - 5 | | | | 0.27 - 1.3 | 90% (50 – 100%) | 10 |
| Na$^+$ | 0.08 - 0.5 | | | Na$^+$ | 0.02 - 0.13 | 72% (46 – 85%) | 170 |
| | 1 - 52 | | | | 0.27 - 14 | 89% (60 – 100%) | 48 |

¹Equivalent gas concentrations, based on denuder extraction volumes of 3 mL (NH$_3$) and 5 mL (HNO$_3$, SO$_2$, HCl) and air volume of 15 m³ (typical volume of air sampled by DELTA® system over a month).

5  ²Equivalent aerosol concentrations, based on aerosol filter extraction volume of 4 mL (NH$_4^+$) and 5 mL (NO$_3^-$, SO$_4^{2-}$, Cl$^-$, Na$^+$, Ca$^{2+}$ and Mg$^{2+}$) and air volume of 15 m³ (typical volume of air sampled by DELTA® system over a month).

Figure 4: Summary of reported results from all laboratories in wet chemistry proficiency testing (PT) schemes for chemical

10  analysis of aqueous inorganic ions (2006 – 2010: EMEP, WMO-GAW and NitroEurope), expressed as a percentage deviation

from the true value (PT reference solutions). The grey shaded areas in the graphs show values that are within ± 10 % of true

value.





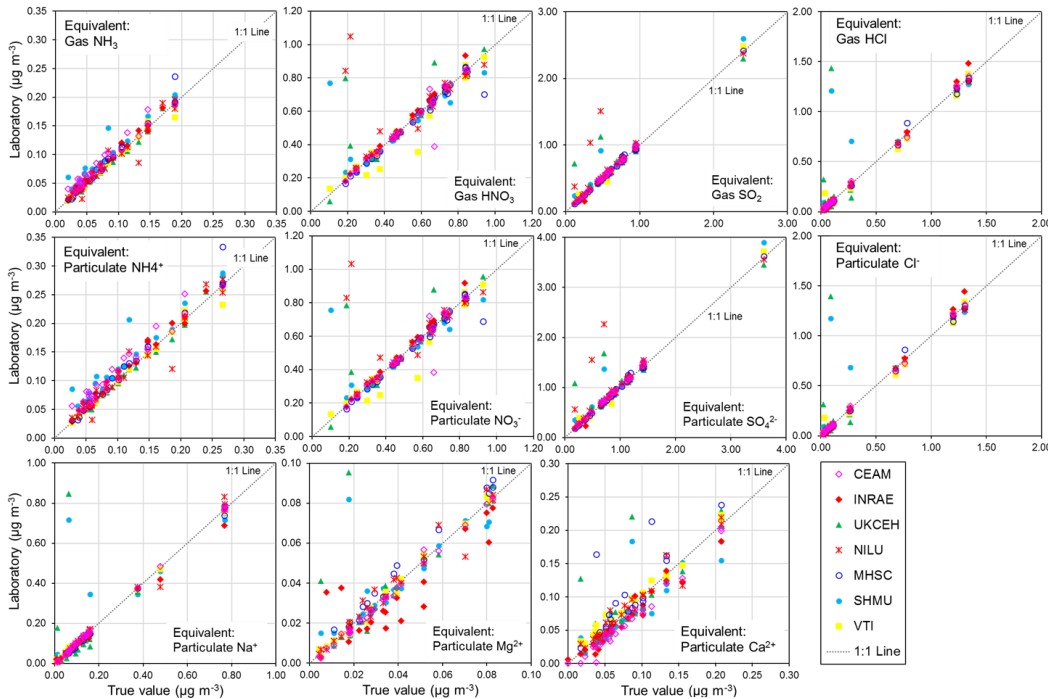

5    Figure 5: Scatter plots comparing all NEU laboratory reported results from wet chemistry proficiency testing (PT) schemes
(2006 – 2010: EMEP, WMO-GAW and NitroEurope) *vs* true values (PT reference solutions). All aqueous ion concentrations
(mg L$^{-1}$) from Figure 4 are converted to equivalent gas and aerosols concentrations (µg m$^{-3}$) for the comparisons.





| Lab | Gas: NH₃ | | | Gas: HNO₃ | | | Gas: SO₂ | | | Gas: HCl | | |
|---|---|---|---|---|---|---|---|---|---|---|---|---|
| | $R^2$ | slope | $n$ | $R^2$ | slope | $n$ | $R^2$ | slope | $n$ | $R^2$ | slope | $n$ |
| CEAM | 0.87 | 0.89 | 41 | 0.80 | 0.90 | 39 | 0.66 | 0.94 | 41 | 0.16 | 1.77 | 41 |
| INRAE | 0.99 | 1.00 | 8 | 0.99 | 0.99 | 8 | 0.88 | 1.25 | 7 | 0.02 | 1.73 | 8 |
| UKCEH | 0.99 | 1.00 | 42 | 0.96 | 1.10 | 42 | 0.92 | 0.96 | 42 | 0.43 | 0.52 | 42 |
| NILU | 0.92 | 1.17 | 30 | 0.96 | 0.93 | 30 | 0.91 | 0.95 | 30 | 0.08 | 0.70 | 4 |
| MHSC | 0.87 | 1.21 | 41 | 0.93 | 1.08 | 37 | 0.92 | 1.01 | 38 | 0.58 | 0.58 | 39 |
| SHMU | 0.96 | 1.0 | 38 | 0.98 | 1.0 | 37 | 0.62 | 0.88 | 39 | 0.62 | 1.37 | 39 |
| VTI | 0.92 | 0.91 | 42 | 0.94 | 0.88 | 42 | 0.91 | 1.08 | 42 | 0.87 | 0.96 | 42 |

| Lab | Particle: NH₄ | | | Particle: NO₃ | | | Particle: SO₄²⁻ | | | Particle: Cl | | |
|---|---|---|---|---|---|---|---|---|---|---|---|---|
| | $R^2$ | slope | $n$ | $R^2$ | slope | $n$ | $R^2$ | slope | $n$ | $R^2$ | slope | $n$ |
| CEAM | 0.22 | 0.42 | 41 | 0.96 | 1.03 | 41 | 0.89 | 1.20 | 41 | 0.54 | 1.01 | 40 |
| INRAE | 0.98 | 0.93 | 8 | 0.72 | 0.82 | 8 | 0.75 | 0.75 | 8 | 0.70 | 1.31 | 8 |
| UKCEH | 0.90 | 0.93 | 43 | 0.98 | 0.98 | 39 | 0.96 | 0.99 | 38 | 0.77 | 0.87 | 37 |
| NILU | 0.80 | 0.94 | 26 | 0.82 | 0.92 | 27 | 0.76 | 0.91 | 27 | - | 2.61 | 2 |
| MHSC | 0.80 | 1.26 | 40 | 0.93 | 1.02 | 41 | 0.78 | 0.89 | 39 | 0.80 | 0.85 | 39 |
| SHMU | 0.91 | 1.09 | 39 | 0.85 | 0.92 | 39 | 0.59 | 0.90 | 39 | 0.38 | 0.85 | 39 |
| VTI | 0.87 | 1.02 | 41 | 0.91 | 0.91 | 40 | 0.88 | 0.88 | 41 | 0.68 | 0.91 | 41 |

| Lab | Particle: Na⁺ | | | Particle: Ca²⁺ | | | Particle: Mg²⁺ | | | | | |
|---|---|---|---|---|---|---|---|---|---|---|---|---|
| | $R^2$ | slope | $n$ | $R^2$ | slope | $n$ | $R^2$ | slope | $n$ | | | |
| CEAM | 0.53 | 1.40 | 12 | 0.52 | 1.60 | 11 | 0.66 | 1.86 | 12 | | | |
| INRAE | 0.99 | 0.99 | 8 | 0.39 | 0.57 | 8 | 0.04 | 0.33 | 8 | | | |
| UKCEH | 0.82 | 0.95 | 38 | 0.77 | 0.92 | 38 | 0.86 | 1.05 | 40 | | | |
| NILU | 0.84 | 2.24 | 4 | 0.75 | 4.72 | 4 | 0.48 | 2.56 | 4 | | | |
| MHSC | 0.49 | 0.88 | 34 | 0.42 | 1.74 | 40 | 0.49 | 2.42 | 39 | | | |
| SHMU | 1.0 | 0.78 | 27 | 0.82 | 1.01 | 39 | 0.70 | 0.74 | 39 | | | |
| VTI | 0.82 | 1.0 | 41 | 0.75 | 0.88 | 37 | 0.84 | 0.95 | 41 | | | |

Figure 6: Scatter plots comparing atmospheric gas (NH₃, HNO₃, SO₂ and HCl) and aerosol (NH₄⁺, NO₃⁻, SO₄²⁻, Cl⁻, Na⁺, Ca²⁺, Mg²⁺) concentrations measured by each of the NEU laboratories with the median estimate of all laboratories. Data from all field inter-comparisons (2006 – 2009) for all test sites (Auchencorth-UK, Braunschweig-Gemany, Montelibretti-Italy and Paterna-Spain) are combined in the analysis. A summary of the regression results is shown in the table below the graphs. Note (i) there are fewer data points for INRAE because they joined the NEU network later in 2007 and participated in the 2008 and 2009 inter-comaprisons only, (ii) low number of observations in some cases were due to some laboratories not reporting all parameters. NILU: HCl, Cl⁻, Na⁺, Ca²⁺ and Mg²⁺ reported for 2008 inter-comparisons only; CEAM: Na⁺, Ca²⁺, Mg²⁺ reported for 2007-2009 inter-comparisons only.





Figure 7: (LEFT) Annual averaged gas and aerosol concentrations (2007 – 2010) of sites in the NEU DELTA® network, grouped according to ecosystem types: crops (*n* = 10), grassland (*n* = 9 + 1 parallel), semi-natural (*n* = 11 + 1 parallel) and forests (*n* = 34 + 2 parallel). (RIGHT) Percentage composition of gas and aerosol components measured at NEU DELTA® network sites (*n* = 64 + 4 parallel sites) (mean of all annual mean concentrations from 2007 to 2010). Years with < 7 months of data, including 2006, are excluded. Where the number of years contributing to the annual average is < 4, the number is shown in brackets beside the site data. $Ca^{2+}$ and $Mg^{2+}$ data are not included as these were mostly at or below limit of detection. Replicated DELTA measurements are made at 4 sites: FR-Fgs/FR-FgsP (NaCl instead of $K_2CO_3$/glycerol coated denuders - HCl not measured), SK04/SK04P; UK-Ebu/UK-EbuP and UK-AMo/UK-AMoP ($NH_3$/$NH_4^+$ only).


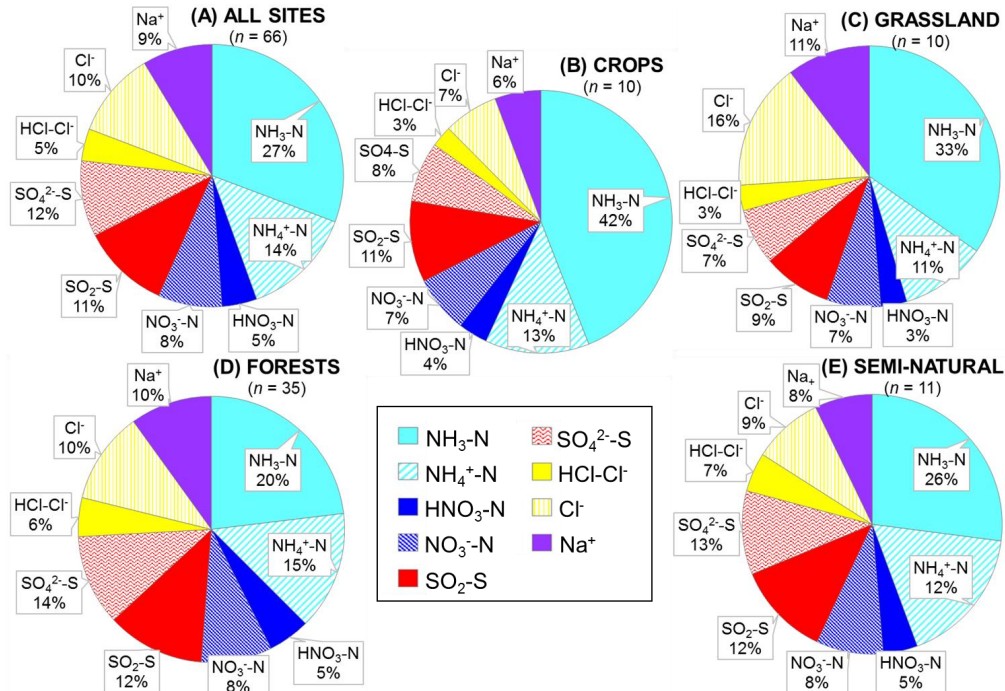

| Annual mean (µg m⁻³) | (A) ALL SITES (n = 66) % | | | (B) CROPS (n = 10) % | | | (C) GRASSLAND (n = 10) % | | | (D) FORESTS (n = 35) % | | | (E) SEMI-NATURAL (n = 11) % | | |
|---|---|---|---|---|---|---|---|---|---|---|---|---|---|---|---|
| | mean | min | max | mean | min | max | mean | min | max | mean | min | max | mean | min | max |
| NH₃-N | 27 | 6 | 56 | 42 | 24 | 56 | 33 | 18 | 47 | 20 | 6 | 46 | 26 | 7 | 39 |
| NH₄⁺-N | 14 | 6 | 23 | 13 | 7 | 21 | 11 | 6 | 18 | 15 | 9 | 23 | 12 | 6 | 20 |
| HNO₃-N | 5 | 1 | 9 | 4 | 2 | 5 | 3 | 1 | 7 | 5 | 3 | 9 | 5 | 1 | 8 |
| NO₃⁻-N | 8 | 0 | 15 | 7 | 4 | 13 | 7 | 3 | 9 | 8 | 1 | 13 | 8 | 0 | 15 |
| SO₂-S | 11 | 3 | 40 | 11 | 4 | 28 | 9 | 3 | 26 | 12 | 4 | 40 | 12 | 6 | 20 |
| SO₄²⁻-S | 12 | 3 | 31 | 8 | 3 | 12 | 7 | 4 | 13 | 14 | 5 | 31 | 13 | 5 | 26 |
| HCl-Cl⁻ | 5 | 1 | 21 | 3 | 1 | 3 | 3 | 1 | 5 | 6 | 2 | 16 | 7 | 1 | 21 |
| Cl⁻ | 10 | 2 | 29 | 7 | 3 | 17 | 16 | 3 | 28 | 10 | 2 | 26 | 9 | 2 | 29 |
| Na⁺ | 9 | 1 | 21 | 6 | 2 | 13 | 11 | 3 | 21 | 10 | 1 | 21 | 8 | 1 | 17 |
| Total | 100 | | | | 100 | | | | | | | | 100 | | |
| Sum Nᵣ | 54 | 24 | 80 | 66 | 54 | 80 | 54 | 41 | 73 | 49 | 24 | 80 | 51 | 24 | 67 |
| Sum Nᵣₑd | 41 | 17 | 70 | 55 | 41 | 70 | 44 | 29 | 59 | 35 | 17 | 64 | 38 | 19 | 56 |
| Sum Nₒₓ | 13 | 2 | 24 | 11 | 5 | 17 | 10 | 5 | 16 | 13 | 5 | 20 | 13 | 2 | 24 |
| Sum S | 23 | 7 | 53 | 18 | 11 | 36 | 16 | 7 | 35 | 26 | 11 | 53 | 25 | 15 | 38 |
| Sum (NH₄⁺-N + NO₃-N + SO₄²⁻-S) | 34 | 15 | 57 | 28 | 17 | 40 | 25 | 15 | 36 | 37 | 24 | 57 | 33 | 20 | 40 |
| Percentage contribution: by groups of components measured (by mass) | | | | | | | | | | | | | | | |
| Nᵣₑd / Nᵣ | 76 | 60 | 97 | 84 | 76 | 91 | 81 | 69 | 91 | 72 | 62 | 82 | 75 | 60 | 97 |
| NaCl / total aerosol | 20 | 4 | 45 | 12 | 6 | 27 | 27 | 6 | 43 | 20 | 4 | 42 | 17 | 4 | 45 |

Figure 8: (TOP) Pie charts showing the mean atmospheric composition of gas and aerosol components from annual averaged concentrations (µg m⁻³) measured at NEU DELTA® sites, for A) All sites (n = 66) and sites grouped according to ecosytem types, B) Crops (n = 10), C) Grassland (n = 10), D) Forests (n = 35) and E) Semi-natural (n = 11). UK-AmoP (parallel DELTA® at Auchencorth: NH₃/NH₄⁺ only) and FR-FgsP (parallel DELTA® at Fougéres: different sample train) were excluded in this analysis. (BOTTOM) Summary statistics on percentage composition by mass (µg m⁻³ element) measured. Sum Nᵣ = sum(NH₃-N + NH₄⁺-N + HNO₃-N + NO₃⁻-N), Sum S = sum (SO₂-S + SO₄²⁻-S), Nᵣₑd = sum reduced N (NH₃-N + NH₄⁺-N), Nₒₓ = sum oxidised N (HNO₃-N + NO₃⁻-N).

Figure 9: Comparisons of annual averaged gas and aerosol concentrations (2007 – 2010) of sites in the NEU DELTA® network, grouped by countries, with the respective 4-year averaged annual emission densities of gases (NH₃, NOₓ and SO₂) over the same period. Monitoring data from 3 national monitoring networks: *UK NAMN (NH₃ from 72 sites and NH₄⁺ from 30 sites; Tang et al., 2018a), *UK AGANet (raw uncorrected HNO₃, SO₂, HCl, NO₃⁻, SO₄²⁻, Cl⁻, Na⁺ from 30 sites; Tang et al. 2018b) and *NL-LML (NH₃ and SO₂ from 8 sites; van Zanten et al. 2017) are also included to illustrate the wider range of concentrations from larger numbers of sites. Error bars show the minimum and maximum concentrations measured in each country in the network.

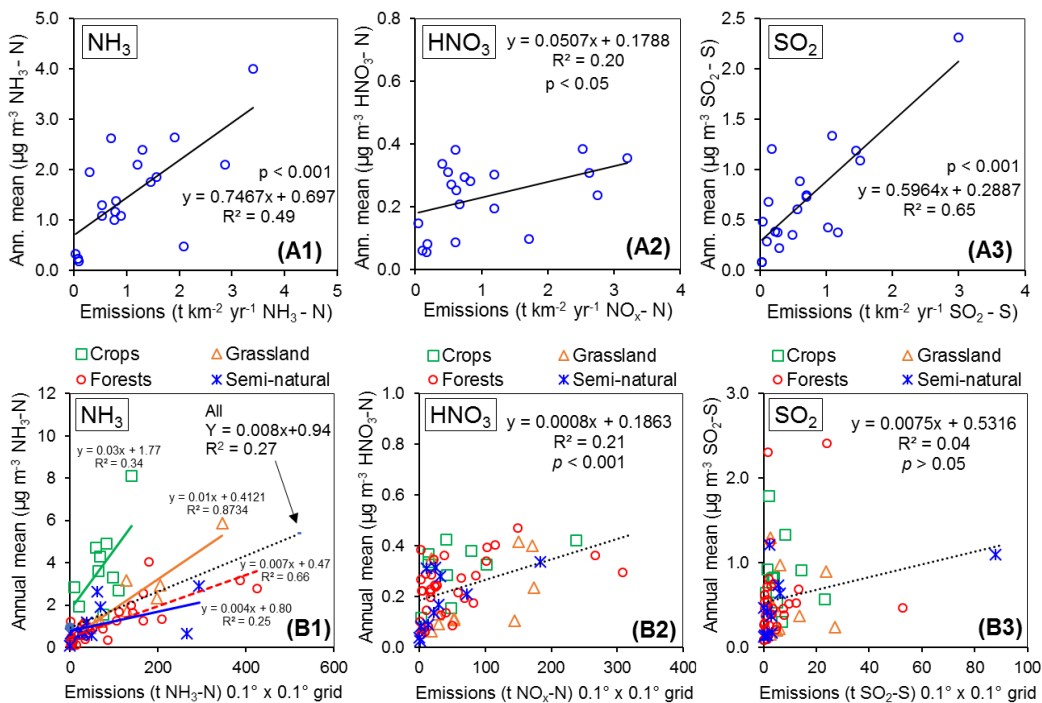

Figure 10: (A) Regression plots of national annual averaged gas ($NH_3$, $HNO_3$, $SO_2$) concentrations (2007 – 2010) *vs* 4-year national averaged emission densities of respective gases ($NH_3$, $NO_x$ and $SO_2$: tonnes $km^{-2}$ $yr^{-1}$) from each country over the same period ($n = 20$). (B) Regression plots of annual averaged gas ($NH_3$, $HNO_3$, $SO_2$) concentrations (2007 – 2010) at each site in the NEU DELTA® network *vs* 4-year averaged total emissions of gases ($NH_3$, $NO_x$ and $SO_2$: tonnes $yr^{-1}$) from single EMEP grids (0.1° x 0.1°) in which each site is located ($n = 66$). Coloured symbols indicate the ecosystem classification of each site (Crops, $n = 10$; Grassland, $n = 10$; Forests, $n = 35$ and Semi-natural, $n = 11$).

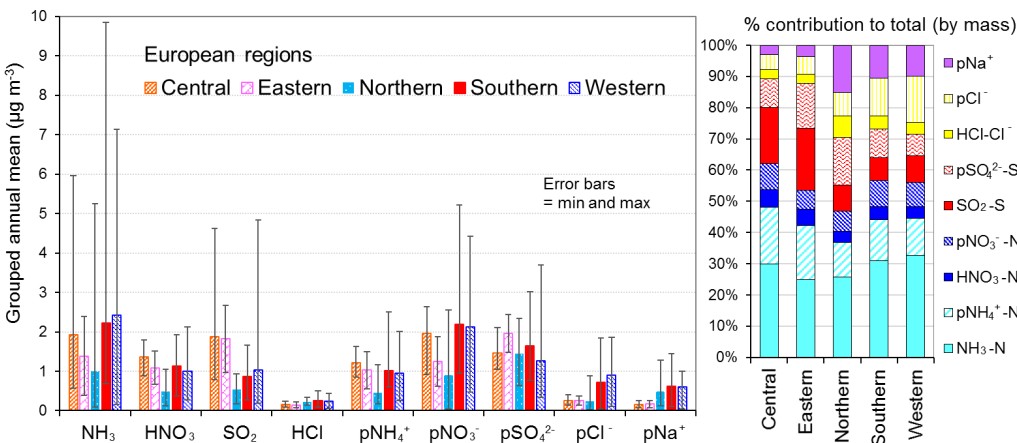

Figure 11: (LEFT) Spatial variation in annual averaged gas and aerosol concentrations (2007 to 2010) measured in the NEU

5    DELTA® network across Europe, grouped according to geographical distribution of the monitoring sites: Central ($n = 17$),

Eastern ($n = 2$), Northern ($n = 11$), Southern ($n = 12$) and Western ($n = 26$). p in front of component name denotes particulate.

(RIGHT) Percentage composition of gas and aerosol components according to European regions.



**(A) Gases: % contribution to total (sum of NH₃, HNO₃, SO₂, HCl) (nmol m⁻³)**

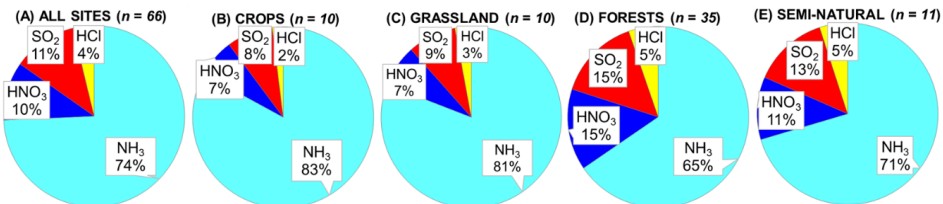

**(B) Aerosols: % contribution to total (sum of NH₄⁺, NO₃⁻, SO₄², Cl⁻) (nmol m⁻³)**

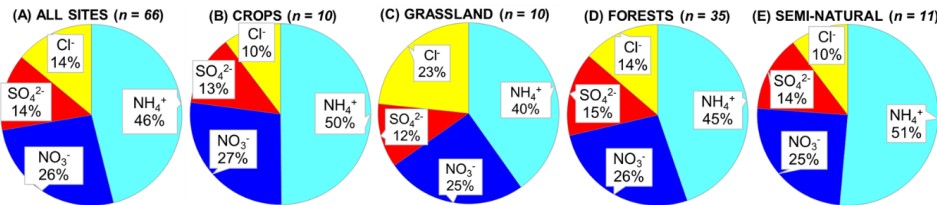

Figure 12: Pie charts of mean relative proportions of (TOP) Gases: $NH_3$, $HNO_3$, $SO_2$, HCl, and (BOTTOM) Aerosols: $NH_4^+$, $NO_3^-$, $SO_4^{2-}$, $Cl^-$. Data are annual averaged concentrations (nmol m⁻³) measured at NEU DELTA® sites, for (A) All sites ($n = 66$) and sites grouped according to ecosystem types,( B) Crops ($n = 10$), C) Grassland ($n = 10$), D) Forests ($n = 35$) and E) Semi-natural ($n = 11$). UK-AmoP (parallel DELTA® at Auchencorth: $NH_3/NH_4^+$ only) and FR-FgsP (parallel DELTA® at Fougéres: different sample train) were excluded in this analysis.



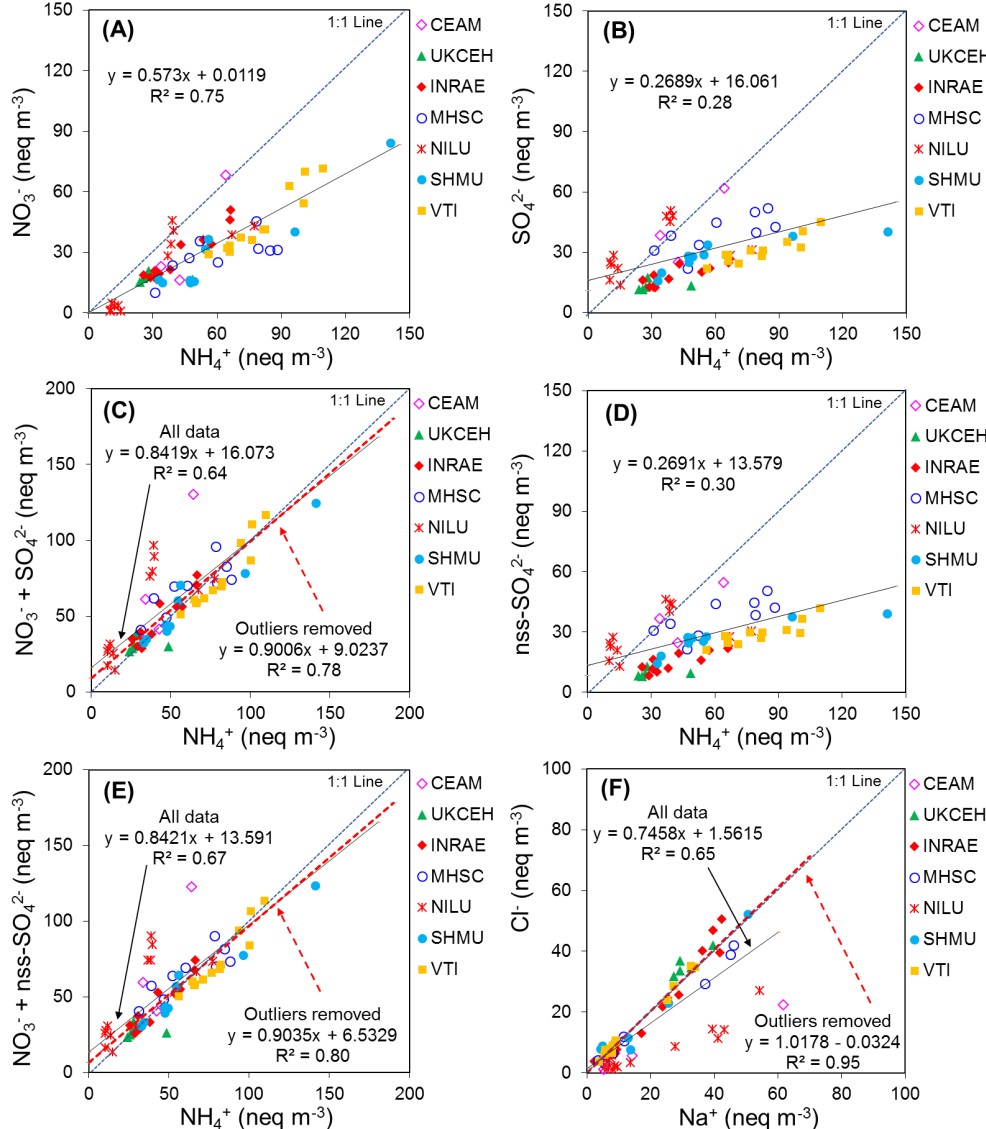

Figure 13: Regression plots between mean molar equivalent concentrations of (A) $NH_4^+$ and $NO_3^-$, (B) $NH_4^+$ and $SO_4^{2-}$, (C) $NH_4^+$ and sum($NO_3^- + SO_4^{2-}$), (D) $NH_4^+$ and nss-$SO_4^{2-}$, (E) $NH_4^+$ and sum($NO_3^- + $ nss-$SO_4^{2-}$) and (F) $Na^+$ and $Cl^-$, measured

5    in the NEU DELTA® network. Each data point represents the mean of all monthly measurements at each site, with different coloured symbols for each laboratory making the measurements. Outliers: where equivalent concentrations of $NH_4^+$:sum (anions) < 0.5 and Na:Cl > 2.



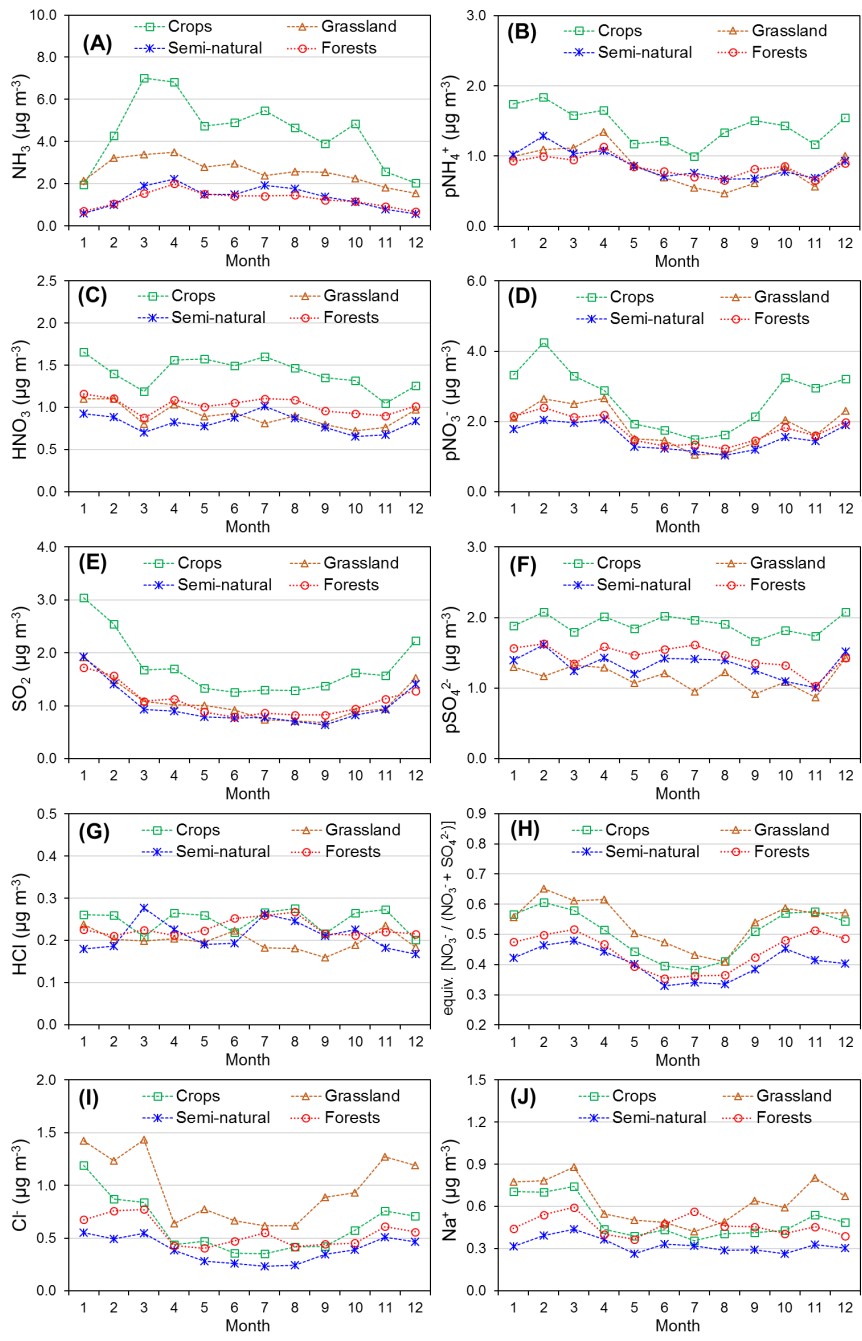

Figure 14: Seasonal variability in atmospheric gas (A) NH$_3$, (C) HNO$_3$, (E) SO$_2$, (G) HCl) and aerosol concentrations (B) pNH$_4^+$, (D) pNO$_3^-$, (F) pSO$_4^{2-}$, (I) pCl$^-$, (J) pNa$^+$ (p in front of component name denotes particulate). Each data point is the monthly averaged concentrations of grouped sites for the period 2006 to 2010, classified according to four ecosystem types: crops ($n = 10$), grassland ($n = 10$), semi-natural ($n = 11$) and forests ($n = 35$). Graph (H) shows the monthly mean ratio of molar equivalent (equiv.) concentrations of NO$_3^-$ to sum(NO$_3^-$ + SO$_4^2$). Month 1 = January and Month 12 = December.



Figure 15: Seasonal variability at sites grouped according to European regions in atmospheric gas (A) NH₃, (C) HNO₃, (E) SO₂, (G) HCl) and aerosol concentrations (B) pNH₄⁺, (D) pNO₃⁻, (F) pSO₄²⁻, (I) pCl⁻, (J) pNa⁺ (p in front of component name denotes particulate). Each data point is the monthly averaged concentrations of grouped sites for the period 2006 to 2010, classified according to five European regions: Central ($n = 17$), Eastern ($n = 2$), Northern ($n = 11$), Southern ($n = 12$) and Western ($n = 26$). Graph (H) shows the monthly mean ratio of molar equivalent (equiv.) concentrations of NO₃⁻ to sum(NO₃⁻ + SO₄²⁻). Month 1 = January and Month 12 = December.





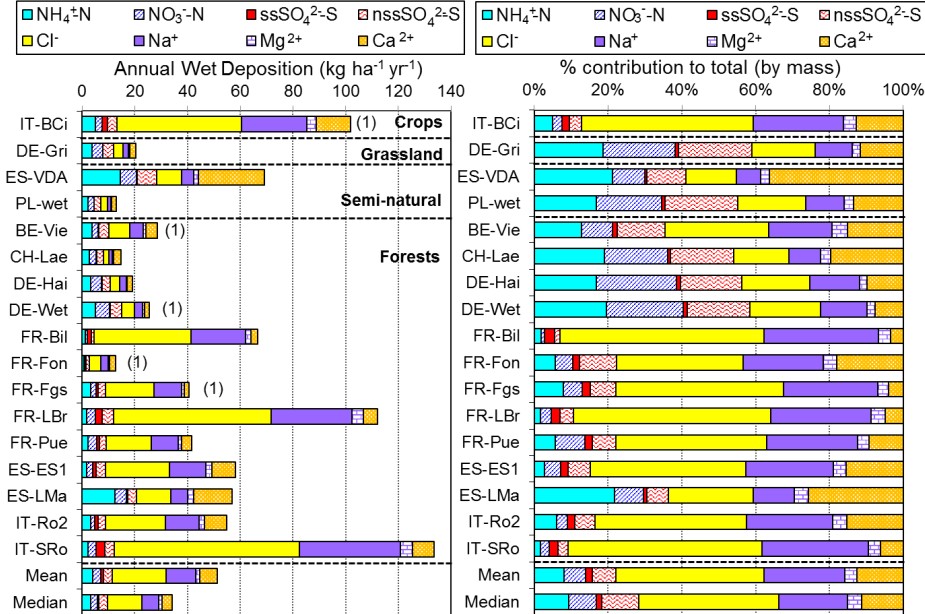

Figure 16: (LEFT) Annual wet deposition of inorganic components (kg ha⁻¹ yr⁻¹) estimated from Rotenkamp bulk precipitation collectors in the NEU bulk wet deposition network. (RIGHT) Percentage contribution of inorganic components to total (by mass) measured at 17 sites from 2008 to 2010. The data shown are 2-year averaged deposition, made between 2008 and 2010, except at 5 sites with 1 year of measurement only, as indicated in the graph in brackets.





Table 1: Details of annual NitroEurope (NEU) DELTA® field inter-comparisons conducted between 2006 and 2010.

| Inter-comparison period | Test sites | Participating laboratories | Number of monthly measurement periods |
|---|---|---|---|
| 2006 (Jul – Oct) | Auchencorth, UK<br>Braunschweig, Germany<br>Montelibretti, Italy<br>Paterna, Spain | 6 | 4 |
| 2007 (Jul – Aug) | Auchencorth, UK Montelibretti, Italy | 6 | 2 |
| 2008 (Apr – May) | Auchencorth, UK<br>Braunschweig, Germany | 7 (INRAE = new laboratory) | 2 |
| 2009 (Nov – Dec) | Auchencorth, UK<br>Montelibretti, Italy | 7 (INRAE = new laboratory) | 2 |





Table 2: Inter-comparison of results from 7 European laboratories at 4 different field test sites for all years (2006 – 2010). The results shown are the mean concentrations from each laboratory for each site and the averaged median estimates derived from all laboratories for each site.

| Site | Median (all years) | CEAM | % diff | CEH | % diff | MHSC | % diff | NILU | % diff | SHMU | % diff | VTI | % diff | *Median (2008/09) | *INRAE | *% diff |
|---|---|---|---|---|---|---|---|---|---|---|---|---|---|---|---|---|
| **NH₃** | | | | | | | | | | | | | | | | |
| Auchencorth | 1.42 | 1.23 | -13 | 1.39 | -2 | 1.51 | 6 | 1.60 | 13 | 1.48 | 4 | 1.38 | -2 | 1.06 | 1.17 | 10 |
| Braunschweig | 4.32 | 3.61 | -16 | 4.34 | 0 | 4.62 | 7 | 4.87 | 13 | 4.27 | -1 | 4.41 | 2 | 6.40 | 6.64 | 4 |
| Montelibretti | 2.46 | 1.66 | -33 | 2.44 | -1 | 2.89 | 18 | 2.77 | 12 | 2.63 | 7 | 2.34 | -5 | 1.91 | 1.91 | 0 |
| Paterna | 5.21 | 4.39 | -16 | 5.27 | 1 | 7.00 | 34 | 6.22 | 19 | 5.55 | 7 | 4.57 | -12 | | | |
| **NH₄⁺** | | | | | | | | | | | | | | | | |
| Auchencorth | 0.73 | 0.69 | -6 | 0.64 | -13 | 0.92 | 26 | 0.73 | 0 | 0.96 | 31 | 0.74 | 2 | 0.58 | 0.60 | 2 |
| Braunschweig | 1.55 | 1.54 | -1 | 1.61 | 4 | 2.15 | 39 | 1.18 | -24 | 1.64 | 6 | 1.45 | -6 | 1.38 | 1.31 | -5 |
| Montelibretti | 0.95 | 0.87 | -9 | 0.86 | -9 | 1.21 | 27 | 0.72 | -24 | 1.13 | 19 | 0.93 | -3 | 0.96 | 0.96 | 0 |
| Paterna | 1.80 | 0.50 | -72 | 1.56 | -13 | 2.12 | 18 | 1.64 | -9 | 2.04 | 13 | 2.26 | 25 | | | |
| **HNO₃** | | | | | | | | | | | | | | | | |
| Auchencorth | 0.57 | 0.57 | -1 | 0.53 | -7 | 0.69 | 21 | 0.62 | 9 | 0.59 | 3 | 0.49 | -15 | 0.55 | 0.59 | 7 |
| Braunschweig | 2.36 | 1.79 | -24 | 2.82 | 19 | 2.67 | 13 | 2.43 | 3 | 2.48 | 5 | 2.09 | -11 | 2.85 | 2.85 | 0 |
| Montelibretti | 2.64 | 2.53 | -4 | 2.74 | 4 | 3.08 | 17 | 2.60 | -2 | 2.77 | 5 | 2.31 | -13 | 1.70 | 1.70 | 0 |
| Paterna | 2.67 | 2.82 | 6 | 2.73 | 2 | 3.18 | 19 | 2.61 | -2 | 2.40 | -10 | 2.05 | -23 | | | |
| **NO₃⁻** | | | | | | | | | | | | | | | | |
| Auchencorth | 1.21 | 1.24 | 3 | 1.18 | -2 | 1.16 | -4 | 1.27 | 4 | 1.20 | -1 | 1.18 | -3 | 1.26 | 1.14 | -9 |
| Braunschweig | 3.26 | 3.70 | 14 | 3.43 | 5 | 3.33 | 2 | 2.28 | -30 | 3.09 | -5 | 2.36 | -28 | 2.92 | 2.94 | 1 |
| Montelibretti | 1.81 | 2.00 | 10 | 1.84 | 1 | 1.57 | -13 | 1.28 | -29 | 1.91 | 5 | 1.56 | -14 | 2.11 | 2.11 | 0 |
| Paterna | 4.52 | 4.73 | 5 | 4.34 | -4 | 4.60 | 2 | 4.34 | -4 | 4.57 | 1 | 4.32 | -4 | | | |
| **SO₂** | | | | | | | | | | | | | | | | |
| Auchencorth | 0.95 | 0.91 | -4 | 0.88 | -7 | 0.99 | 4 | 1.10 | 15 | 0.91 | -4 | 1.05 | 10 | 0.93 | 1.21 | 30 |
| Braunschweig | 1.49 | 1.33 | -11 | 1.49 | 0 | 1.65 | 10 | 1.32 | -12 | 1.41 | -5 | 1.45 | -3 | 1.05 | 1.17 | 11 |
| Montelibretti | 1.12 | 1.29 | 15 | 1.15 | 2 | 1.48 | 31 | 0.94 | -16 | 1.45 | 29 | 0.99 | -12 | 0.54 | 0.54 | 0 |
| Paterna | 1.96 | 2.07 | 6 | 1.96 | 0 | 2.04 | 4 | 1.93 | -2 | 1.99 | 2 | 1.78 | -9 | | | |
| **SO₄²⁻** | | | | | | | | | | | | | | | | |
| Auchencorth | 1.04 | 1.21 | 17 | 0.80 | -23 | 1.14 | 10 | 1.66 | 60 | 1.23 | 19 | 0.97 | -7 | 0.82 | 0.58 | -29 |
| Braunschweig | 2.04 | 2.67 | 31 | 2.12 | 4 | 2.35 | 15 | 1.58 | -22 | 1.72 | -16 | 1.51 | -26 | 1.61 | 1.37 | -15 |
| Montelibretti | 1.55 | 1.89 | 22 | 1.35 | -13 | 1.61 | 4 | 1.49 | -4 | 1.79 | 16 | 1.43 | -8 | 0.83 | 0.83 | 0 |
| Paterna | 3.28 | 4.19 | 28 | 3.06 | -7 | 3.06 | -7 | 3.68 | 12 | 3.01 | -8 | 3.21 | -2 | | | |
| **HCl** | | | | | | | | | | | | | | | | |
| Auchencorth | 0.20 | 1.01 | 396 | 0.19 | -9 | 0.15 | -28 | 0.21 | 4 | 0.33 | 62 | 0.19 | -6 | 0.22 | 0.74 | 244 |
| Braunschweig | 0.39 | 1.35 | 247 | 0.22 | -43 | 0.16 | -59 | 0.08 | -78 | 0.63 | 62 | 0.35 | -9 | 0.16 | 0.10 | -37 |
| Montelibretti | 0.40 | 1.01 | 151 | 0.33 | -18 | 0.40 | -1 | - | - | 0.58 | 45 | 0.36 | -11 | 0.54 | 0.54 | 0 |
| Paterna | 0.73 | 1.77 | 141 | 0.42 | -42 | 0.47 | -36 | - | - | 1.32 | 80 | 0.81 | 10 | | | |
| **Cl⁻** | | | | | | | | | | | | | | | | |
| Auchencorth | 0.84 | 0.93 | 10 | 0.73 | -13 | 0.86 | 3 | 0.26 | -69 | 1.17 | 39 | 0.85 | 1 | 0.95 | 0.81 | -15 |
| Braunschweig | 0.52 | 0.78 | 51 | 0.35 | -32 | 0.57 | 10 | - | - | 0.81 | 56 | 0.36 | -30 | 0.33 | 0.21 | -39 |
| Montelibretti | 0.85 | 0.94 | 11 | 0.76 | -11 | 0.84 | -1 | - | - | 1.19 | 41 | 0.86 | 1 | 0.66 | 0.66 | 0 |
| Paterna | 1.37 | 1.74 | 27 | 1.11 | -19 | 1.31 | -5 | - | - | 2.10 | 54 | 1.06 | -23 | | | |
| **Na⁺** | | | | | | | | | | | | | | | | |
| Auchencorth | 0.53 | 0.79 | 47 | 0.55 | 2 | 0.60 | 13 | 1.25 | 134 | 0.68 | 28 | 0.56 | 5 | 0.65 | 0.57 | -11 |
| Braunschweig | 0.37 | 0.38 | 4 | 0.21 | -43 | 0.37 | 1 | 0.24 | -34 | 0.85 | 131 | 0.37 | 1 | 0.27 | 0.19 | -29 |
| Montelibretti | 0.59 | 0.99 | 67 | 0.62 | 4 | 0.70 | 18 | - | - | 0.84 | 42 | 0.59 | -1 | 0.51 | 0.51 | 0 |
| Paterna | 0.94 | - | - | 1.01 | 7 | 0.71 | -25 | - | - | 0.94 | -1 | 0.95 | 1 | | | |
| **Ca²⁺** | | | | | | | | | | | | | | | | |
| Auchencorth | 0.06 | 0.06 | -5 | 0.06 | -11 | 0.32 | 415 | 0.15 | 137 | 0.05 | -27 | 0.06 | -12 | 0.03 | 0.04 | 38 |
| Braunschweig | 0.16 | 0.07 | -57 | 0.14 | -15 | 0.61 | 272 | 0.36 | 122 | 0.09 | -47 | 0.11 | -34 | 0.07 | 0.08 | 15 |
| Montelibretti | 0.16 | 0.54 | 241 | 0.16 | -1 | 0.45 | 183 | - | - | 0.15 | -4 | 0.16 | 2 | 0.08 | 0.08 | 0 |
| Paterna | 0.64 | - | - | 0.53 | -17 | 1.69 | 163 | - | - | 0.49 | -24 | 0.57 | -12 | | | |
| **Mg²⁺** | | | | | | | | | | | | | | | | |
| Auchencorth | 0.05 | 0.07 | 27 | 0.05 | -3 | 0.14 | 172 | 0.18 | 251 | 0.05 | -6 | 0.05 | -8 | 0.05 | 0.09 | 65 |
| Braunschweig | 0.05 | 0.03 | -33 | 0.04 | -26 | 0.10 | 114 | 0.08 | 61 | 0.03 | -35 | 0.02 | -56 | 0.02 | 0.04 | 77 |
| Montelibretti | 0.06 | 0.13 | 113 | 0.06 | -2 | 0.18 | 185 | - | - | 0.05 | -13 | 0.06 | 2 | 0.04 | 0.04 | 0 |
| Paterna | 0.13 | - | - | 0.13 | -4 | 0.33 | 147 | - | - | 0.10 | -24 | 0.13 | -2 | | | |





Table 3: Summary statistics of regression analyses between national annual averaged gas ($NH_3$, $HNO_3$, $SO_2$) and aerosol ($NH_4^+$, $NO_3^-$, $SO_4^{2-}$) concentrations, and national emission densities (4-year average for period 2007 to 2010, expressed as emissions per unit area of the country per year) for each of the 20 countries in the NEU DELTA® network.

| National annual average ($n = 20$) ($\mu g\ m^{-3}$) | National emission densities (20 countries) | | | | | | | | |
| | $NH_3$ (tonnes N $km^{-2}\ yr^{-1}$) | | | $NO_x$ (tonnes N $km^{-2}\ yr^{-1}$) | | | $SO_2$ (tonnes S $km^{-2}\ yr^{-1}$) | | |
| | slope | intercept | $R^2$ | slope | intercept | $R^2$ | slope | intercept | $R^2$ |
|---|---|---|---|---|---|---|---|---|---|
| Gas $NH_3$ - N | 0.75 | 0.70 | 0.49*** | 0.57 | 0.90 | 0.30* | 0.05 | 1.46 | $0.00^{ns}$ |
| Gas $HNO_3$ - N | 0.06 | 0.17 | 0.24* | 0.05 | 0.18 | 0.20* | 0.08 | 0.18 | 0.25* |
| Gas $SO_2$ - S | 0.17 | 0.52 | $0.24^{ns}$ | 0.22 | 0.46 | $0.16^{ns}$ | 0.60 | 0.29 | 0.65*** |
| Aerosol $NH_4$ - N | 0.23 | 0.50 | 0.36** | 0.19 | 0.54 | 0.27* | 0.20 | 0.61 | $0.16^{ns}$ |
| Aerosol $NO_3^-$ - N | 0.18 | 0.20 | 0.57*** | 0.15 | 0.23 | 0.44** | 0.08 | 0.33 | $0.07^{ns}$ |
| Aerosol $SO_4^{2-}$ - S | 0.06 | 0.47 | $0.07^{ns}$ | 0.07 | 0.45 | $0.12^{ns}$ | 0.12 | 0.44 | $0.18^{ns}$ |

Table 4: Annual averaged concentrations of gas and aerosol concentrations, measured at all sites and at grouped sites classified according to each of 4 ecosystem types in the NEU DELTA® network.

| NEU Network | Annual averaged concentrations ($\mu g\ m^{-3}$) (2007 – 2010) | | | | | | | | |
| | $NH_3$-N | $NH_4$-N | $HNO_3$-N | $pNO_3^-$-N | $SO_2$-S | $pSO_4^{2-}$-S | HCl-$Cl^-$ | $Cl^-$ | $Na^+$ |
|---|---|---|---|---|---|---|---|---|---|
| All sites ($n = 66$) | 1.63 | 0.73 | 0.23 | 0.42 | 0.58 | 0.48 | 0.22 | 0.57 | 0.46 |
| Crops ($n = 10$) | 3.81 | 1.11 | 0.32 | 0.61 | 0.87 | 0.63 | 0.24 | 0.58 | 0.49 |
| Grassland ($n = 10$) | 2.16 | 0.67 | 0.20 | 0.42 | 0.53 | 0.38 | 0.21 | 0.98 | 0.64 |
| Forest ($n = 35$) | 1.04 | 0.65 | 0.23 | 0.39 | 0.54 | 0.48 | 0.22 | 0.52 | 0.45 |
| Semi-natural ($n = 11$) | 1.11 | 0.70 | 0.18 | 0.35 | 0.50 | 0.43 | 0.22 | 0.37 | 0.30 |

15  Table 5: Regression correlations ($R^2$) between the mean molar concentrations (nmol $m^{-3}$) of gas and aerosol components at sites ($n = 66$) in the NEU DELTA® network.

| | $HNO_3$ | HCl | $SO_2$ | $NH_3$ | $NO_3^-$ | $Cl^-$ | 2 x $SO_4^{2-}$ | 2 x nss-$SO_4^{2-}$ | $NH_4^+$ | $Na^+$ |
|---|---|---|---|---|---|---|---|---|---|---|
| $HNO_3$ | 1 | | | | | | | | | |
| HCl | 0.13** | 1 | | | | | | | | |
| $SO_2$ | 0.46*** | $0.05^{ns}$ | 1 | | | | | | | |
| $NH_3$ | 0.28*** | 0.11** | 0.08* | 1 | | | | | | |
| $NO_3^-$ | 0.66*** | 0.21** | 0.19*** | 0.43*** | 1 | | | | | |
| $Cl^-$ | $0.00^{ns}$ | 0.22*** | $0.01^{ns}$ | 0.11** | 0.06* | 1 | | | | |
| 2 x $SO_4^{2-}$ | 0.34*** | 0.24*** | 0.33*** | 0.18*** | 0.39*** | $0.01^{ns}$ | 1 | | | |
| 2 x nss-$SO_4^{2-}$ | 0.35*** | 0.17*** | 0.36*** | 0.15** | 0.35*** | $0.04^{ns}$ | 0.98*** | 1 | | |
| $NH_4^+$ | 0.72*** | 0.06*** | 0.34*** | 0.43*** | 0.75*** | $0.00^{ns}$ | 0.28*** | 0.30*** | 1 | |
| $Na^+$ | $0.00^{ns}$ | 0.42*** | $0.00^{ns}$ | 0.10** | 0.13** | 0.65*** | 0.09* | $0.03^{ns}$ | $0.00^{ns}$ | 1 |

Significance level: * $p < 0.05$, ** $p < 0.01$, *** $p < 0.001$, $ns$ = non-significant ($p > 0.05$)



Table 6: Mean molar concentrations of gases and NH₃:acid gas ratios measured at sites ($n = 66$) in the NEU DELTA® network.

| All NEU sites | Molar concentrations (nmol m⁻³) | | | | | Ratios | | |
|---|---|---|---|---|---|---|---|---|
| | $NH_3$ | $HNO_3$ | $SO_2$ | $HCl$ | sum acids | $NH_3 : HNO_3$ | $NH_3 : SO_2$ | $NH_3$ : sum acids |
| mean | 115 | 16.5 | 18.3 | 6.4 | 41.1 | 7.5 | 7.7 | 2.9 |
| min | 5.4 | 2.0 | 2.5 | 1.6 | 10.9 | 0.8 | 0.5 | 0.3 |
| max | 566 | 33.8 | 78.2 | 13.4 | 122 | 34 | 33 | 13 |
| SD | 108 | 8.4 | 14.7 | 2.8 | 22.4 | 7.2 | 6.6 | 2.6 |
| $n$ | 66 | 66 | 66 | 66 | 66 | 66 | 66 | 66 |

5  Table 7: Linear regressions between the mean molar equivalent concentrations of aerosol components (neq m⁻³) at sites ($n = 66$) in the NEU DELTA® network.

| Linear Regression | Mean molar equivalent concentrations (neq m⁻³) | | | | | | |
|---|---|---|---|---|---|---|---|
| | $NH_4^+$ vs $NO_3^-$ | $NH_4^+$ vs $SO_4^{2-}$ | $NH_4^+$ vs sum ($NO_3^-$ + $SO_4^{2-}$) | $Na^+$ vs nss-$SO_4^{2-}$ | $NH_4^+$ vs sum ($NO_3^-$ + nss-$SO_4^{2-}$) | $Na^+$ vs $Cl^-$ (all data) | $Na^+$ vs $Cl^-$ (outliers excluded) |
| $R^2$ | 0.75*** | 0.28*** | 0.64*** | 0.30*** | 0.67*** | 0.65*** | 0.95*** |
| slope | 0.57*** | 0.27*** | 0.84ns | 0.27*** | 0.84* | 0.75*** | 1.01ns |
| intercept | 0.01ns | 16.1*** | 16.1** | 13.6*** | 13.6** | 1.56ns | -0.05ns |
| No. of sites: $n$ | 66 | 66 | 66 | 66 | 66 | 66 | 50 |

Significance level: * $p < 0.05$, ** $p < 0.01$, *** $p < 0.001$, $ns$ = non-significant ($p > 0.05$)

10  Table 8: Mean molar concentrations of aerosols and ratios measured at sites ($n = 66$) in the NEU DELTA® network.

| All NEU sites | Molar concentrations (nmol m⁻³) | | | | Ratios | | | |
|---|---|---|---|---|---|---|---|---|
| | $NH_4^+$ | $NO_3^-$ | $SO_4^{2-}$ | nss-$SO_4^{2-}$ | $NH_4^+ : NO_3^-$ | $NH_4^+ : 2xSO_4^{2-}$ | $NH_4^+ : 2xnss-SO_4^{2-}$ | $NH_4^+ : (NO_3^- + 2xSO_4^{2-})$ |
| mean | 52.8 | 30.2 | 15.1 | 13.9 | 2.4 | 1.8 | 2.1 | 0.9 |
| min | 10.1 | 0.7 | 5.8 | 4 | 0.9 | 0.4 | 0.4 | 0.4 |
| max | 141 | 84.3 | 38.4 | 35.8 | 21 | 3.6 | 5.1 | 1.6 |
| SD | 27.6 | 18.2 | 7.0 | 6.8 | 2.7 | 0.8 | 0.9 | 0.3 |
| $n$ | 66 | 66 | 66 | 66 | 66 | 66 | 66 | 66 |