# Peer review of "Pan-European rural monitoring network shows dominance of NH3 gas and NH4NO3 aerosol in inorganic atmospheric pollution load"

_Atmospheric Chemistry and Physics, 2020_

## Referee Comment (RC1) · Martijn Schaap (Referee) · 8 Jul 2020

Understanding the budgets of sulfur and nitrogen compounds and how they interact by e.g. inorganic aerosol formation is of key importance. The NEU network provides an outstanding contribution as it provides a comprehensive and quality controlled dataset across many countries. This paper clearly shows the large efforts required to set-up and run such a large monitoring network. Hence, although the dataset is from some time ago, it should be published and I recommend to publish the paper with a number of revisions.

My main concern is that the paper is quite long. I have the feeling that some features

which are now presented at different locations could be merged to guide the reader. One of these is the message that ammonium nitrate dominates above ammonium sulfate which is concluded from the correlation between components, ion balance, seasonality, etc. I would appreciate if the authors could try to focus the results section into a more integrative storyline than the stepwise approach chosen now.

Two parts I feel are less important for the paper are the following:

1. Concentration to Country emission correlation: The short life time of ammonia and NOx cause substantial gradients within larger countries. For that reason I would argue that the correlation between country emissions and averaged concentration levels is not saying a lot. Figure 9 presents these data and is hardly discussed in the paper. The emission density in the surroundings cells to me sounds more appropriate and tells something about the representativeness of the stations for the different pollutants.

2. Section 4: This section is hardly connected to the monitoring network results. I would rather see a discussion on the future of this network. Should it be continued? Adapted? Or?

The two main findings presented in the conclusions section are not new, and a few references to earlier works could be provided.

Content wise, I have the feeling that the role of chloride depletion reactions of sea salt are interpreted as outliers in the interpretation of data, see below.

As a modeler I would be very eager to compare our model results to the dataset and hope that the data will be openly available.

Individual remarks:

Title: I would recommend to move the word "atmospheric" to in "inorganic atmospheric pollution"

Line 7: Vieno reference is a bit strange here – not a monitoring work

Line 9: the negative impacts . . . should not be a new paragraph. The first two paragraphs contain two sentences now.

Section 2.2.1 page 7 line 25: Could you indicate the breakthrough estimation is in comparison to ammonium aerosol levels, especially for the agricultural sites.

Page 8, line 13: please refer forward to the results section on the impact of the NaCL denuders.

Section 2.6: Some countries may have large shipping contributions to NOx and SO2, how did you treat these in the indicator used here? Why did you choose 4 grids around a station and not the nine around and including the grid cell with the station?

Section 3.3.1 page 14 Line 1-3 details on the dry deposition schemes seem out of place here.

Page 14 line 36-43: The comparison between N and S is based on mass here. Given the scope on ecosystem deposition provided elsewhere I could imagine that a comparison based on acid equivalents makes more sense than the mass. I do not see the consistency between the currently higher N levels and emission reductions since the nineties as the emissions did not start from a ratio of 1:1.

Page 15, line 16. The 10-50% contributions in Putaud et al refer to the ammonium salts, not only ammonium. Please correct. Moreover, this paragraph seems more appropriate in the discussion or implication section than in the results chapter.

Page 16 line 3. Here the correlation between precursor and aerosol is discussed in the paragraph on the correlation with emission densities. Right place?

Page 17 line 3. HNO3 maybe highest in eastern Europe, but NOx emissions aren't. Could it be that the lower ammonia and hotter summer climate plays a pronounced role in the explanation as indicated in the seasonal cycle with summer maxima in the region (in contrast to western Europe). Similarly, in the presentation of the oxidized nitrogen on page 20 (L 24) the limitation on ammonia availability could be mentioned.

The higher correlation between nitrate and ammonia emissions is indicative for this issue as well. Ammonium nitrate formation could be checked with the ammonium salt ion balance. Often inverse relationships between nitric acid and ammonia are modelled due to the limiting impact of the equilibrium with ammonium nitrate. Do you see this feature in the data?

Page 21, the current levels are interpreted in relation to emission reductions which are not indicated from this network. The SO2 to SO4 ratio variability across the network may be the most interesting feature concerning sulfur for model developers. Did you see the anticipated systematic behavior for this ratio?

Page 21 the Bugac discussion interrupts the main information flow.

Page 22. The ion balance for southern Scandinavia may be affected by sodium nitrate formation and not so much by an overestimation of SO4. Na:CL depletion ratio may give a hint here. Further down on the same page the remark is made but no connection is made.

On page 24 another check is made on ion balances with hard statements on lab quality– are these issues not connected and is one actually looking at sea salt depletion reactions?

Page 22: does the HCl distribution provide a hint at the importance of the marine source for it?

Page 27 line 1-5: the impact of ammonia and temperature on seasonality of nitric acid is not discussed ad should be mentioned. OK, it is done in the next paragraph. Why not combine these?
* * *

---

## Referee Comment (RC2) · Anonymous Referee #1 · 29 Jul 2020

This manuscript describes measurements collected within the EU NitroEurope (NEU) network during the period 2006 – 2010. While some of this data has been previously published, as noted by the authors, the current manuscript provides a comprehensive description of the data quality as well as temporal and spatial patterns of atmospheric chemistry over the lifetime of the network. The data will make a valuable contribution to the field of atmospheric chemistry, in particular with respect to better understanding the role of reduced forms of reactive nitrogen in aerosol processes and for model evaluations. The manuscript is generally well written and the analyses are appropriate, though the manuscript is somewhat lengthy. I recommend publication subject to treatment of the relatively minor comments outlined below, some of which are technical in

[Figure]

nature and others seek to reduce the length of the paper.

Page 5 – Replicate measurements. It would be useful to see a bit more detail on the replicated measurements to get a better sense of overall precision. For example, scatterplots and summary statistics could be added to the Supplemental Material.

Page 6 – Coordinating laboratories. Some brief discussion of the analytical methods employed by the laboratories (ion chromatography or colorimetry) should be included, along with some discussion or reference to method detection limits (MDL). This information could also be included in the Supplemental.

Page 8 – Bulk precipitation measurements. These measurements will no doubt be useful for deposition assessments. However, as currently written the data do not add much to the current manuscript and could be removed to reduce overall length.

Page 11 – Line 20. The comparison is referred to here as "field inter-comparison" but as "laboratory inter-comparison" in the 3.2 section heading. I understand the distinction, but it is a little confusing at first glance.

Page 12 – Line 14. Knowledge of the laboratory blanks would be very helpful. Is there no way to recover the results from original chromatograms? Granted it might be time consuming but interlaboratory comparison of blanks, particularly for NH3 which is notoriously difficult, could be enlightening as to some of the laboratory comparisons.

Page 12 – Line 23. Was CEAM the only laboratory that used colorimetric analysis for NH4+? See previous comment on summarizing analytical techniques used by the various laboratories.

Page 15 – Line 4. The overestimation of HNO3, or at least that the HNO3 measurement includes other oxidized N compounds, could be noted again here.

Page 15 – Line 30. See previous comment regarding LOD/MDL for different laboratories/chemical species.

Page 15 – Line 34. I am unsure of the point of the comparisons between air concentrations and emissions, which is not motivated by the description of the NitroEurope project or in the description of the specific objectives of the manuscript. I think this analysis could be removed from the paper without any implication for the main points or conclusions. But if it is to remain, the purpose of the analysis should be clearly stated and it should be shortened where possible., e.g. only including the comparisons to gridded emissions.

Page 17 – Line 25. Were the high concentrations at IT-BCi indicative of highly local emissions, i.e., adjacent to the field site, or is this concentration more indicative of a broader area? It would be impractical to include a description of every site but where such details are relevant, they should be included. In the same regard, it would be good to know if all of the grassland sites are grazed (Page 17 – Line 41). It appears so.

Page 18 – Line 38/39. I believe "will dominate dry NH3-N dry deposition" should be changed to "will dominate dry N deposition", correct?

Page 19 – Line 11. Remove "that are".

Page 19 – Line 12. Change "emission" to "emissions"

Page 21 – Line 7. The sentence beginning "This corroborates…." is quite lengthy.

Page 23 – Section 3.4. It appears that Figure 13 is incorrectly referred to as Figure 12 throughout this section.

Page 24 – Line 16. Are there other potential reasons for the higher sulfate measurements at these sites? Seems worthy of additional investigation/discussion.

Page 24 – Line 16. Regarding the discussion of the CEAM and NILU Na+/Cl- regressions and the data below the 1:1 line, there does seem to be correlation among these outliers. Could this be an issue in the way filter blanks were applied? Perhaps an average Cl- blank biased high by an outlier was subtracted from all of the field measurements?

Page 26 – Line 19. Should "Figure 13A" be "Figure 14A"?

Page 26 – Line 23. "…with possible uptake and removal of NH3 from the atmosphere". Could results from the GRAMINAE project be cited here?

Page 26 – Line 24. Please change "thermodynamic shift to" to "thermodynamic shift of NH4NO3 to".

Page 26 – Section 3.5.2. To what extent could the temporal patterns in HNO3 be confounded by the collection of other oxidized N species on the denuder?

Page 28 – Line 12. Consider changing "were provided by" to "were observed at".

Page 29 – Section 3.6. See previous comment regarding inclusion of precipitation measurements

Page 29 – Section 4.0. It seems like the material in this section could be greatly condensed and integrated into the Conclusions.

Page 30 – Line 10. The sentence beginning "However, SO2 (by mass)……." is quite lengthy.

Page 32 – Line 11. Some additional concluding comments, building on this key feature of the analysis, would be welcomed. For example, what does this shift from a sulfate dominated to nitrate dominated inorganic aerosol regime suggest for future European monitoring needs in support of ecological and human health protection? What else can be gleaned from the current study, with respect to data quality, methods, and ability to resolve spatial and temporal patterns, that can inform future monitoring efforts?

---

## Author Response (AR1)

**RESPONSE TO REVIEWER 1**

Martijn Schaap (Referee)

The authors thank Dr. Schaap for his supportive comments for publication and for taking the time to look at all the details

10 described in the manuscript. We have carefully considered all comments. Please refer to the specific responses.

*Understanding the budgets of sulfur and nitrogen compounds and how they interact by e.g. inorganic aerosol formation is of key importance. The NEU network provides an outstanding contribution as it provides a comprehensive and quality controlled dataset across many countries. This paper clearly shows the large efforts required to set-up and run such a large monitoring network. Hence, although the dataset is from some time ago, it should be published and I recommend to publish the paper with*

15 *a number of revisions.*

*My main concern is that the paper is quite long. I have the feeling that some features which are now presented at different locations could be merged to guide the reader. One of these is the message that ammonium nitrate dominates above ammonium sulfate which is concluded from the correlation between components, ion balance, seasonality, etc. I would appreciate if the authors could try to focus the results section into a more integrative storyline than the stepwise approach chosen now.*

20 **Author response**

Whilst we acknowledge that our paper brings together and interprets a large body of network measurements, we nevertheless believe that it has a coherent flow that guides the reader through the material that is presented.

- Quality assessment of data: laboratory and field intercomparisons
- Spatial variability – with comparisons against national emissions densities (to demonstrate correlation of

25 concentrations with emissions), according to sites grouped by land-use types and geographical regions of Europe, and which required examination of spatial correlation,
- Seasonal variability - according to sites grouped by land-use types and geographical regions of Europe.
- Absolute and relative concentrations of the different inorganic components are also investigated, as well as their spatial and temporal variations

30 - Bulk wet deposition composition.
- A comprehensive final section of key conclusions.

*Two parts I feel are less important for the paper are the following:*

1. *Concentration to Country emission correlation: The short life time of ammonia and NOx cause substantial gradients within larger countries. For that reason I would argue that the correlation between country emissions*

35 *and averaged concentration levels is not saying a lot. Figure 9 presents these data and is hardly discussed in the paper. The emission density in the surroundings cells to me sounds more appropriate and tells something about the representativeness of the stations for the different pollutants.*

**Author response**

A similar comment was posted by Reviewer 2 "*Page 15 – Line 34. I am unsure of the point of the comparisons between*

40 *air concentrations and emissions, which is not motivated by the description of the NitroEurope project or in the description of the specific objectives of the manuscript. I think this analysis could be removed from the paper without any implication for the main points or conclusions. But if it is to remain, the purpose of the analysis should be clearly stated and it should be shortened where possible., e.g. only including the comparisons to gridded emissions*".

45 **Response (as also provided in response to reviewer 2):**

**Sect. 3.3.3. Comparison with gridded emissions**: Deleted and moved to supplementary materials.

**Sect 3.3.2. Comparisons with national gas emissions**: Retained

Additional supporting text added at the end of section 3.3.2 (see below):

"The comparisons here used national emission totals, where emissions have been summed and averaged across very large and heterogeneous areas in each country. Additional analysis were also undertaken to compare the individual site mean data with i) gridded emissions from individual 0.1° x 0.1° EMEP grids in which the NEU sites are located (Supp. Figure S8, S9), and ii) averaged emissions of an extended number of EMEP grids (4 x grids) closest to the site (Supp. Figure S10). Since results from these analysis were similar to the comparisons with national emission densities, they are not included for further discussions in this paper. The purpose of the ranked emission densities is to compare the pollution climate in terms of primary gas emissions ($SO_2$, $NO_2$, $NH_3$) across the 20 European countries and to see if this is matched by the DELTA measurements. Despite the complex relationship between emissions and concentrations, the pollution gradient in Europe is clearly captured by the present data. At the same time, it also demonstrated the potential application of the DELTA® approach in providing national concentration fields, as evidence to compare against spatial and long-term trends in the national emissions data."

Page 15, lines 38 – 39

The lines below has been deleted, as the details are provided in the Figure 9 caption already.

"The error bars, where shown, is the range (min and max) of annual averaged concentrations of sites in each country"

Page 15, lines 39 – 40

The lines below has been moved to Figure 9 caption

"Where error bars are not visible, this indicates either that the country has measurement from just one site, or the range of concentrations measured are very close to the average."

.

*2. Section 4: This section is hardly connected to the monitoring network results. I would rather see a discussion on the future of this network. Should it be continued? Adapted? Or? The two main findings presented in the conclusions section are not new, and a few references to earlier works could be provided. Content wise, I have the feeling that the role of chloride depletion reactions of sea salt are interpreted as outliers in the interpretation of data, see below. As a modeler I would be very eager to compare our model results to the dataset and hope that the data will be openly available.*

Reviewer 2 posted a couple of similar comments *"Page 29 – Section 4.0. It seems like the material in this section could be greatly condensed and integrated into the Conclusions."* And *"Page 32 – Line 11. Some additional concluding comments, building on this key feature of the analysis, would be welcomed. For example, what does this shift from a sulfate dominated to nitrate dominated inorganic aerosol regime suggest for future European monitoring needs in support of ecological and human health protection? What else can be gleaned from the current study, with respect to data quality, methods, and ability to resolve spatial and temporal patterns, that can inform future monitoring efforts?"*

**Author Response:**
**See revised text below which addresses both reviewers' comments:**
(Please note Section 4.0 has been removed and integrated into the Conclusions)

[revised manuscript text omitted]

*Individual remarks:*

*3. Title: I would recommend to move the word "atmospheric" to in "inorganic atmospheric pollution"*

**Author Response:**

Thank you.

"Pan-European rural atmospheric monitoring network shows dominance of $NH_3$ gas and $NH_4NO_3$ aerosol in inorganic pollution load"

Amended to:

"Pan-European rural monitoring network shows dominance of $NH_3$ gas and $NH_4NO_3$ aerosol in inorganic atmospheric pollution load"

*4. Line 7: Vieno reference is a bit strange here – not a monitoring work*

**Author Response:**

The sentence in question copied below:

*"The aerosols, formed through neutralisation reactions between the alkaline NH3 gas and acids generated in the atmosphere by the oxidation of $SO_2$ and $NO_x$ (Huntzicker et al., 1980; AQEG, 2012) are a major component of fine particulate matter ($PM_{2.5}$) (AQEG, 2012; Vieno et al., 2016a) and precipitation (ROTAP, 2012; EMEP, 2019)."*

The modelling work by Vieno et al. looked at the sensitivity of annual-average surface concentrations of $PM_{2.5}$ across the UK to reductions in UK terrestrial anthropogenic emissions in primary $PM_{2.5}$, $NH_3$, $NO_x$, $SO_x$ and non-methane VOC. The work shows that the reactions between $NH_3$, $SO_x$ and $NO_x$ are major contributors to $PM_{2.5}$.

We feel it is a relevant and important reference to cite here.

*5. Line 9: the negative impacts… should not be a new paragraph. The first two paragraphs contain two sentences now.*

**Author Response:**

The first two paragraphs have been merged into a single paragraph.

*6. Section 2.2.1 page 7 line 25: Could you indicate the breakthrough estimation is in comparison to ammonium aerosol levels, especially for the agricultural sites.*

**Author Response:**

Po Valley (IT-PoV) is used as an example agricultural site here:

**Ammonia**
Mean concentration = 4.5 µg $NH_3$ $m^{-3}$, range = 1.6 – 17 µg $NH_3$ $m^{-3}$
Denuder capture efficiency: Mean = 87 %, range = 57 – 96 %, N = 44

5   **Ammonium aerosol**
Mean concentration = 2.5 µg $NH_4^+$ $m^{-3}$, range = 0.4 – 6.2 µg $NH_4^+$ $m^{-3}$, N = 44
Correction for breakthrough: Mean = 4.5 %
For example (Feb-09 data):
Den 1 = 123 µg $NH_4^+$
10   Den 2 = 12.6 µg $NH_4^+$
Blank = 0.29 µg $NH_4^+$
Capture efficiency = 91 %
Volume of air collected = 21.47 $m^3$ (1.5 month exposure)

$$\chi_a \text{ (corrected)} = \chi_a \text{ (Denuder 1)} \times \frac{1}{1 - \left[\frac{\chi_a \text{(Denuder 2)}}{\chi_a \text{(Denuder 1)}}\right]}$$

$NH_3$ (µg $NH_3$ $m^{-3}$) applying infinite series correction equation above = 6.02

$\chi_a$ (Denuder 1) + $\chi_a$ (Denuder 2)
$NH_3$ (µg $NH_3$ $m^{-3}$) by adding Den 1 + Den 2 = 5.96
20   The correction amounted to 0.06 µg $NH_3$ $m^{-3}$ (= 0.06 µg $NH_4^+$ $m^{-3}$)

Aerosol ammonium = 2.54 µg $NH_4^+$ $m^{-3}$
Corrected (by subtracting 0.06 µg $NH_4^+$ $m^{-3}$ breakthrough from denuders) = 2.48 µg $NH_4^+$ $m^{-3}$
Correction = 2.3 %

**Nitric acid**
Mean concentration = 1.9 µg $HNO_3$ $m^{-3}$, range = 0.5 – 4.0 µg $HNO_3$ $m^{-3}$
Denuder capture efficiency: Mean = 84 %, range = 58 – 94 %, N = 44

**Nitrate aerosol**
Mean concentration = 5.2 µg $NO_3^-$ $m^{-3}$, range = 1.5 – 13.5 µg $NO_3^-$ $m^{-3}$, N = 44
Correction for breakthrough: Mean = 1.4 %

35   For example (Feb-09 data):
Den 1 = 41 µg $NO_3^-$
Den 2 = 5.8 µg $NO_3^-$
Blank = 0.24 µg $NO_3^-$
Capture efficiency = 88 %
40   Volume of air collected = 21.47 $m^3$ (1.5 month exposure)

$$\chi_a \text{ (corrected)} = \chi_a \text{ (Denuder 1)} \times \frac{1}{1 - \left[\frac{\chi_a \text{(Denuder 2)}}{\chi_a \text{(Denuder 1)}}\right]}$$

$HNO_3$ (µg $HNO_3$ $m^{-3}$) applying infinite series correction equation above = 2.23 µg $HNO_3$ $m^{-3}$
45   $\chi_a$ (Denuder 1) + $\chi_a$ (Denuder 2)
HNO3 (µg $HNO_3$ $m^{-3}$) by adding Den 1 + Den 2 = 2.19 µg $HNO_3$ $m^{-3}$
The correction amounted to 0.04 ug HNO3 m-3 (= 0.04 µg $NO_3^-$ $m^{-3}$)

Aerosol (µg $HNO_3$ $m^{-3}$) = 12.61 µg $NO_3^-$ $m^{-3}$
50   Corrected (by subtracting 0.04 µg $HNO_3$ $m^{-3}$ breakthrough from denuders) = 12.57 µg $HNO_3$ $m^{-3}$
Correction = 0.00 %

*7. Page 8, line 13: please refer forward to the results section on the impact of the NaCL denuders.*

**Author Response:**

Thank you, see added text at end of sentence (highlighted)

5  "At the French Fougéres parallel site (FR-FgsP), NaCl coated denuders were used to measure $HNO_3$, to compare with results from $K_2CO_3$/glycerol coated denuders at the main site (FR-Fgs) (see Sect. **2.1Error! Reference source not found.** for methodology and Sect. 3.3.1 for data intercomparison results).

10  *8. Section 2.6: Some countries may have large shipping contributions to NOx and SO2, how did you treat these in the indicator used here? Why did you choose 4 grids around a station and not the nine around and including the grid cell with the station?*

**Author Response to the first part of the comment:**

15  *"Some countries may have large shipping contributions to NOx and SO2, how did you treat these in the indicator used here?*

The comparisons made used the EMEP emissions totals, as reported in each of the grid squares. In the UK, estimates for domestic shipping emissions, based on a database of ship movements are included in the emissions inventory, reported to the EC and EMEP. We have not looked at the breakdown of emission sources in the EMEP database, so we can't say whether shipping emissions are also included in the reporting from countries that have contributions from shipping.

20  As the reviewer indicated, it would be interesting to address the question of shipping emissions in a future measurement-model paper with data from this study. It could include scenarios modelling with and without shipping emissions (e.g. update methodology for estimating emissions using individual ship tracking data) and assess contribution/sensitivities to the gas and aerosol pollution load. Of course, it would also be nice to have a monitoring network across Europe with sufficient spatial coverage and providing speciated gas and aerosol measurements to test the models.

25  **Author Response to the second part of the comment:**

*"Why did you choose 4 grids around a station and not the nine around and including the grid cell with the station?"*

Section 2.6: Line 39 – 40: "Extract gas emissions for groups of 4 grids (each = 0.1º x 0.1º) that surrounds a NEU site and derive grid-averaged emissions"

30  To confirm, the 4 grids selected included a grid cell containing the NEU site.

One grid contains the NEU site, and the other three are the closest in proximity to the grid containing the NEU site, i.e. a block of 4 grids containing and surrounding the NEU site.

To make it clearer, it has been reworded:

35  "Extract gas emissions for blocks of 4 grids (each = 0.1º x 0.1º) and derive grid-averaged emissions. One grid contains the NEU site and the other three in the block are the closest in proximity to grid containing the NEU site"

(note that in response to comments from both reviewers, this section has now been moved to supplementary materials)

We can expect there to exist a stronger correlation between emissions ($NH_3$, $NO_x$, $SO_2$) and the concentrations of the primary

40  pollutants ($NH_3$, $NO_x$, $SO_2$) at the local scale (single grid square), since these reactive gases have relatively short atmospheric lifetimes.

For $NH_3$, which is a diffuse source, emitted mainly at ground level from agriculture, there was good correlation comparing national averages, single grid squares or 4 x grid square average.

45  In the case of $SO_2$, the analysis indicates that a single gridded EMEP square (0.1º x 0.1º) may be too local a spatial scale for an emissions-concentration comparison. Likely reasons are that $SO_2$ emissions are highly localised, from a very small number of large point sources at an elevated height.

Secondary pollutants (HNO$_3$, NH$_4^+$, NO$_3^-$) vary on regional scales, since it takes time for chemical transformation (gas to aerosols) and transport (longer atmospheric lifetimes). Emissions from one grid square could lead to secondary aerosols appearing in adjacent grid squares.

5    We therefore chose blocks of 4 grids (each = 0.1° x 0.1°) as the footprint to compare emissions and concentrations. This is approx. 17 x 22 km for sites that are at latitude 40°. As the reviewer suggests, we could have extended the footprint to include the 8 grid cells around the grid containing the station, to see if this improved the correlation. The comparison against the sum of emissions from an extended number of EMEP grids is more or less what is done with comparison with national emissions density (for the smaller countries at least), with similar results in the correlation. We feel that it will not add anything further

10   to the data interpretation by choosing 9 instead of 4 grid squares.

As some useful, interesting features did emerge in comparisons of concentrations with gridded emissions, according to ecosystem types, we have retained this discussion but have moved it to supplementary materials, together with the associated figures and tables.

Please note that in response to both reviewers' comments concerning comparisons made between emissions and concentrations, we have deleted "Sect. 3.3.3. Comparison with gridded emissions" and moved it to supplementary materials, which also helped to reduce the length of the paper.

*9.   Section 3.3.1 page 14 Line 1-3 details on the dry deposition schemes seem out of place here.*

**Author Response:**
25   On re-reading, we agree that the details on the dry deposition schemes does seem out of place and we have deleted the text (see below).

"In some models such as the Concentration Based Estimates of Deposition (CBED) model (Smith et al., 2000; Flechard et al., 2011), a canopy compensation point and the bi-directional exchange of NH3 between vegetation-type and the

30   atmosphere are also considered (e.g. Sutton et al., 1995; Massad et al., 2010; Flechard et al., 2011)."

**Author Response to the first part of the comment:**

*"The comparison between N and S is based on mass here. Given the scope on ecosystem deposition provided elsewhere I could imagine that a comparison based on acid equivalents makes more sense than the mass."*

We have tried to avoid using too many different units, to permit comparability of concentrations, and to avoid confusion, e.g. we have used units of µg N m$^{-3}$, µg S m$^{-3}$ when referring to mass of gas and aerosol concentrations, and neq. m$^{-3}$ when doing ion balances. Readers should be able to make the conversion to acid equivalents.

A comparison of acidification potential is made in "Section 4. Implications for a chemical climate dominated by $NH_3$ and $NH_4NO_3$ in Europe"

"However, $SO_2$ (by mass) has a higher acidification potential (1 kg $SO_2$ = 1.00 kg eq. $SO_2$) than $NO_x$ (1 kg $NO_2$ = 0.70 kg eq. $SO_2$) (see Hauschild and Wenzel, 1998), so $SO_2$ will remain important in contributing to exceedances of critical loads for acidification, estimated to be exceeded in 5 % of the European ecosystem area in 2015 (EEA, 2019). "

Please note that the sentence was reworded following reviewer 2 comment to simplify sentence (see below) and integrated into Conclusions:

"Although the concentrations of $SO_2$ have fallen to very low levels at all sites, $SO_2$ will continue to be important in contributing to the exceedance of acidification in European ecosystems (EEA, 2019), since $SO_2$ has a higher acidification potential than $NO_x$ (0.70 kg $SO_2$ = 1 kg eq. $NO_2$ in acidity) (see Hauschild and Wenzel, 1998).

 Please note Section 4.0 has been removed and integrated into the Conclusions.

**Author Response to the second part of the comment:**

*I do not see the consistency between the currently higher N levels and emission reductions since the nineties as the emissions did not start from a ratio of 1:1.*

The paragraph in question:

*"A key feature in Figure 7 is the dominance of N over S species at most sites, when expressed as µg m-3 of the element. The mean percentage contribution of sumNr (NH$_3$-N, HNO$_3$-N, NH$_4^+$-N, NO$_3^-$-N) concentrations to the total mass of gas and aerosol species measured is 52% (range = 24 − 80%), twice as much as from sumS (SO$_2$-S and SO$_4^{2-}$-S; mean = 23 %, range = 7 − 53%) (Figure 8). This is consistent with more substantial reductions in SO$_2$ emissions (−72%) than achieved with NO$_x$ 40 (−43%) or NH$_3$ (−18%) in Europe between 1991 − 2010 (EEA, 2019). The differences in atmospheric composition of S and N species in the present assessment therefore reflected changes in emissions of the precursor gases, and are also in agreement with a recent assessment of air quality trends showing important changes in S and N composition in air and rain across the EMEP networks (EMEP, 2016)."*

Perhaps the paragraph is a bit ambiguous, so we have rephrased it to:

"A key feature in Figure 7 is the dominance of N over S species at most sites, when expressed as µg m$^{-3}$ of the element. The mean percentage contribution of sumNr (NH$_3$-N, HNO$_3$-N, NH$_4^+$-N, NO$_3^-$-N) concentrations to the total mass of gas and aerosol species measured is 52% (range = 24 – 80%), twice as much as from sumS (SO$_2$-S and SO$_4^{2-}$-S; mean = 23 %, range = 7 – 53%) (Figure 8). This reflects the smaller emissions in SO$_2$ (4-year average = 319 kt SO$_2$ yr$^{-1}$), compared with emissions of nitrogen gases (4-year average = 614 kt NO$_x$ yr$^{-1}$ and 220 kt NH$_3$ yr$^{-1}$) across the 20 countries in the NEU network. The differences in atmospheric composition of S and N species in the present assessment therefore reflected changes in emissions of the precursor gases, and are also in agreement with a recent assessment of air quality trends showing important changes in S and N composition in air and rain across the EMEP networks (EMEP, 2016)."

*11. Page 15, line 16. The 10-50% contributions in Putaud et al refer to the ammonium salts, not only ammonium. Please correct. Moreover, this paragraph seems more appropriate in the discussion or implication section than in the results chapter.*

**Author Response:**

Thank you. We have corrected in the text – see below:

"Secondary $NH_4^+$ particles are mainly in the 'fine' mode with diameters of less than 2.5 μm ($PM_{2.5}$) and estimated to contribute between 10 to 50 % of ambient $PM_{2.5}$ mass concentration in some parts of Europe (Putaud et al., 2010, Schwartz et al., 2016)."

Amended to:

"Secondary $NH_4^+$ particles are mainly in the 'fine' mode with diameters of less than 2.5 μm ($PM_{2.5}$), with ==ammonium salts== estimated to contribute between 10 to 50 % of ambient $PM_{2.5}$ mass concentration in some parts of Europe (Putaud et al., 2010, Schwartz et al., 2016)."

Section 3.3.1. Comparisons according to ecosystem types is under Chapter 3: Results and Discussions

*12. Page 16 line 3. Here the correlation between precursor and aerosol is discussed in the paragraph on the correlation with emission densities. Right place?*

**Author Response:**

"The particulate components $NH_4^+$ and $NO_3^-$ were also correlated with both precursor gases $NH_3$ and $HNO_3$ (Table 3). By contrast, there was no relationship between $SO_4^{2-}$ with any of the three gases, possibly because of contributions to $SO_4^{2-}$ from long-range transport. All regression plots of concentrations against emission densities, including summary statistics are provided in Supp. Figure S2. "

To clarify, the comparison of particulate components $NH_4^+$ and $NO_3^-$ are with emission densities of $NH_3$ and $NO_2$.

Text amended (Supp Figure no. also updated):

"The particulate components $NH_4^+$ and $NO_3^-$ were also correlated with ==emission densities== of $NH_3$ and $HNO_3$ (Table 3). By contrast, there was no relationship between $SO_4^{2-}$ with ==emission densities of== any of the three gases, possibly because of contributions to $SO_4^{2-}$ from long-range transport. All regression plots of concentrations against emission densities, including summary statistics are provided in Supp. Figure S7. "

**Author Response to the first part of the comment:**

*"HNO3 maybe highest in eastern Europe, but NOx emissions aren't. Could it be that the lower ammonia and hotter summer climate plays a pronounced role in the explanation as indicated in the seasonal cycle with summer maxima in the region (in contrast to western Europe)."*

The larger $SO_2$ concentrations in Eastern Europe (mean 1.8 μg $SO_2$ m$^{-3}$) could also mop up available $NH_3$ (mean = 1.4 μg $NH_3$ m$^{-3}$), limiting available $NH_3$ to react with $HNO_3$.

Additional text added:

"$HNO_3$ formation by photochemical processes may be enhanced in hotter, sunnier summer weather in Russia. Since $SO_2$ concentrations (mean = 0.49 μg $SO_2$-S) at the Russian site (RU-Fyo) is in molar excess over the low levels of $NH_3$ (mean = 0.32 μg $NH_3$-N m$^{-3}$), removal of $HNO_3$ by reaction with $NH_3$ will also be limited."

**Author Response to the second part of the comment:**

*"Similarly, in the presentation of the oxidized nitrogen on page 20 (L 24) the limitation on ammonia availability could be mentioned."*

An explanation is already offered for the higher $HNO_3$ at the Russian site, so we feel that mentioning the limitation on ammonia availability as a possible mechanism in controlling atmospheric concentrations of $HNO_3$ is unnecessary repetition.

14. *" The higher correlation between nitrate and ammonia emissions is indicative for this issue as well. Ammonium nitrate formation could be checked with the ammonium salt ion balance. Often inverse relationships between nitric acid and ammonia are modelled due to the limiting impact of the equilibrium with ammonium nitrate. Do you see this feature in the data?"*

**Author Response to the first part of the comment:**

*Ammonium nitrate formation could be checked with the ammonium salt ion balance :* this is already covered in Section 3.4 Correlations between gas and aerosol components.

"In the aerosol phase, $NH_4^+$ correlated well with $NO_3^-$ ($R^2 = 0.75$, $p < 0.001$, Figure 13A) and $SO_4^{2-}$ ($R^2 = 0.75$, $p < 0.001$, Figure 13B) (Tables 5 and 7), but not with $Cl^-$ (Table 5). Regression of the molar equivalent concentrations of the sum of $NO_3^-$ and $SO_4^{2-}$ against $NH_4^+$ show points close to the 1:1 line (slope = 0.84) and significant correlation ($R^2 = 0.64$, $p < 0.001$), which demonstrates the close coupling between the base $NH_4^+$ and the acid $NO_3^- + SO_4^{2-}$ aerosols (Figure 13C, Table 7)."

**Author Response to the second part of the comment:**

*"inverse relationships between nitric acid and ammonia"*

Below is a plot of mean $HNO_3$ versus site mean $NH_3$. We don't see an inverse relationship, although there appears to be a curvilinear relationship, with $HNO_3$ concentrations plateauing at $NH_3 > 2$ µg $NH_3$ m$^{-3}$.

[Figure]

The analysis is provided here to address the reviewers question only and has not been added to the paper.

15. *Page 21, the current levels are interpreted in relation to emission reductions which are not indicated from this network. The SO2 to SO4 ratio variability across the network may be the most interesting feature concerning sulfur for model developers. Did you see the anticipated systematic behavior for this ratio?*

**Author Response:**

Please see plots prepared below:

There appears to be different trends in $SO_2$ and $SO_4^{2-}$ according to geographic regions.

The $SO_2$ to $SO_4^{2-}$ ratio therefore also varies according to grouped regions.

[Figure]

A decrease in ratio of SO₂ to SO₄²⁻ : would suggest increased dry deposition of SO₂ ("co-deposition due to increasing ratio of NH₃ to SO₂ in the atmosphere). This results in a larger decrease in atmospheric SO₂ concentrations than would be achieved by emissions reduction alone.

A stable ratio of SO₂ to SO₄²: would suggest that maximum deposition rates for SO₂ may have been reached with the smaller SO₂ concentrations since 2006.

Since there are only 4 years of data, the analysis and discussion is provided here to address the reviewers question only and have not been added to the paper.

*16. Page 21 the Bugac discussion interrupts the main information flow.*
**Author Response:**
We have simplifed / shortened the text. Discussions on gridded emissions was removed, since

i) section on comparison with gridded emissions has been deleted, and
ii) does not add substantively to understanding of what is happening at the site.

See amended text below:

"SO₂ concentrations were also correlated with SO₂ emission density ($R^2 = 0.65$, $p < 0.001$, $n = 20$) in each country (Figure 10A3, Table 3). The smallest and largest SO₂ annual average concentrations corresponded with the lowest emissions in Norway and highest in the Czech Republic (Figure 9C). By contrast, SO₂ concentrations from the single measurement site Bugac in Hungary (HU-Bug) are much higher than expected on the basis of SO₂ emission density estimated for the country. This suggests that Bugac is likely to be affected by proximity to sources. This contrasts with the BKFores site in the Czech Republic (CZ-BK1) which had smaller NH₃ concentrations due to its location away from sources. "

-
-
-

~~Although the Bugac site is located in a grid with low emissions of all the gases, the higher SO₂ (1.2 µg S m⁻³), together with elevated NH₃ (2.6 µg N m⁻³) and HNO₃ (0.3 µg N m⁻³) concentrations measured at this site suggests that it is likely to be affected by proximity to sources. This contrasts with the BKFores site in the Czech Republic (CZ-BK1) which had smaller NH₃ concentrations due to its location away from sources. "~~

*17. Page 22. The ion balance for southern Scandinavia may be affected by sodium nitrate formation and not so much by an overestimation of SO4. Na:CL depletion ratio may give a hint here. Further down on the same page the remark is made but no connection is made.*
*On page 24 another check is made on ion balances with hard statements on lab quality – are these issues not connected and is one actually looking at sea salt depletion reactions?*

**Author Response:**

Southern Scandinavian sites

| id | name | μg m⁻³ | | | | | | | | | Ratio | |
|---|---|---|---|---|---|---|---|---|---|---|---|---|
| | | NH₃ | HNO₃ | SO₂ | HCl | pNH₄⁺ | pNO₃⁻ | pSO₄²⁻ | pCl⁻ | pNa⁺ | Na:Cl (neq) | NH₄⁺: (NO₃⁻ + SO₄²⁻) (neq) |
| 63 | Brandbjerg | 0.77 | 0.76 | 0.85 | 0.25 | 1.16 | 1.74 | 2.01 | 0.21 | 0.63 | 4.6 | 0.93 |
| 34 | Rimi | 1.47 | 1.05 | 0.74 | 0.26 | 0.74 | 2.46 | 2.34 | 0.88 | 1.28 | 2.3 | 0.47 |
| 35 | Risbyholm | 5.26 | 0.70 | 0.61 | 0.20 | 0.71 | 1.82 | 2.00 | 0.33 | 1.02 | 4.8 | 0.56 |
| 30 | Soroe | 1.54 | 0.97 | 0.91 | 0.33 | 0.74 | 2.55 | 2.32 | 0.53 | 0.92 | 2.7 | 0.46 |
| 62 | Birkenes | 0.29 | 0.37 | 0.17 | 0.34 | 0.28 | 0.42 | 1.03 | 0.26 | 0.43 | 2.5 | 0.55 |
| 36 | Norunda | 0.32 | 0.23 | 0.17 | 0.14 | 0.25 | 0.22 | 1.09 | 0.05 | 0.21 | 6.1 | 0.52 |
| 37 | Skyttorp | 0.14 | 0.33 | 0.17 | 0.17 | 0.19 | 0.30 | 1.14 | 0.09 | 0.20 | 3.5 | 0.37 |

[Figure]

The four Danish sites Brandbjerg (63), Rimi (34), Risbyholm (35) and Soroe (30) are all very close together (see map).

The Na:Cl ratios varied between 2.3 at Rimi to 4.8 at Risbyholm.

5  The appearance of excess sodium at the sites may be due to uncertainty (underestimation of chloride concentrations) at these sites, as discussed in the manuscript.

The two Swedish sites Norunda (36) and Skyttorp (37) are also close together (see map).

The Na:Cl ratios varied between 3.5 at Skyttorp to 6.1 at Norunda.

10  The appearance of excess sodium at the sites may therefore likely be due to underestimation of chloride concentrations, which are very close to or below the detection limit of the method. The quality (e.g. variability) of the blanks (data not available) at such low concentrations will also have a proportionately large effect on the calculated Cl⁻ concentrations.

Finland

| id | name | μg m⁻³ | | | | | | | | | Ratio | |
|---|---|---|---|---|---|---|---|---|---|---|---|---|
| | | NH₃ | HNO₃ | SO₂ | HCl | pNH₄⁺ | pNO₃⁻ | pSO₄²⁻ | pCl⁻ | pNa⁺ | Na:Cl (neq) | NH₄⁺: (NO₃⁻ + SO₄²⁻) (neq) |
| 41 | Hyytiälä | 0.11 | 0.46 | 0.54 | 0.19 | 0.22 | 0.20 | 1.38 | 0.03 | 0.17 | 9.8 | 0.3 |
| 31 | Sodankylä | 0.17 | 0.23 | 0.57 | 0.17 | 0.17 | 0.09 | 1.24 | 0.06 | 0.16 | 3.8 | 0.3 |
| 32 | Kaamanen | 0.79 | 0.12 | 0.93 | 0.17 | 0.32 | 0.05 | 0.64 | 0.15 | 0.14 | 1.4 | 1.3 |
| 33 | Lompolojänkkä | 0.09 | 0.17 | 0.25 | 0.22 | 0.19 | 0.06 | 0.79 | 0.06 | 0.12 | 3.0 | 0.6 |

The three sites in Finland Sodankylä (31), Kaamanen (32) and Lompolojänkkä (33) are all inland sites, in close proximity to each other in the North of Finland (see map).

The Na:Cl ratio at Kaamanen was 1.4, whereas the two nearby sites showed ratios of 3.0 and 3.8.

At Hyytiälä (41), a site that is further south in Finland, the ratio was even larger, at 9.8.

The LOD for aerosol chloride measurement on the DELTA system is around $0.1 - 0.16$ µg Cl⁻ m⁻³ for monthly exposures. The appearance of excess sodium at the sites may therefore likely be due to underestimation of chloride concentrations, which are very close to or below the detection limit of the method. The quality (e.g. variability) of the blanks (data not available) at such low concentrations will also have a proportionately large effect on the calculated Cl⁻ concentrations.

*18. Page 22: does the HCl distribution provide a hint at the importance of the marine source for it?*

**Author Response:**

At coastal sites, HCl released from the reaction of sea salt with $HNO_3$ and $H_2SO_4$ can be a significant source. Part of the chloride of sea salt can be substituted by $SO_4^{2-}$ and $NO_3^-$ through a reaction with $H_2SO_4$ and $HNO_3$, known as the Cl⁻ deficit

Sea salt depletion: NaCl (p) + H⁺ (p) => Na⁺ (p) + HCl (g)

H⁺ = from $H_2SO_4$, $HNO_3$

p = particle, g = gas

Looking at the spatial distribution of HCl, site mean concentrations varied between 0.06 at Renon (Italy, inland, site 10) to 0.50 at Espirra (Portugal, coastal, site 12). So it appear at first glance that HCl is elevated at the coastal Espirra site, possibly from the reaction described above.

However, site mean HCl concentrations at other coastal sites in the network were in the range of 0.14 (Solohead, Ireland) to 0.34 µg HCl m⁻³ (Birkenes, Norway), similar to the range across the entire network (0.06 = 0.50 µg HCl m⁻³ described above)

It cannot therefore be concluded that there is a potential marine source for HCl.

*19. Page 27 line 1-5: the impact of ammonia and temperature on seasonality of nitric acid is not discussed ad should be mentioned. OK, it is done in the next paragraph. Why not combine these?*

**Author Response:**

Paragraph 1 focuses on the influence of photochemistry on the formation of $HNO_3$.

Paragraph 2 goes on to look at other drivers: temperature and $NH_3$ on formation and partitioning between the gas and aerosol phase. We feel that the discussion can be split into two paragraphs in this way.

**Other updates made to paper:**

Co-author name and affiliation:

Francisco would prefer to be listed with the following name and address:

Francisco Sanz

Fundación CEAM, C/Charles R. Darwin, 46980 Paterna (Valencia), Spain

Acknowledgment:

Updated
10 The authors thank reviewer 2 for their supportive comments for publication and for taking the time to look at all the details described in the manuscript. We have carefully considered all comments. Please refer to the specific responses.

*"This manuscript describes measurements collected within the EU NitroEurope (NEU) network during the period 2006*
15 *– 2010. While some of this data has been previously published, as noted by the authors, the current manuscript provides a comprehensive description of the data quality as well as temporal and spatial patterns of atmospheric chemistry over the lifetime of the network. The data will make a valuable contribution to the field of atmospheric chemistry, in particular with respect to better understanding the role of reduced forms of reactive nitrogen in aerosol processes and for model evaluations. The manuscript is generally well written and the analyses are appropriate, though the manuscript is*
20 *somewhat lengthy. I recommend publication subject to treatment of the relatively minor comments outlined below, some of which are technical in nature and others seek to reduce the length of the paper."*

**Individual remarks:**
25 *1. "Page 5 – Replicate measurements. It would be useful to see a bit more detail on the replicated measurements to get a better sense of overall precision. For example, scatterplots and summary statistics could be added to the Supplemental Material."*

**Author Response 1:**
30 ➢ Regression analyses and statistics, including t-tests are added in Supp. Figures S2 to S5 for the 4 parallel sites:
- UK Auchencorth Moss: UK-AMoP vs UK-AMo
- UK Easter Bush: UK-EBuP vs UK-EBu
- French Fougéres: FR-FgsP vs FR-Fgs
- Slovakian EMEP site: SK04P vs SK04

35 ➢ Comparisons of annual and overall means for the above 4 sites added in Supp. Tables S2 to S5 (attached at the end of this document)

➢ Subsequent Supp. Figures and Supp. tables renumbered accordingly after insertion of above.

**Page 14:** references to Supp. Figures and Tables are inserted in text (see highlighted text in the text copied below). Months
40 where paired data are not available have been excluded in the updated regression analyses, t-tests and in the comparisons of mean concentrations, whereas previous analyses included all data points. Numbers are therefore also updated to reflect the updated analyses (see highlighted text).

**3.3.1 Comparisons according to ecosystem types, paragraph 3**

"Sites with parallel (P) DELTA® measurements were Auchencorth Moss (UK-AMoP), Easter Bush (UK-EBuP), Fougéres
45 (FR-FgsP) and SK04P (EMEP site in Slovakia) (Figure 7). Overall, good reproducibility in DELTA® measurements was demonstrated by the parallel measurements (Supp. Figures S3 - S6). At the Auchencorth Moss parallel site (UK-AMoP), NH$_3$ and NH$_4^+$ only were measured, and agreement for these 2 components were on average within 54 % at the low concentrations measured at this site (annual mean: 0.5 – 0.9 µg NH$_3$ m$^{-3}$ and 0.3 – 0.5 µg NH$_4^+$ m$^{-3}$) (Supp. Table S5). Parallel measurements at Easter Bush (UK-EBuP) stopped in March 2010. With the exception of Ca$^{2+}$ and Mg$^{2+}$, the comparison of annual mean data
50 from the replicated measurements for 20076 to 2009 provided excellent agreement of 24 % (NO$_3^-$) to 1312 % (SO$_4^{2-}$NH$_3$) at Easter Bush (Supp. Table S6). At Fougéres (Supp. Table S7), HNO$_3$ concentration measured on K$_2$CO$_3$/Glycerol coated denuders (FR-Fgs) was about 2-fold higher than on NaCl coated denuders in the parallel DELTA® system (FR-FgsP), consistent with over-estimation of HNO$_3$ (on average 45 %) on carbonate coated denuders (see Sect. 2.2.3). The disadvantage

of a NaCl coating, however, is that it can only collect $HNO_3$ and not the other acid gases. A third carbonate denuder is necessary in the sample train to collect and measure $SO_2$, since $SO_2$ is only partially captured and HCl cannot be measured on NaCl denuders (Tang et al., 2015, 2018b). This explains the smaller $SO_2$ concentrations reported by the FR-FgsP site, with break-through of $SO_2$ (inefficiently captured by NaCl denuders) onto the aerosol filters resulting in larger particulate $SO_4^{2-}$

5 concentrations than the Fr-Fgs site. For the SK04 site, measurement reproducibility for the 4 years of parallel data for N and S component was good, with agreement ranging from 0.41.2 % ($NH_4^+$) to 59 % ($SO_4^{2}$) (Supp. Table S8). HCl and $Na^+$ and determinations were however more uncertain with differences of 2167 and 2843 %, respectively (Supp. Table S8). It has to be noted, however, that the concentrations of the two components were very low, at < 0.2 μg HCl m$^{-3}$ and < 0.4 μg $Na^+$ m$^{-3}$. The differences in concentrations are therefore actually within ± 0.1 μg m$^{-3}$ for HCl and within ± 0.2 μg m$^{-3}$ for $Na^+$. "

*2. Page 6 – Coordinating laboratories. Some brief discussion of the analytical methods employed by the laboratories (ion chromatography or colorimetry) should be included, along with some discussion or reference to method detection limits (MDL). This information could also be included in the Supplemental.*

**Author Response:**
Supp. Table S3 added:
"Supp. Table S3. Details on analytical methods used in the analysis of anions ($NO_3^-$, $SO_4^{2-}$, Cl$^-$) and cations ($NH_4^+$, $Na^+$, $Ca^{2+}$, $Mg^{2+}$) in aqueous denuder and filter extracts in the NEU DELTA® network (all labs) and in precipitation

20 samples from the NEU wet deposition network (INRA and SHMU)."
In manuscript:
Text added (see highlighted text below)

Page 6, Section 2.1.1 Coordinating laboratories, paragraph 1, line 3:
"A team of seven European laboratories shared responsibility for running the network. Measurement was on a monthly

25 timescale, with each laboratory preparing and analysing the monthly samples with documented analytical methods (see Supp. Table S3 for information on analytical methods and limit of detection (LOD)) for between 5 and 16 DELTA sites (Figure 2).

3. *Page 8 – Bulk precipitation measurements. These measurements will no doubt be useful for deposition assessments. However, as currently written the data do not add much to the current manuscript and could be removed to reduce overall length.*

**Author Response:**

Cutting out the Bulk precipitation measurement sections would reduce the overall length by about a page only and it would be a shame to cut out this valuable dataset which might otherwise not be so readily available to the community. We feel it is important to retain the wet deposition measurements in the paper, as it highlights where DELTA® and bulk wet deposition data are co-located and provides parallel information on gas and aerosol concentrations (for dry deposition modelling) and wet deposition at those sites. The co-located data is important for deriving N budgets and linking to ecosystem response (e.g. recent paper by Flechard et al. 2020) and invaluable for modellers.

Section 3.6 Bulk wet deposition measurements

Page 29, lines 27 - 30

*"The intention of the bulk network at the outset was to provide wet deposition data at DELTA® sites that do not already have such measurements on site. The wet deposition data on $NH_4^+$ and $NO_3^-$, combined with a wider precipitation chemistry dataset (e.g. from EMEP and other national precipitation networks) was used to estimate total $N_r$ deposition to a site (Flechard et al., 2011; 2020)."*

This has been reworded to set out the intentions of the bulk wet deposition measurements and moved to Introduction (last paragraph).

"In this paper, we present and discuss four years of monthly reactive gas ($NH_3$, $HNO_3$, $HCl$) and aerosol ($NH_4^+$, $NO_3^-$, $SO_4^{2-}$, $Cl^-$, $Na^+$, $Ca^{2+}$, $Mg^{2+}$) measurements from the Level 1 network set up under the NEU integrated project (Figure 2). A harmonised measurement approach with a simple, cost-efficient time-integrated method, applied with high spatial coverage allowed a comprehensive assessment across Europe. The gas and aerosol network was complemented by two years of bulk wet deposition data made at a subset of the sites (Figure 3). The intention of the smaller wet deposition network was two-fold, i) to provide wet deposition estimates at DELTA® sites that do not already have such measurements on site, and ii) to compare the relative importances of reduced and oxidized N versus sulfur in the atmospheric pollution load. Measurements across the network were coordinated between multiple European laboratories. The measurement approach and the operations of the networks, including the implementation of annual inter-comparisons to assess comparability between the laboratories, are described. The data are discussed in terms of spatial and temporal variation in concentrations, relative contribution of the inorganic nitrogen and sulfur components to the inorganic pollution load, and changes in atmospheric concentrations of acid gases and their interactions with $NH_3$ gas and $NH_4^+$ aerosol.

Additional text is also added at the end of Section 3.6. Bulk wet deposition measurements on the relevance of the wet deposition data:

"The wet deposition measurements in this paper highlights where DELTA® and bulk wet deposition data are co-located and provides parallel information on gas and aerosol concentrations (for dry deposition modelling) and wet deposition at those sites. The co-located data is important for deriving N budgets and linking to ecosystem response (e.g. Flechard et al. 2020) and invaluable for modellers."

And in Section 3.4. (page 23, lines 40 - 43) (highlighted text)

This demonstrates that sea salt $SO_4^{2-}$ (ss-$SO_4^{2-}$) aerosol makes up a large and variable fraction of the total $SO_4^{2-}$ measured, consistent with observations of the contribution by ss-$SO_4^{2-}$ to the total $SO_4^{2-}$ in precipitation observed in the wet deposition measurements in this study (Figure 11) and across Europe (ROTAP, 2012).

*4. Page 11 – Line 20. The comparison is referred to here as "field inter-comparison" but as "laboratory inter-comparison" in the 3.2 section heading. I understand the distinction, but it is a little confusing at first glance.*

**Author Response:**

Thank you. On re-reading, we agree with the reviewer that the section heading is misleading.

"Section 3.2. Laboratory inter-comparison results: DELTA® measurements"

Changed to

"Section 3.2. Field inter-comparison results: DELTA® measurements"

Section 2.5. Laboratory inter-comparisons: DELTA® measurements"

Also changed to:

Section 2.5. Field inter-comparisons: DELTA® measurements"

*5. Page 12 – Line 14. Knowledge of the laboratory blanks would be very helpful. Is there no way to recover the results from original chromatograms? Granted it might be time consuming but interlaboratory comparison of blanks, particularly for NH₃ which is notoriously difficult, could be enlightening as to some of the laboratory comparisons.*

Page 12, Lines 12 - 14

*"A possible cause may be the quality and/or variability in the aerosol filter blank values for $NH_4^+$, as laboratory blanks are subtracted from exposed samples to estimate aerosol $NH_4^+$ concentrations. Laboratory blank results were however not reported to allow this assessment."*

**Author Response:**

We have managed to extract laboratory and field blank data from the original submitted laboratory files (covering the DELTA intercomparison periods).

Supplementary Figure S2 with boxplots comparing lab and field blanks added (see below):

Text has been amended to:

"A possible cause may be the quality and/or variability in the aerosol filter blank values for $NH_4^+$, as laboratory blanks are subtracted from exposed samples to estimate aerosol $NH_4^+$ concentrations. ==While the laboratory blanks reported by MHSC for aerosol $NH_4^+$ were low (mean = 0.48 µg $NH_4^+$) and smaller than other laboratories (mean = 0.64 – 1.20 µg $NH_4^+$) (Supp. Fig. S2), their field blanks in the 2006 DELTA intercomparison exercise were on average 5.5 times larger than the laboratory blanks. This is likely due to extensive delays in getting samples released from customs in Slovakia at the start of the network."==

The comparison shows similar range in blank values in the acid coated aerosol filters and denuders between laboratories. The mean amount of ammonium ($NH_4^+$) in the acid coated aerosol filters ranged between 0.48 to 1.76 µg across all laboratories. This is equivalent to an atmospheric concentration of 0.06 to 0.23 µg $NH_4^+$ $m^{-3}$, based on air volume of 7.5 $m^3$ sampled by DELTA® system over a 2-week exposure period in the 2006 DELTA intercomparison at the clean site (Auchencorth).

[Figure]

| | Aerosol filter: Lab Blank | | Aerosol filter: Field Blank (FB) | |
|---|---|---|---|---|
| | μg NH$_4^+$ in extract | Equivalent aerosol concentration for 2 week exposure (μg NH$_4^+$ m$^{-3}$) [1] | μg NH$_4^+$ in extract | Equivalent aerosol concentration for 2 week exposure (μg NH$_4^+$ m$^{-3}$) [1] |
| VTI | mean = 0.99 (*n* = 8) range = 0.62 – 1.37 | mean = 0.13 (*n* = 8) range = 0.08 – 0.18 | | |
| NILU | mean = 0.99 (*n* = 8) range = 0.62 – 1.37 | mean = 0.23 (*n* = 8) range = 0.19 – 0.29 | | |
| SHMU | mean = 0.64 (*n* = 4) range = 0.44 – 1.05 | mean = 0.10 (*n* = 4) range = 0.06 – 0.14 | | |
| MHSC | mean = 0.48 (*n* = 9) range = 0.28 – 0.76 | mean = 0.06 (*n* = 9) range = 0.04 – 0.10 | mean = 2.70 (*n* = 9) range = 1.53 – 3.97 | mean = 0.36 (*n* = 9) range = 0.20 – 0.53 |
| UKCEH | mean = 1.05 (*n* = 10) range = 0.88 – 1.33 | mean = 0.14 (*n* = 10) range = 0.12 – 0.18 | mean = 1.24 (*n* = 6) range = 0.98 – 1.50 | mean = 0.16 (*n* = 6) range = 0.13 – 0.20 |
| CEAM | mean = 1.21 (*n* = 7) range = 0.59 – 1.78 | mean = 0.16 (*n* = 8) range = 0.08 – 0.24 | | |

Equivalent aerosol concentrations, based on air volume of 7.5 m$^3$ sampled by DELTA® system over a 2-week exposure period in the 2006 DELTA intercomparison.

Figure S2: Comparison of laboratory and field blanks (where reported) for ammonium aerosol filters from the 2006 DELTA intercomparison exercise between six participating laboratories.

To put the blank values into context, the amount of NH$_4^+$ in the lab. and field blanks are compared with amount of NH$_4^+$ collected in exposed DELTA aerosol samples in the 2006 DELTA intercomparison. At the four intercomparison sites (Auchencorth, Braunschweig, Montelibretti and Paterna), the amount of ammonium (μg NH$_4^+$) ranged between 2.9 to 16.9. The lab. blank values were therefore acceptably low, being ~1/10$^{th}$ of the smallest concentrations at the cleanest site, Auchencorth.

Field blanks were reported by two laboratories only (MHSC and UKCEH), and compared in the box plots below. While the reported lab and field blanks were not dissimilar from the UKCEH lab, the field blanks for MHSC were on average 5.5 times larger than their lab. blanks. In the DELTA protocol, lab. blanks are subtracted from exposed samples, whereas field blanks serves as a quality check on potential contamination during storage and transport. The larger MHSC field blank values may be due to returned samples being held for extended periods of time in customs in Slovakia and may account for the larger aerosol NH$_4^+$ concentrations reported by MHSC.

[Figure]

Comparison of the amount of NH$_4^+$ in the lab. and field blanks (where available) with amount of NH$_4^+$ collected in exposed DELTA® aerosol samples in the 2006 DELTA intercomparison.

[Figure]

Equivalent gas concentrations estimated for lab and field blanks, based on air volume of 7.5 m$^3$ sampled by DELTA$^®$ system over a 2-week exposure period in the 2006 DELTA intercomparison.

A comparison of denuder lab and field blanks (where reported) are also shown below, to demonstrate the good quality of denuder blanks achieved in ammonia measurements by the DELTA$^®$ method.

[Figure]

6. *Page 12 – Line 23. Was CEAM the only laboratory that used colorimetric analysis for NH4+? See previous comment on summarizing analytical techniques used by the various laboratories.*

**Author Response:**
No, see author response to Comment 2
Other labs that used colorimetric analysis for determination of aqueous $NH_4^+$ are:
- INRA (Salicylic acid)
- NILU (Indophenol)

7. *Page 15 – Line 4. The overestimation of HNO3, or at least that the HNO3 measurement includes other oxidized N compounds, could be noted again here.*

**Author Response:**
Text added – see below (highlighted)
"Most of the $N_r$ concentrations at each site in turn are dominated by reduced N ($NH_3$-N, $NH_4^+$-N), rather than by oxidised N species ($HNO_3$-N (includes other oxidized N compounds, see Sect. 2.2.3) and $NO_3^-$-N)."

8. *Page 15 – Line 30. See previous comment regarding LOD/MDL for different laboratories/chemical species.*

**Author Response:**

Supp. Table S3 with details of analytical methods and LODs is now added.

Reference to Supp. Table S3 added in text

"The concentrations of $Ca^{2+}$ and $Mg^{2+}$ were very low across the network, with values (mean of all sites < 0.1 µg m$^{-3}$) that were at or below method limit of detection (LOD = ~ 0.1 µg m$^{-3}$) (Supp. Table S3)."

**Author Response:**

**Sect. 3.3.3. Comparison with gridded emissions**: Deleted and moved to supplementary materials.

**Sect 3.3.2. Comparisons with national gas emissions**: Retained

Additional supporting text added at the end of section 3.3.2 (see below):

"The comparisons here used national emission totals, where emissions have been summed and averaged across very large and heterogeneous areas in each country. Additional analysis were also undertaken to compare the individual site mean data with i) gridded emissions from individual 0.1° x 0.1° EMEP grids in which the NEU sites are located (Supp. Figure S8, S9), and ii) averaged emissions of an extended number of EMEP grids (4 x grids) closest to the site (Supp. Figure S10). Since results from these analyses were similar to the comparisons with national emission densities, they are not included for further discussions in this paper. The purpose of the ranked emission densities is to compare the pollution climate in terms of primary gas emissions ($SO_2$, $NO_2$, $NH_3$) across the 20 European countries and to see if this is matched by the DELTA measurements. Despite the complex relationship between emissions and concentrations, the pollution gradient in Europe is clearly captured by the present data. At the same time, it also demonstrated the potential application of the DELTA$^®$ approach in providing national concentration fields, as evidence to compare against spatial and long-term trends in the national emissions data."

10. *Page 17 – Line 25. Were the high concentrations at IT-BCi indicative of highly local emissions, i.e., adjacent to the field site, or is this concentration more indicative of a broader area? It would be impractical to include a description of every site but where such details are relevant, they should be included. In the same regard, it would be good to know if all of the grassland sites are grazed (Page 17 – Line 41). It appears so.*

**Author Response:**

IT-BCi is an ecosystem station located in a 15 ha field (arable crops) on the Sele Plain, an agricultural area with intensive buffalo farming in Southern Italy. The site is not affected by close proximity to sources (e.g. animal housing or manure stores), so the annual mean concentrations of 8 ug $NH_3$ m$^{-3}$ is indicative of the broader area. Close to sources (e.g. within 300 m), annual mean concentrations can be expected to be even higher.

"Borgo Cioffi (IT-BCi) in an intensive buffalo farming region of Southern Italy provided the highest 4-year average of 8.1 µg $NH_3$-N m$^{-3}$ (cf. group mean = 3.8 µg $NH_3$-N m$^{-3}$ , n = 10) (Table 4, Supp. Table S4)."

Further details of the site has been added in text:

"Borgo Cioffi (IT-BCi) is an ecosystem station located in a 15 ha field (arable crops) on the Sele Plain, an agricultural area with intensive buffalo farming in Southern Italy and this provided the highest 4-year average of 8.1 µg $NH_3$-N m$^{-3}$ (cf. group mean = 3.8 µg $NH_3$-N m$^{-3}$ , n = 10) (Table 4, Supp. Table S4)."

*11. Page 18 – Line 38/39. I believe "will dominate dry NH3-N dry deposition" should be changed to "will dominate dry N deposition", correct?*

**Author Response:**

Thank you.
"…then NH$_3$ will dominate dry NH$_3$- N dry deposition and exert the larger ecological impact."

Corrected (highlighted):

"…then NH$_3$ will dominate dry N deposition and exert the larger ecological impact."

*12. Page 19 – Line 11. Remove "that are".*
*13. Page 19 – Line 12. Change "emission" to "emissions"*

**Author Response:**
"….ranging from 0.05 to 6.7 µg NH$_3$-N m$^{-3}$ that are consistent with smaller NH$_3$ emission from the UK (Figure 9A).

Amended (highlighted):

"….ranging from 0.05 to 6.7 µg NH$_3$-N m$^{-3}$, consistent with smaller NH$_3$ emissions from the UK (Figure 9A).

*14. Page 21 – Line 7. The sentence beginning "This corroborates…." is quite lengthy.*

**Author Response:**
*"Annual averaged SO$_2$ concentrations measured across the network were between 0.9 and 2.3 µg SO$_2$-S m$^{-3}$ (Figure 9C, Supp. Table S9). This corroborates observations from monitoring made in the EMEP networks of large reductions in ambient concentrations and deposition of sulfur species during the last decades (EMEP, 2016), reflecting successes of air quality policies across Europe in achieving substantial reductions in SO$_2$ emissions, which decreased by 74 % between 1990 and 2010. Annual mean SO2 concentrations of 0.03 to 5.5 µg SO$_2$-S m$^{-3}$ were reported from the EMEP network from 58 rural background sites across Europe over the period of 2007 – 2010, with largest SO$_2$ concentrations from North Macedonia and Serbia (EMEP, 2016). Since the highest emitting countries in European countries were not included in the DELTA® network, the SO$_2$ concentrations provided by the DELTA® network are smaller, but are within the range reported by EMEP (EMEP, 2016)."*

Paragraph rephrased – see below:
(Supp. Table numbering also updated)

"Annual averaged SO$_2$ concentrations measured across the network were between 0.9 and 2.3 µg SO$_2$-S m$^{-3}$ (Figure 9C, Supp. Table S14). By comparison, the EMEP network of 58 urban background sites reported annual mean concentrations of 0.03 and 5.5 µg SO$_2$-S m$^{-3}$ over the same period, with largest SO$_2$ concentrations from North Macedonia and Serbia. Since these high emitting countries were not included in the DELTA® network, the range of SO$_2$ concentrations are smaller. Together, the small SO$_2$ concentrations reflect the substantial reductions in SO$_2$ emissions across Europe (-74 % between 1990 and 2010) and large reductions in ambient concentrations and deposition of sulfur species across Europe during the last decades (EMEP, 2016).

*15. Page 23 – Section 3.4. It appears that Figure 13 is incorrectly referred to as Figure 12 throughout this section.*

**Author Response:**

Thank you for spotting the mistake.

Page 23 – Section 3.4

Throughout this section:

1st paragraph: Figure 11 corrected to Figure 12 (two times)

Rest of section: Figure 12 corrected to Figure 13 (eleven times)

*16. Page 24 – Line 16. Are there other potential reasons for the higher sulfate measurements at these sites? Seems worthy of additional investigation/discussion.*

**Author Response:**

Figure 13C from paper is copied below, showing outliers in the regression plot.

[Figure]

Regression plots of individual monthly measurements at all sites managed by NILU are shown below.

Ratio [neq $NH_4^+$ : sum (neq $NO_3^-$ + neq $SO_4^=$)]

- Ratio = 0.9 – 1.1: 10.4% of data
- Ratio < 0.5: 42.7 % of data
- Ratio > 1.5: 7.9 % of data

[Figure]

This indicates either an over-read of the anions ($NO_3^-$, $SO_4^{2-}$), or under-read of $NH_4^+$ concentrations.

On closer examination of individual monthly site data:

- 14.6 % of aerosol $NH_4^+ \leq 0.1$ µg m$^{-3}$.
- 17.1 % of $NO_3^-$ (µg m$^{-3}$) $\leq 0.1$ µg m$^{-3}$
- Only 0.7 % of all $SO_4^{2-}$ (µg m$^{-3}$) data were $\leq 0.1$ µg m$^{-3}$.

This then points to a potential under-read of $NH_4^+$ and $NO_3^-$. Possible reasons:

i)   loss of $NH_4^+$, $NO_3^-$ from filters (e.g. microbial degradation),

ii)  non-capture on the aerosol filters (e.g. aerosol filters installed wrong way round),

iii)     filters mixed up and wrong analysis performed on the respective acid and base-coated filters,

iv)     high $NH_4^+$, $NO_3^-$ blanks subtracted from already low concentrations at clean sites.

Possibilities also still remain of an over-read in $SO_4^{2-}$.

Regression plots of individual monthly measurements at all sites managed by CEAM are shown below.

Ratio [neq $NH_4^+$: sum (neq $NO_3^-$ + neq $SO_4^=$)]

- Ratio = 0.9 – 1.1: 5.4 % of data
- Ratio < 0.5: 36.2 % of data
- Ratio > 1.5: 6.2 % of data

This indicates either an over-read of the anions ($NO_3^-$, $SO_4^{2-}$) or under-read of $NH_4^+$ concentrations.

On closer examination of individual monthly site data:

- 1.5 % of aerosol $NH_4^+$ < 0.1 µg m$^{-3}$.
- 0.8 % of $NO_3^-$ (µg m$^{-3}$) < 0.1 µg m$^{-3}$
- All $SO_4^{2-}$ (µg m$^{-3}$) > 0.1 µg m$^{-3}$

This does not show any apparent low outliers in the data. The regression plots also show points distributed on either side of 1:1 line.

[Figure]

**In manuscript**

*"Removal of the outlier NILU (7 out of 16) and CEAM (1 out of 3) data points with ion balance ratio < 0.5 improved both the slope (new slope = 0.90) and correlation (new $R^2$ = 0.78) (Figure 13C). This indicates either an over-read of the anions ($NO_3^-$, $SO_4^{2-}$) or under-read of $NH_4^+$ concentrations by the two laboratories at some sites. Results reported by NILU in the DELTA® field inter-comparisons (Sect. 3.2) showed that, with the exception of a few high $NH_4^+$ and $NO_3^-$ readings, there was on average no overall bias in the $NH_4^+$, $NO_3^-$ or $SO_4^{2-}$ measurements by the NILU laboratory that could account for the high $SO_4^{2-}$ outliers in the regression (Figure 13). The ion balance checks suggest possible over-read and increased uncertainty in the $SO_4^{2-}$ measurements for 7 sites: Hyytiälä (FI-Hyy), Sodankylä (FI-Sod), Rimi (DK-Rim), Risbyholm (DK-Ris), Soroe (DK-Sor), Skyttorp (SE-Sk2) and Vielsalm (BE-Vie). For the CEAM lab, the uncertainty in $SO_4^{2-}$ measurements affected 2 sites, El Saler (ES-Els) and Las Majadas (ES-Lam) (see also Sect. 3.3.4)."*

**Text revised to:**

"Removal of the outlier NILU (7 out of 16) and CEAM (1 out of 3) data points with ion balance ratio < 0.5 improved both the slope (new slope = 0.90) and correlation (new $R^2$ = 0.78) (Figure 13C). An inspection of individual monthly site data reported by NILU showed that 15 % of aerosol $NH_4^+$ and 17 % of $NO_3^-$ concentrations were below 0.1 µg m$^{-3}$, compared with only 0.7 % of all $SO_4^{2-}$ data. This then points to a potential under-read in $NH_4^+$ and $NO_3^-$. Possible reasons include:

i)     loss of $NH_4^+$, $NO_3^-$ from filters (e.g. microbial degradation),

ii)     non-capture on the aerosol filters (e.g. aerosol filters installed wrong way round),

iii)     filters mixed up and wrong analysis performed on the acid and base-coated filters,

iv)     high blanks subtracted from already low concentrations at clean sites.

Possibilities still remain, however, of a potential over-read in $SO_4^{2-}$. The ion balance checks suggest increased uncertainty in the $NH_4^+$, $NO_3^-$ and $SO_4^{2-}$ measurements for 7 sites: Hyytiälä (FI-Hyy), Sodankylä (FI-Sod), Rimi (DK-Rim), Risbyholm (DK-Ris), Soroe (DK-Sor), Skyttorp (SE-Sk2) and Vielsalm (BE-Vie). Examination of monthly site data from CEAM showed only 1.5 % of aerosol $NH_4^+$ and 0.8 % of $NO_3^-$ concentrations below 0.1 µg $m^{-3}$, whereas all $SO_4^{2-}$
5   data were above 0.1 µg $m^{-3}$. For the CEAM lab, the uncertainty in $NH_4^+$, $NO_3^-$ and $SO_4^{2-}$ measurements affected 2 sites, El Saler (ES-Els) and Las Majadas (ES-Lam) (see also Sect. 3.3.4)."

10   *17. Page 24 – Line 16. Regarding the discussion of the CEAM and NILU Na+/Cl- regressions and the data below the 1:1 line, there does seem to be correlation among these outliers. Could this be an issue in the way filter blanks were applied? Perhaps an average Cl- blank biased high by an outlier was subtracted from all of the field measurements?*

15   **Author Response:**
Na and Cl data for NILU and CEAM are plotted separately below, which shows good correlation, but with a slope of 0.37 and 0.28, respectively.

[Figure]

20   **All data points**

As the reviewer suggests, the under-estimate in $Cl^-$ concentrations could be caused by high aerosol $Cl^-$ blank values. Aerosol blank values were unfortunately not reported by the labs, but denuder blanks were. Box-plots of blank $Cl^-$ data
25   from base-coated denuders ($K_2CO_3$-Glycerol, same coating as that used to coat the aerosol filters to collect $Cl^-$) from the network measurements are shown below. An average blank value was submitted by the laboratories for each month between 2006 to 2010. The plots show a larger range of blank $Cl^-$ values reported by the NILU lab (mean = 2.51 µg $Cl^-$ (0.05 – 5.22)), equivalent to an average air concentration of 0.17 µg $Cl^-$ $m^{-3}$ (range = 0.0 – 0.35 µg $Cl^-$ $m^{-3}$), based on 15 $m^3$ of air sampled over a month. So if the blank $Cl^-$ values in aerosol filters were similarly variable, then uncertainty in
30   blank $Cl^-$ values could contribute to the error. However, blank denuder $Cl^-$ values reported by CEAM (mean = 0.26 ug $Cl^-$) were less variable (range = 0.05 to 0.73 µg $Cl^-$, equivalent to 0.0 – 0.05 µg $Cl^-$ $m^{-3}$, based on 15 $m^3$ of air sampled over a month) and comparable with other labs.

[Figure]

*18. Page 26 – Line 19. Should "Figure 13A" be "Figure 14A"?*

**Author Response:**

Yes, thank you. Corrected to Figure 14A in paper.

*19. Page 26 – Line 23. ". . .with possible uptake and removal of NH3 from the atmosphere". Could results from the GRAMINAE project be cited here?*

**Author Response:**

15  Yes, thank you. Graminae reference added:

Sutton, M. A., Nemitz, E., Milford, C., Campbell, C., Erisman, J. W., Hensen, A., Cellier, P., David, M., Loubet, B., Personne, E., Schjoerring, J. K., Mattsson, M., Dorsey, J. R., Gallagher, M. W., Horvath, L., Weidinger, T., Meszaros, R., Dämmgen, U., Neftel, A., Herrmann, B., Lehman, B. E., Flechard, C., and Burkhardt, J.: Dynamics of ammonia exchange with cut grassland: synthesis of results and conclusions of the GRAMINAE Integrated Experiment, 20  Biogeosciences, 6, 2907–2934, https://doi.org/10.5194/bg-6-2907-2009, 2009.

*20. Page 26 – Line 24. Please change "thermodynamic shift to" to "thermodynamic shift of NH4NO3 to".*

25  **Author Response:**

Yes, thank you. Changed to "thermodynamic shift of $NH_4NO_3$ to"

*21. Page 26 – Section 3.5.2. To what extent could the temporal patterns in HNO3 be confounded by the collection of other oxidized N species on the denuder?*

**Author Response:**

Oxidized N species that are potentially also collected on the denuders include HONO, $NO_2$, $N_2O_5$ and PAN, as well as other inorganic and organic nitrogen species.

$NO_x$ emissions are dominated by vehicular sources which are not expected to show large seasonal variations. Co-collection of $NO_2$ (estimated to be between $3 – 5\%$ on carbonate coated denuders) should therefore exert negligible effect on the temporal patterns in $HNO_3$.

Of these, HONO is most likely to contribute the largest interference, since it is collected effectively on a carbonate coating.

Tropospheric HONO sources include chemical formation and direct emissions.

$$NO + OH\ (+ M) \rightarrow HONO\ (+ M)$$
$$H_2O + 2\ NO_2 \rightarrow HNO_3 + HONO$$

Emission sources include fossil fuel combustion, microbial activities in soil, and biomass burning.
The diurnal cycle in HONO is well established, but there remains limited information on its seasonal behaviour.

Li et al. (2018) reported maximum HONO concentrations in winter and elevated $HONO/NO_2$ ratio in summer at an urban site in China.
Wang et al. (2017) reported highest HONO concentration in autumn, and lowest in winter at an urban site in Beijing.

The temporal patterns in $HNO_3$ derived from the DELTA network is therefore likely to be HONO seasonal cycle superimposed onto the $HNO_3$ seasonal cycle.

Dandan Li, Likun Xue, Liang Wen, Xinfeng Wang, Tianshu Chen, Abdelwahid Mellouki, Jianmin Chen, Wenxing Wang, Characteristics and sources of nitrous acid in an urban atmosphere of northern China: Results from 1-yr continuous observations, Atmospheric Environment, 182, 2018, 296-306, https://doi.org/10.1016/j.atmosenv.2018.03.033

Jiaqi Wang, Xiaoshan Zhang, Jia Guo, Zhangwei Wang, Meigen Zhang, Observation of nitrous acid (HONO) in Beijing, China: Seasonal variation, nocturnal formation and daytime budget, Science of The Total Environment, 587–588, 2017, 350-359, https://doi.org/10.1016/j.scitotenv.2017.02.159.

**Additional text added:**

"Since the $HNO_3$ data is actually the sum of $HNO_3$ and HONO, with a small contribution from $NO_2$ (see Sect 2.2.3), the temporal patterns seen are likely to be the superimposed profiles of both $HNO_3$ and HONO. $NO_2$ are predominantly from vehicular sources which are not expected to show large seasonal variations and should therefore exert negligible effect on the temporal patterns in $HNO_3$. "

*22. Page 28 – Line 12. Consider changing "were provided by" to "were observed at".*

**Author Response:**

Yes, thank you. Changed to "were observed at"

*23. Page 29 – Section 3.6. See previous comment regarding inclusion of precipitation measurements*

**Author Response:**

Please see our response above regarding inclusion of precipitation measurements.

*24. Page 29 – Section 4.0. It seems like the material in this section could be greatly condensed and integrated into the Conclusions.*

**Author Response:**

10 Section 4.0 has been removed and integrated into the Conclusions – see author response to final reviewer comment.

*25. Page 30 – Line 10. The sentence beginning "However, SO2 (by mass). . .. . ." is quite lengthy.*

15 **Author Response:**

*"However, $SO_2$ (by mass) has a higher acidification potential ($1 \, kg \, SO_2 = 1.00 \, kg \, eq. \, SO_2$ than $NO_x$ ($1 \, kg \, NO_2 = 0.70 \, kg \, eq.$ $SO_2$ (see Hauschild and Wenzel, 1998), so $SO_2$ will remain important in contributing to exceedances of critical loads for acidification, estimated to be exceeded in 5 % of the European ecosystem area in 2015 (EEA, 2019)."*

Has been reworded to:

20 "However, since $SO_2$ has a higher acidification potential than $NO_x$ ($0.70 \, kg \, SO_2 = 1 \, kg \, eq. \, NO_2$ in acidity) (see Hauschild and Wenzel, 1998), $SO_2$ will remain important in contributing to exceedance of critical loads for acidification in European ecosystems (EEA, 2019). "

This paragraph has also been moved to conclusions, in response to the previous reviewer comment.

*26. Page 32 – Line 11. Some additional concluding comments, building on this key feature of the analysis, would be welcomed. For example, what does this shift from a sulfate dominated to nitrate dominated inorganic aerosol regime suggest for future European monitoring needs in support of ecological and human health protection?*
30 *What else can be gleaned from the current study, with respect to data quality, methods, and ability to resolve spatial and temporal patterns, that can inform future monitoring efforts?*

**Author Response:**

**See revised text below which addresses the reviewer comments (as also provided in response to reviewer 1):**

(Please note Section 4.0 has been removed and integrated into the Conclusions)

[revised manuscript text omitted]

5  Figure S4: Comparison of replicated DELTA monthly measurement of gases (a: NH$_3$, b: HNO$_3$, c: SO$_2$, g: HCl) and particulates (d: NH$_4^+$, e: NO$_3^-$, f: SO$_4^{2-}$, h: Cl$^-$, i: Na$^+$, j: Ca$^{2+}$, k: Mg$^{2+}$) at the UK Bush site (UK-Bu) and its' parallel site (UK-BuP). Independent samples t test was carried out on R ($p < 0.05$ = statistically significant difference in mean concentration between the replicates; $p > 0.05$ = not a statistically significant difference).

[Figure]

Figure S5: Comparison of replicated DELTA monthly measurement of gases (a: NH₃, *b: HNO₃, **c: SO₂) and particulates (d: NH₄⁺, e: NO₃⁻, **f: SO₄²⁻, g: Cl⁻, h: Na⁺, i: Ca²⁺, j: Mg²⁺) at the French Fougéres site (FR-Fgs) and its' parallel site (FR-FgsP). Independent samples t-test was carried out on R ($p < 0.05$ = statistically significant difference in mean concentration between the replicates; $p > 0.05$ = not a statistically significant difference).

10 *K₂CO₃/glycerol coated denuder used at FR-Fgs (HNO₃ determination includes potential inteference from co-collected oxidised N species) *vs* NaCl coated denuder at FR-FgsP.(selective for HNO₃). ** SO₂ is partially captured on NaCl coated denuders only, with break-through of SO₂ onto the aerosol filters resulting in larger particulate SO₄²⁻ concentrations than the Fr-Fgs site.

[Figure]

Figure S6: Comparison of replicated DELTA monthly measurement of gases (a: NH₃, b: HNO₃, c: SO₂, g: HCl) and particulates (d: NH₄⁺, e: NO₃⁻, f: SO₄²⁻, h: Cl⁻, i: Na⁺, j: Ca²⁺, k: Mg²⁺) at the Slovakian site (SK06) and its' parallel site (SK06P). Independent samples t test was carried out on R ($p < 0.05$ = statistically significant difference in mean concentration between the replicates; $p > 0.05$ = not a statistically significant difference).

[revised manuscript text omitted]

Under the 2012 UNECE Gothenburg protocol, EU member states must jointly cut their emissions of $NH_3$ by 6 % between 2005 and 2020. As a precursor to PM, controlling $NH_3$ is also important to reducing particle concentrations of $PM_{2.5}$ and $PM_{10}$. Indications from the current and projected trends in emissions of $SO_2$, $NO_x$ and $NH_3$ are that $NH_3$ and $NH_4NO_3$ will continue to dominate the inorganic pollution load over the next decades, contributing to ecosystem effects through acid and N deposition and to harmful effects on human health in the formation of fine PM. Changes in the relative concentrations of the pollutant gases also suggests that the deposition rates of $SO_2$ and $NH_3$ will increasingly be controlled by the molar ratio of $NH_3$ to combined acidity (sum of $SO_2$, $HNO_3$ and HCl) and deposition models should take these changes into account. The growing relative importance of $NH_3$ and $NH_4^+$ to total acidic and total N deposition indicates that strategies to tackle acidification and eutrophication need to include measures to abate emissions of $NH_3$ (Sutton and Howard, 2018).

Although the DELTA® approach is included in the EMEP Level 1 measurement strategy in the EMEP manual (EMEP, 2014), it has to date not been implemented across its networks. There is therefore a need for a monitoring network with sufficient coverage across Europe that provides long-term speciated data, to address uncertainties in deposition modelling, in particular of N species (EMEP, 2019). A target of > 125 sites (at a minimum site density of 1 site per 50,000 $km^2$) was previously suggested by Torseth and Hov (2003) as a reasonable number of sites to map the concentration fields across Europe. The EMEP daily filter-pack network continues to provide a large set of total nitrate and ammonium data, but which are not very helpful for understanding changes in the gas and aerosol phase N. Since the daily filter-pack approach is resource intensive and does not deliver required speciation, it would be better to redirect the measurement effort using a DELTA® approach. As demonstrated by the DELTA® network, it is possible to coordinate a network of that size across Europe with the participation of multiple laboratories, as is currently done with the daily filter-pack network.

[revised manuscript text omitted]